# Revealing the biophysics of lamina-associated domain formation by integrating theoretical modeling and high-resolution imaging

Monika Dhankhar [1,2,8], Zixian Guo [1,3,8], Aayush Kant [1,2,8], Ramin Basir [1,2], Rohit Joshi[1,2], Vinayak Vinayak [1,2], Su Chin Heo[4,5,6], Robert L. Mauck [1,4,5,6], Melike Lakadamyali [1,7] & Vivek B. Shenoy [1,2,3] ✉

Chromatin-lamina interactions regulate gene activity by forming lamina-associated domains (LADs), which contribute to cellular identity through gene repression. However, the strength of these interactions and their responsiveness to environmental cues remain unclear. Here, we develop a theoretical framework to predict LAD morphology in human mesenchymal stem cells (MSCs), whose differentiation potential depends on the stiffness of the microenvironment. Our model integrates chromatin-lamina interactions with histone modifications, revealing a bimodal distribution of chromatin-lamina affinity shaped by nuclear heterogeneities such as nuclear pores. We predict that contractility-driven translocation of histone deacetylase 3 (HDAC3) enhances chromatin-lamina affinity, leading to LAD thickening on soft substrates—a prediction validated through imaging and functional perturbations. Notably, in tendinosis, a condition marked by collagen degeneration and tissue softening, LAD thickening mirrors the behavior of MSCs on soft substrates, highlighting how microenvironmental mechanics influence genome organization and stem cell fate.

Tissue development, maintenance, and healing necessitate cells to continuously assess their three-dimensional microenvironment and regulate their activity based on local biochemical and mechanical signals[1]. The nucleus, as the hub of cellular genetic information, plays a vital role in dictating cell fate and function by altering the packing of chromatin in response to these stimuli[2]. Chromatin is spatially organized into euchromatin, which is loosely packed and transcriptionally active, and heterochromatin, which is densely packed and transcriptionally silent[3]. Crucially, at the nuclear periphery, interactions between chromatin and the nuclear lamina lead to the formation of lamina-associated domains (LADs), which exhibit heterochromatic characteristics and play a pivotal role in functionally organizing the genome by silencing lineage-inappropriate genes during differentiation[4–6]. LADs appear as key genomic features influenced by and influencing epigenomic states and the overall architecture of the genome. Despite previous research emphasizing the significance of

[1]Center for Engineering Mechanobiology, University of Pennsylvania, Philadelphia, PA, USA. [2]Department of Materials Science and Engineering, University of Pennsylvania, Philadelphia, PA, USA. [3]Department of Mechanical Engineering and Applied Mechanics, University of Pennsylvania, Philadelphia, PA, USA. [4]Department of Bioengineering, University of Pennsylvania, Philadelphia, PA, USA. [5]McKay Orthopaedic Research Laboratory, Department of Orthopaedic Surgery, Perelman School of Medicine, University of Pennsylvania, Philadelphia, PA, USA. [6]Translational Musculoskeletal Research Centre, Corporal Michael J, Crescenz VA Medical Centre, Philadelphia, PA, USA. [7]Department of Physiology, Perelman School of Medicine, University of Pennsylvania, Philadelphia, PA, USA. [8]These authors contributed equally: Monika Dhankhar, Zixian Guo, Aayush Kant ✉e-mail: vshenoy@upenn.edu

LADs, the quantitative strengths of the interactions between chromatin and the nuclear lamina and how they are influenced by cellular microenvironments remain unknown.

Recent advances in sequencing and imaging techniques, such as DamID, lamin B1 ChIP-seq, and super-resolution microscopy, have provided key insights into chromatin-lamina interactions. These methods have revealed the roles of specific DNA sequences, chromatin states, nuclear envelope proteins (such as lamins, LBR, and LAP2β), and histone modifications in tethering heterochromatin to the nuclear periphery[7]. Previous studies showed that heterochromatin marks including H3K9me2 and H3K27me3 are enriched at the nuclear periphery[8,9] and these marks are also over-represented within LADs, with H3K27me3 being particularly enriched at LAD borders[10,11]. In addition, previous work showed a role for both H3K9me2 and H3K27me3 for the formation and maintenance of lamina-proximal positioning of loci[12,13]. However, sequencing-based approaches often obscure cell-to-cell heterogeneity by averaging data across large populations. In contrast, single-cell sequencing and super-resolution imaging techniques like STORM and FISH offer insights at a single-cell resolution, providing a more granular view of chromatin-lamina interactions. These experiments have indeed shown cell to cell heterogeneity in chromatin folding and spatial organization. For example, STORM imaging showed that chromatin folds into heterogenous groups of nucleosome groups called nucleosome clutches[3]. In addition, chromatin tracing studies showed heterogeneity in the boundaries of topologically associating domains (TADs) at the single cell level[14].

To integrate and help interpret information across single cell and population averaged studies, experimental techniques need to be complemented by first-principles theories of chromatin-lamina interactions along with computational approaches to infer the principles underlying chromatin organization. Computational approaches based on polymer modeling, ranging from individual nucleosomes to whole-genome simulations, as well as continuum-scale simulations based on the Cahn-Hilliard equation, have enhanced our understanding of the principles governing chromatin organization[15–21]. While polymer models treat chromatin as a flexible sequence of beads, focusing on its folding and interactions with binding proteins[22], continuum models have been used to describe the nucleus-wide segregation of distinct chromosome territories via global free-energy minimization[23], without resolving sub-chromosome features such as chromatin compartments or nucleosome clutches. Other approaches, like the strings and binders switch (SBS) model[24], focus on chromatin-protein interactions, while the loop extrusion model explains the formation of topologically associating domains (TADs) via cohesin complexes[25]. Polymer models for interactions of chromatin with the nuclear lamina have been developed[26,27], but they do not account for non-equilibrium contributions, such as the dynamics of methylation and acetylation reactions. As a result, they are unable to predict the shapes and sizes of LADs. The absence of these dynamic considerations limits our understanding of how molecular interactions, such as chromatin-lamina affinity, regulate the morphology of LADs and adapt to changes in the microenvironment or cell signaling.

Here, we introduce a multimodal framework to decipher the spatially heterogeneous biophysical mechanisms governing chromatin organization, with a specific focus on the strength of chromatin-lamina tethering and its modulation in response to the cell's microenvironment. Our framework integrates super-resolution microscopy with a mesoscale mathematical model that accounts for interactions between chromatin segments and chromatin-lamina, as well as the kinetics of histone methylation and acetylation. This approach allows us to quantitatively extract key biophysical parameters, such as chromatin-lamina interaction strengths and histone methylation rates, from single-cell imaging data, thereby offering insights into how chromatin-lamina interactions vary across individual cells and are modulated by

changes in extracellular stiffness and pharmacological treatments targeting epigenetic regulators. We show that chromatin-lamina interactions, combined with histone methylation and acetylation, mechanistically shape the morphology of heterochromatin both in the nuclear interior and periphery. Our findings suggest that softening of extracellular matrix, which occurs in diseases such as tendinosis, regulates chromatin-lamina interactions and epigenetic regulation, resulting in nucleus-wide spatial chromatin reorganization. We test this hypothesis by using our predictive framework to analyze cells from healthy donors and patients with tendinosis. Our multifaceted integrative framework, which combines theoretical, and imaging approaches, provides insights into specific mechanosensitive molecular alterations driving genome organization, and has wider implications for translating these findings into understanding pathological conditions impacting the extracellular environment.

## Results

### Chromatin organization features interior heterochromatin domains and LADs with distinctive sizes

To understand how chromatin is spatially segregated into heterochromatin and euchromatin, we used STORM microscopy[3,28] to visualize the distribution of histone H2B in the nuclei of human mesenchymal stromal cells (hMSCs) cultured on a glass substrate at nanoscale resolution (~20 nm). To gain quantitative insights into the spatial chromatin organization, we used Voronoi tessellation-based segmentation to identify regions of high H2B density (defined as inverse of the Voronoi polygon area[28]) as shown in Fig. 1A. These high H2B density regions correspond to tightly packed heterochromatin (HC, red in Fig. 1A), while low-density regions indicate euchromatin (EC) (discussed in detail in Methods section). These images reveal that localizations of H2B form distinct clusters, that can be visualized both at the periphery of the nucleus and away from the periphery in the nucleus interior (Fig. 1A, zoom). We used a density-based spatial clustering algorithm to identify the HC clusters from the H2B localizations and classified them as peripheral or interior HC domains depending on their distance from the nuclear periphery in the STORM images.

Further, we observed that interior HC domains exhibit a characteristic size distribution. On quantifying the sizes of the interior domains, by estimating their area to measure their radii (Fig. 1A, Bottom), we observed that the interior domain radii follow a statistical distribution about a characteristic mean radius of ~50 nm (Fig. 1B, Top). Similarly, for the peripheral heterochromatin domains (herein referred to as lamina associated domains, LADs), we measured their thickness from the nuclear periphery (Fig. 1A, Right) as described in Methods section. We find that the distribution of LAD thicknesses along the nuclear periphery also exhibits a characteristic mean of ~200 nm (Fig. 1B, Bottom). Notably, in addition to their characteristic thickness, LADs exhibit diverse shapes ranging from small bead-like to elongated domains (Fig. 1A, Right).

Despite the literature focus on the spatial organization of chromatin, its quantification in terms of the characteristic sizes and shapes of the HC domains as well as the underlying biophysical origins of such size selection are not yet known. To uncover the key biophysical determinants of the chromatin organization in terms of the morphologies of interior HC domains and LADs, we next discuss the theoretical framework we have developed based on first-principles of non-equilibrium thermodynamics and active epigenetic reactions.

### Theory uncovers the biophysical determinants of heterochromatin shape and size

To investigate how nuclear-scale chromatin organization emerges from local biophysical and biochemical interactions, we develop a reaction-diffusion–based mesoscale model of chromatin organization. This mathematical model incorporates the energetics of chromatin-

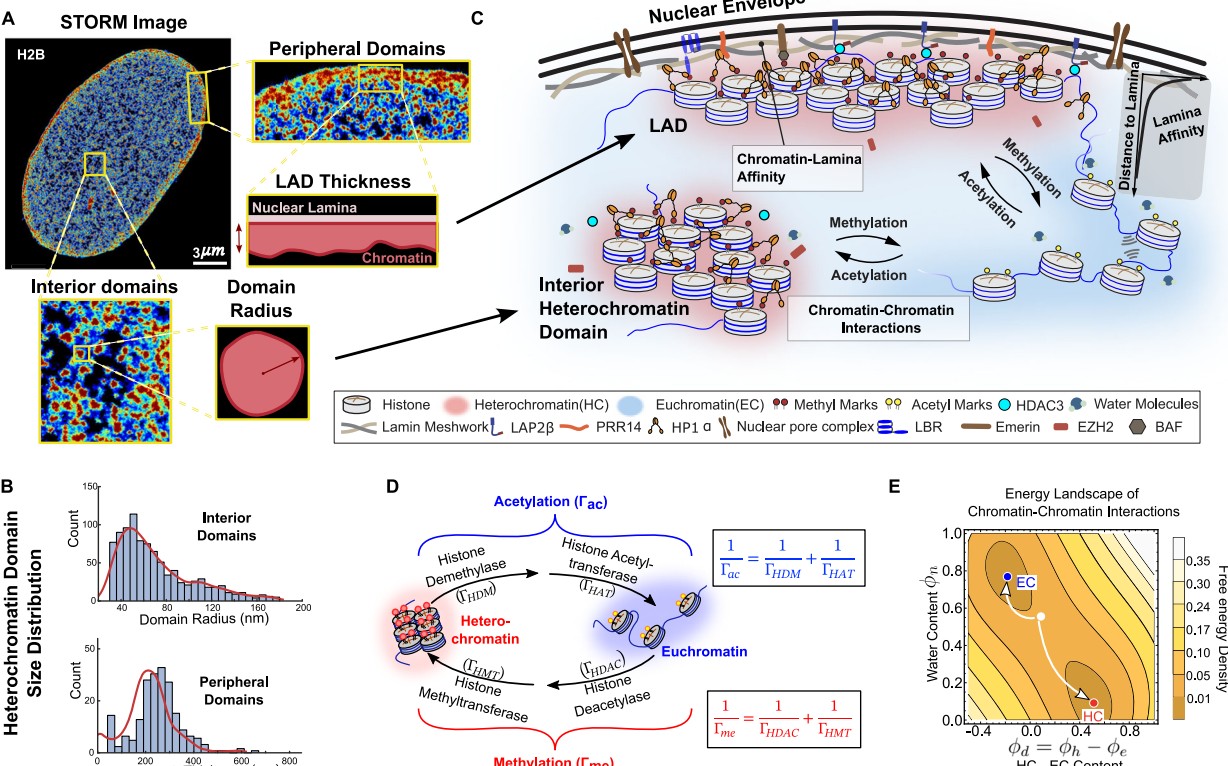

**Fig. 1 | Schematic of the numerical model capturing the formation of interior HC domains and peripheral HC domains. A** Representative Voronoi density rendering of STORM images of hMSC nuclei, color-coded to indicate histone H2B density levels (blue for low density and red for high density). **B** The distribution of heterochromatin domain sizes obtained from STORM images in interior (Top) and at periphery (Bottom). Notably the distributions show a characteristic mean size. **C** Chromatin-chromatin interactions result in segregation of chromatin into euchromatin and heterochromatin phases. These interactions incorporate chromatin-chromatin interactions, including those mediated by crosslinking molecules such as HP1α, as well as the direct interactions between segments of chromatin. The model also includes the interactions between chromatin and the lamina, mediated by anchoring proteins such as LAP2α/β and LBR. These

chromatin-lamina interactions, localized at the nuclear periphery, result in the formation of heterochromatin rich lamina-associated domains (LADs). **D** Epigenetic factors, such as HDAC and HMT, regulate acetylation and methylation reactions that allow interconversion of heterochromatin and euchromatin phases, captured via first-order reaction kinetics. The diffusion of water and epigenetic markers is included in the continuum model. The anchoring of chromatin to the nuclear lamina is also mediated by HDAC3[7,40]. **E** The contour plot of free energy density shows the two wells (local minima) corresponding to the two stable phases of chromatin – euchromatin (blue) and heterochromatin (red). We schematically show how an initial homogeneous distribution of chromatin (white circle) will evolve towards the two energy wells.

chromatin and chromatin-lamina interactions coupled with the kinetics of nucleoplasm diffusion and spatially inhomogeneous interconversion of acetylated and methylated histones via epigenetic regulation (Fig. 1C). The coupled energetic and kinetic contributions lead to a far-from-equilibrium segregation of chromatin into compacted heterochromatin domains in the interior as well as at the periphery of the nucleus. We integrate our theoretical predictions with the experimentally observed chromatin organization quantified via super-resolution STORM imaging to extract the genome-wide distributions of key biophysical parameters, including the strength of chromatin-lamina interactions and methylation rates. This integrative framework allows us to evaluate changes in the distributions of these biophysical parameters, shedding light on the mechanisms underlying the chromatin reorganization due to pharmacological treatments, biophysical signaling perturbations and disease conditions. A comprehensive description of the modeling framework, experimental protocols, and imaging techniques is provided in the Methods section.

## Model predicts the formation of stable interior domains and LADs observed through super-resolution imaging

The mathematical model developed in Methods section (Eqs. (3)–(5)) is numerically solved to predict chromatin distribution in the nucleus (details in the Supplementary Information Section 8). We observe the

segregation of water-poor, condensed heterochromatin region (corresponding to the heterochromatic energy well (red) in Fig. 1E from a distinctly water-rich euchromatin region (euchromatic energy well (blue) in Fig. 1E. The energetically driven segregation leads to the formation of nascent domains of heterochromatin which grow over time, attaining a quasi-periodic distribution both in the interior and at the nuclear periphery, as shown in Fig. 2B. The domains of heterochromatin observed at the steady-state are stable against further coarsening. Consistent with our simulations, histone density mapping obtained from super-resolution STORM imaging of nuclei in cells plated on 2D substrates[29] confirms that chromatin organizes into compacted heterochromatin, characterized by high histone density, which is clearly distinguishable from the low-density euchromatin region (Fig. 2A). We observe heterochromatin domains localized within the interior of the nucleus and along the nuclear periphery (LADs) in both simulations and high-resolution STORM images (Fig. 2A, B). Our model predicts that at steady-state, the size distribution of the heterochromatin domains in the interior of the nucleus shows a characteristic mean (Supplementary Fig. 13), in excellent agreement with the size-distribution of condensed chromatin domains observed via STORM imaging (Fig. 1B). Importantly, the model also predicts that near the nuclear periphery, LADs form either discrete domains or continuous layers (Fig. 2B) and show a size distribution

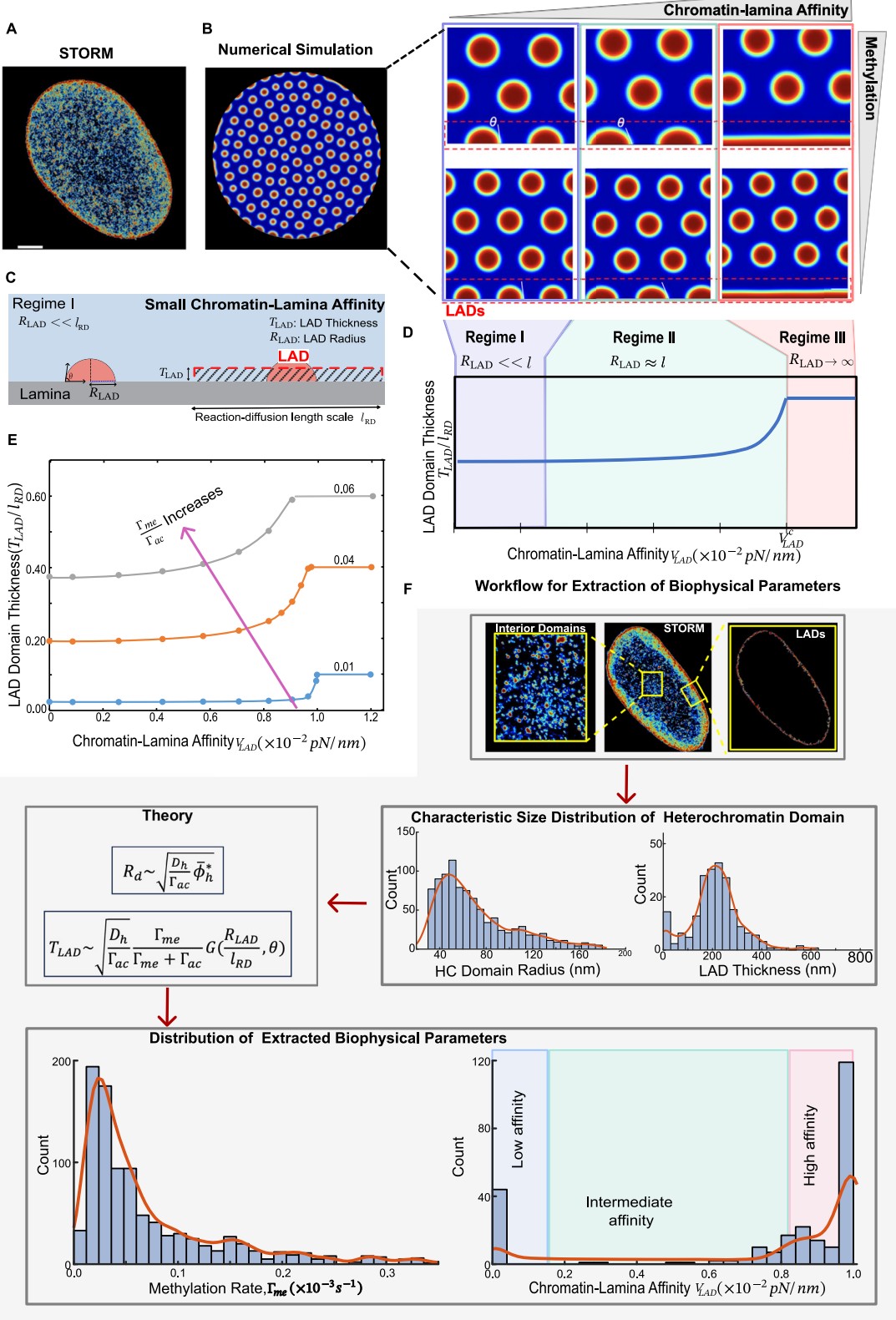

that peaks at a characteristic length-scale, as shown in (Supplementary Fig. 13). The characteristic size-scale of the LADs is also confirmed from super-resolution imaging of cells (Fig. 1B).

Upon compaction, chromatin can be partitioned into heterochromatin domains within the nucleus interior or LADs along the nuclear periphery. Thus, the total heterochromatin content of the nucleus is shared between the interior domains and LADs such that,

$$Total\ HC = HC^{interior} + HC^{Periphery} \tag{1}$$

In Eq. (1), 'Total HC' is the average heterochromatin content in the nucleus $\bar{\phi}_h$ times the total nuclear area. In the presence of epigenetic

**Fig. 2 | The morphology of interior heterochromatin domains and LADs is regulated by the methylation levels and the strength of chromatin-lamina affinity. A** Representative Voronoi density rendering of H2B STORM images, color-coded to indicate density levels (blue for low density and red for high density). **B** Steady-state chromatin organization predicted by numerical simulations showing phase separation into compacted heterochromatin domains (red) and loosely packed euchromatin domains (blue) and the formation of heterochromatin-rich LADs at the nuclear periphery. Numerical simulations show the synergistic effects of chromatin-lamina affinity and methylation rates on LADs and inner hetero-chromatin domain morphology. At low levels of chromatin-lamina affinity, as the methylation level increases, the radius of the discrete LADs increases at a fixed value of the contact angle ($\theta$). The morphology of LADs is determined by both the level of methylation and chromatin-lamina affinity. Low affinity results in the formation of isolated LADs, while at greater levels of affinity LAD spread along the lamina. **C** Schematic illustrating the morphology of LAD at small level of chromatin-lamina affinity. **D** The scaling relationship between the thickness of LAD and chromatin-lamina affinity at a given methylation level showing three distinct regimes. **E** The scaling relation between the thickness of LAD and chromatin-lamina affinity at varying methylation levels. **F** Workflow of the theoretical framework for extracting methylation rates and chromatin-lamina affinity by integrating super resolution STORM images of DNA with theoretical analyses. Within the interior of the nucleus, STORM image analysis gives the distribution of heterochromatin domain radii ($R_d$), and at periphery, it provides the distribution of LAD thickness $T_{LAD}$. The combination of these distributions with the theoretical framework predicts the corresponding distribution of histone methylation rates $\Gamma_{me}$ and the chromatin-lamina affinities $V_{LAD}$. Notably the distribution of $V_{LAD}$ is bimodal. The red curve in all plots shows the smoothed density plot. All source data are provided as a Source Data file.

reactions ($\Gamma_{me}$: methylation rate and $\Gamma_{ac}$: the acetylation rate), the average amount of heterochromatin is determined by the epigenetic reaction rates, $\bar{\phi}_h = \frac{\Gamma_{me}}{\Gamma_{me} + \Gamma_{ac}}$ (detailed derivation in Supplementary Information Section 2); the average amount of heterochromatin in the nucleus increases with the repressive methylation rate. Note here that the repressive methylation rate ($\Gamma_{me}$) comprises deacetylation of histones via HDACs and the addition of repressive methyl groups (e.g. H3K9me2/3 and H3K27me3) by HMTs (as shown in Fig. 1D). Next, we determine the biophysical parameters that govern the size of hetero-chromatin domains in the interior and at the periphery of the nucleus, incorporated in Eq. (1).

### Sizes of the interior heterochromatin domains are determined by the balance between epigenetic reactions and diffusion kinetics

To understand chromatin organization in the nucleus, we first consider the formation, growth and stabilization of heterochromatin domain in the nucleus's interior, away from the nuclear periphery. The initial homogeneous state of chromatin with HC and EC volume fractions, $(\bar{\phi}_h, \bar{\phi}_e)$, is determined by the reactions and do not lie in either of the energy wells (white circle in Fig. 1E). However, the free energy can be lowered by nucleating heterochromatin domains (red circle in Fig. 1E) surrounded in the immediate vicinity by euchromatin (blue circle in Fig. 1E). The resulting concentration gradient (as shown in Supplementary Fig. 2) drives a flow of methyl marks toward the domain, promoting its growth (Supplementary Fig. 2). At the same time, within the growing domains, methylated histones are epigenetically modified into acetylated euchromatin. These newly acetylated histones are pushed out of the domain, due to preferential interactions of like-marked histones. This epigenetic reaction-driven outflux of euchro-matin marks opposes the diffusive influx of methyl marks, establishing a kinetic balance that leads to a quasi-periodic distribution of hetero-chromatin domains stable against further growth (Fig. 2B). A detailed description of the growth and stabilization of the heterochromatin domains is given in the Supplementary Information (Supplementary Section 3). The kinetic balance gives the characteristic steady-state radius $R_d$ for interior heterochromatin domain, away from the nuclear periphery as,

$$R_d = \sqrt{3\frac{D_h \bar{\phi}_h^*}{\Gamma_{ac}}} \qquad (2)$$

Here, $D_h$ is the diffusion constant, $\Gamma_{ac}$ is the acetylation rate, and $\bar{\phi}_h^*$ is the far-field heterochromatin concentration- that is, the steady-state heterochromatin concentration in the euchromatin regions distant from heterochromatin domain. A more detailed discussion of $\bar{\phi}_h^*$ including how it is affected by peripheral sequestration, is provided in Supplementary Information Section 4 (see also Supplementary Fig. 2). The $D_h \bar{\phi}_h^*$ term in Eq. (2) corresponds the heterochromatin influx which is opposed by the acetylation driven outflux (corresponding to

the $\Gamma_{ac}$ term), determining the size of the interior heterochromatin domains. Next, we determine the size scaling of the LADs at the nuclear periphery with respect to the histone methylation rate and the strength of chromatin-lamina interactions.

### LAD shapes are determined by chromatin-lamina affinity, while their sizes are regulated by epigenetic reactions

The presence of chromatin-anchoring proteins such as LAP2$\beta$, HDAC3 and LBR[6,7,30] that sequester chromatin to the lamina is captured in our model through the chromatin-lamina interaction energy per unit surface area of nuclear lamina, $V_{LAD}$ (Eq. (3)). Notably, the chromatin-lamina interaction energy reflects the collective impact of several chromatin-anchoring proteins, and any alterations in it signify either a change in the number or affinity of these proteins. Increase in chromatin-lamina interactions lead to preferential localization of histones modified with repressive methylation marks along the nuclear periphery, thereby affecting the shape of the LADs, which we quantify via their thickness $T_{LAD}$ (Fig. 2C).The dependence of LAD thickness $T_{LAD}$ on the chromatin-lamina affinity, relative to chromatin-chromatin interaction strength $V_{LAD}$ determined numerically is shown in Fig. 2D (refer Supplementary Information Section 6 for detailed numerical methodology). At low chromatin-lamina affinity relative to chromatin-chromatin interactions, methylated histones preferentially interact with other methylated histones rather than with the lamina leading to the formation of bead-like peripheral heterochromatin domains, with minimal effective LAD thickness (Fig. 2B, C) with contact angles, $\theta$ close to 90°. A slight increase in chromatin-lamina affinity within this regime (Regime I, Fig. 2D) does not significantly affect LAD morphology, as chromatin-chromatin interactions dominate over chromatin-lamina interactions. However, as the chromatin-lamina affinity continues to increase the HC domains spread with smaller contact angles and elongate along the lamina forming more lamellar LADs. This decreases the spacing between the LADs effectively increasing the LAD thickness (Regime II, Fig. 2D). In Regime III, high chromatin-lamina affinity results in a near-continuous HC layer along the lamina (Fig. 2D), with the maximal LAD thickness.

In addition to the chromatin-lamina interactions, the epigenetic reactions also contribute to the increase in LAD size. The stable size of the peripheral heterochromatin domains is determined by the kinetic balance between the diffusion-driven influx of methylated histones (due to their preferential interactions with each other) and the acetylation-driven outflux of histones as in the case of interior domains. Thus, as the overall histone methylation rate increases (or acetylation rate decreases) the size of LADs increases, effectively increasing their thickness. Numerically (as depicted in Fig. 2E), we find that for all levels of chromatin-lamina interactions, an increase in methylation rate enhances heterochromatin content in the nucleus, amplifying LAD thickness. Further, we theoretically and numerically confirm that the thickness of the LADs follows distinct scaling relationship with both epigenetic reactions and chromatin-lamina affinity

as discussed in detail in the Supplementary Information (Supplementary Section 7). Thus, the chromatin-lamina interactions and the epigenetic reaction rates, respectively, determine the shapes and the sizes of the peripheral heterochromatin domains, effectively determining the LAD thickness.

To summarize, we have theoretically and numerically determined the sizes of the heterochromatin domains within the nuclear interior (Eq. 2) and at the periphery (Fig. 2E, Supplementary Eq. (S36)) taking into account their interdependence (Eq. (1)). We find that the heterochromatin domain sizes are self-consistently determined by the biophysical parameters governing epigenetic reaction kinetics ($\Gamma_{me}$) and the strength of chromatin-lamina interactions $V_{LAD}$. Thus, using the distribution of the interior domain radius and LAD thickness obtained from the quantification of STORM images (discussed in Methods section), Eqs. (1), (2) and Supplementary Eq. (S36) can be solved simultaneously to determine the nucleus-wide distribution of biophysical parameters, as we discuss next.

### Extracting biophysical parameters from super-resolution images using theory

In this section, we use the theoretical relations derived in previous section to extract the epigenetic reaction rates of methylation $\Gamma_{me}$ and the strength of the chromatin-lamina interactions relative to chromatin-chromatin interactions $V_{LAD}$ from 2D STORM images. The intranuclear environment is inherently heterogeneous due to spatial variations in epigenetic regulators, chromatin anchoring proteins, and the presence of other nuclear constituents. For instance, the Lamin B proteins, which are involved in heterochromatin association with the lamina, are heterogeneously distributed in the nuclear envelope[7]. Additionally, nuclear membrane structures such as nuclear pore complexes[31] may exhibit relatively weaker binding affinity for chromatin segments. Similarly, spatial variations in the availability of epigenetic factors like HDAC (Figs. 5F and 6A)[32] can result in heterogeneous histone methylation rates.

We adopt a two-step procedure (Fig. 2F) to extract the distribution of histone methylation rate $\Gamma_{me}$ and strength of chromatin-lamina interactions relative to the chromatin-chromatin interactions $V_{LAD}$ from STORM images. The theoretical expressions derived in previous section (which assume homogeneous histone methylation rate and chromatin-lamina affinity) are taken to hold locally in the interior and the nuclear periphery. To visualize the spatial chromatin organization within cells, we generate nuclear-wide H2B density heatmaps via Voronoi tessellation-based segmentation of super-resolution images[28,29]. By analyzing these images (as described in Methods Section), we obtain the statistical distributions of the interior domain radii $R_d$ and the LAD thickness $T_{LAD}$. The obtained interior domain and LAD sizes are locally related to the methylation rate $\Gamma_{me}$ via Eq. (2) and Fig. 2E, Supplementary Eq. (S36), allowing us to extract the nucleus-wide distribution of $\Gamma_{me}$. Note that to infer $\Gamma_{me}$ and $V_{LAD}$ we used fixed values for $D_h$ and $\Gamma_{ac}$, which were determined based on data from the literature (discussed in Supplementary Information section 9). Using the extracted distributions of $\Gamma_{me}$ and LAD thickness, we use Supplementary Eq. (S36) (numerically depicted in Fig. 2E) to deduce the distribution of chromatin-lamina affinity relative to chromatin-chromatin interaction strength $V_{LAD}$ along the nuclear periphery.

Further, to assess the reliability of the extracted distributions of $\Gamma_{me}$ and $V_{LAD}$, we compare simulation-based predictions of chromatin organization with experimental observations, despite the simplifying assumptions involved in the theoretical derivation of Eqs. (1), (2) and Supplementary Eq. (S36) (numerically depicted in Fig. 2E). Specifically, the mean values of the extracted parameters $\Gamma_{me}$ and $V_{LAD}$ are used as inputs to simulate chromatin organization in the nucleus via numerical solutions of the phase-field equations (Eqs. (3)–(5)) using COMSOL Multiphysics. The pharmacological or biophysical perturbation driven changes in the distributions of biophysical parameters, extracted from

the theoretical framework, are incorporated into the simulation by a proportionate change in the mean of the parameters $\Gamma_{me}$ and $V_{LAD}$. We then quantify the simulation-predicted chromatin reorganization and assess the relative changes in LAD and interior domain sizes, validating these against STORM-imaged data (Figs. 3D, 4C, 5B, 6D and 7C; Supplementary Table 4). To incorporate spatial heterogeneity, local values of $\Gamma_{me}$ and $V_{LAD}$ are sampled from a normal distribution, whose mean matches the values obtained from STORM observations. This approach allows for biologically realistic variability in the simulated biophysical landscape while preserving consistency with the experimentally inferred averages. Heterochromatin domains in the simulation emerge naturally from the interplay of spatially varying parameters and reaction-diffusion dynamics. These domains can also interact freely with one another, capturing the intrinsically emergent behavior characteristic of chromatin organization.

Thus, our theory-based image analysis framework leverages the mechanistic principles regulating chromatin organization to utilize super-resolution imaging of chromatin as a readout of the underlying biophysical parameters.

### Chromatin-lamina affinity extracted from STORM images shows a bimodal distribution

We first implement the integrated theoretical framework outlined in previous section to analyze chromatin organization in hMSC nuclei cultured on soft hydrogel (taken as the control)[29] to extract the distributions of histone methylation rate, $\Gamma_{me}$ and chromatin-lamina affinity, $V_{LAD}$. We previously showed that soft hydrogels promote an increase in heterochromatin formation and higher amount of heterochromatin at the nuclear periphery compared to stiff hydrogels[29]. The quantification of heterochromatin domain size scales reveals a skewed distribution of the interior domain radii $R_d$ (Fig. 2F, Supplementary Fig. 17) and LAD thickness $T_{LAD}$ (Fig. 2F, Supplementary Fig. 17). Across all nuclei the distributions show a comparable characteristic mean (Supplementary Fig. 17). Next, we employ the theoretical relations between the heterochromatin domain size scales and the biophysical parameters methylation rate $\Gamma_{me}$ and chromatin-lamina affinity $V_{LAD}$ for both the interior domains and the LADs (Eq. (2), Fig. 2E, Supplementary Eq. (S36)). The distribution of $\Gamma_{me}$ across all control-treated nuclei exhibits a similar skewness (Fig. 2F, Supplementary Fig. 17) with a comparable mean value (supplementary Fig. 17).

Notably, the distribution of chromatin-lamina affinity $V_{LAD}$ is bimodal, with two distinctive peaks at very low and very high affinities (Fig. 2F). The peak associated with vanishing chromatin-lamina affinity likely corresponds to the regions along nuclear periphery deficient in chromatin binding proteins, such as nuclear pore complex regions which are not known to interact strongly with heterochromatin[33–35]. Spatially, regions with low chromatin-lamina affinity typically correspond to lamina regions with sparse H2B localizations. The second peak occurs at a very high chromatin-lamina interaction strength, comparable to the chromatin-chromatin interactions, attributable to the presence of chromatin anchoring proteins with a strong affinity for methylated histones. In STORM images, such regions spatially correlate with dense H2B localizations, and high LAD thickness along the lamina. Between the two peaks, we observe a spectrum of chromatin-lamina affinities where the chromatin-lamina interactions are weaker but still nonzero.

The predicted spectrum of chromatin-lamina affinities aligns qualitatively with a range of chromatin-LaminB1 interaction strengths observed experimentally using LaminB1-chromatin immunoprecipitation (LB1 ChIP) in multiple cell types (Supplementary Fig. 21)[36]. Regions along the chromatin polymer exhibiting strong LaminB1 association are classified as Type 1 LADs, resembling the peak of high chromatin-lamina affinity we observe. Furthermore, as we predict, these LADs have highly compacted chromatin apparent from their very low chromatin accessibility[36]. Regions of chromatin polymer exhibiting a weaker (but nonzero) LaminB1 association, with marginally increased

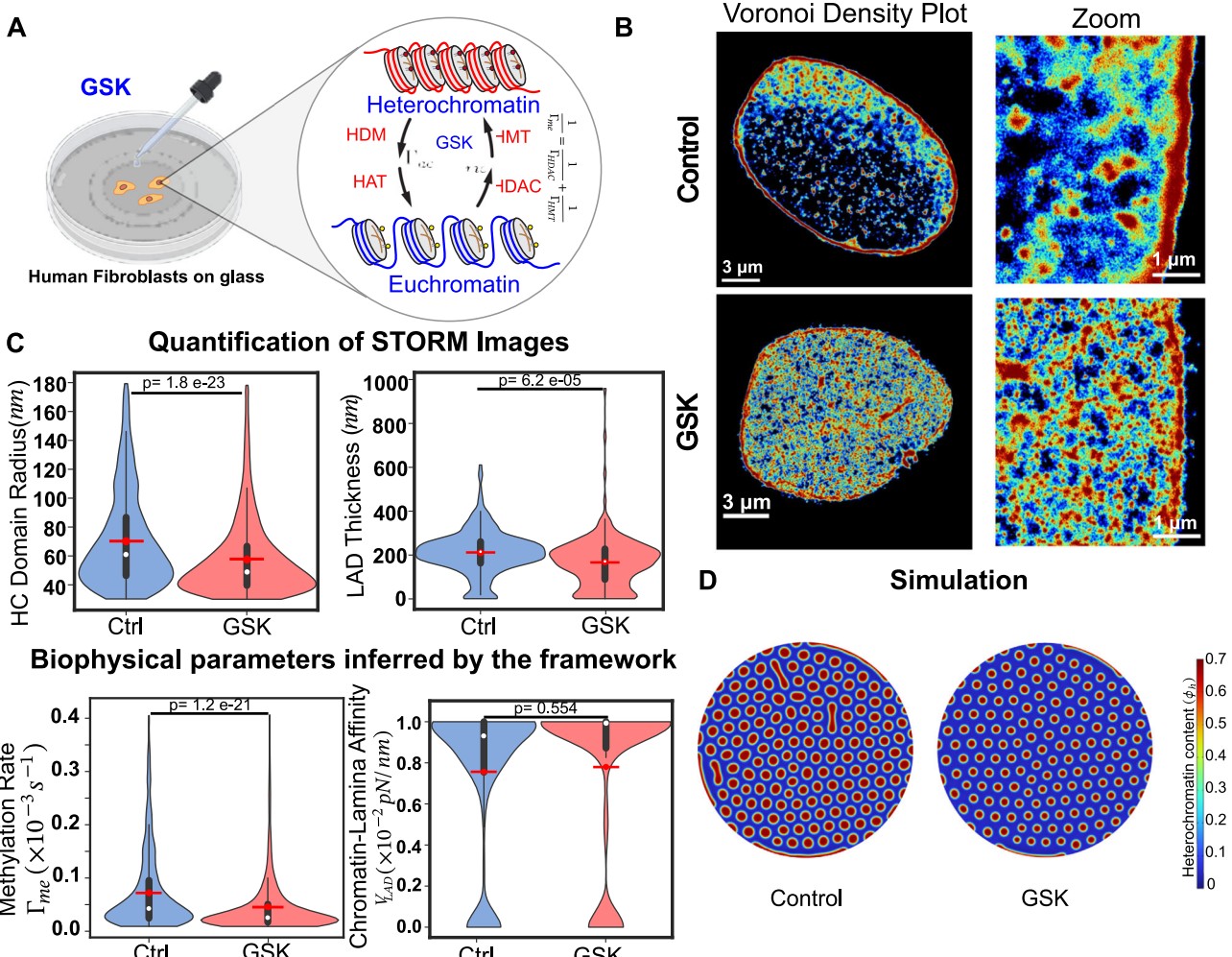

**Fig. 3 | GSK treatment inhibits histone methyltransferase EZH2 resulting in smaller LADs. A** A schematic showing the role of GSK, which inhibits histone methyltransferases (HMTs) and induces chromatin decompaction. Petri-dish and dropper icons are created in BioRender. Member, S. (2025) https://BioRender.com/3eozx52. **B** Super-resolution STORM images revealing the reorganization of chromatin upon GSK treatment. (Left panels) Voronoi density plots representing the extent of spatial compaction of chromatin in control (top) and GSK treated nuclei (bottom) along with a zoomed-in view (right panels). **C** Quantification of STORM images in the control and GSK treated nuclei showing the distribution of interior domain radii (left top) and LAD thickness (right top). Distribution of the corresponding methylation rate and the chromatin-lamina affinity (bottom panels) obtained by our integrative theoretical framework. After GSK treatment, the methylation rate decreases although there is no effect on chromatin-lamina affinity (Control: $n = 938$ loci, 5 nuclei; GSK-treated: $n = 1766$ loci, 4 nuclei; unpaired two tail test). All violin plots show a symmetric kernel density estimate (outline), the quartiles (black boxplot), the median (white dot), and the mean (red dot with line). **D** Numerical predictions of the size of heterochromatin (in red) in the control and GSK-treated nuclei using the extracted biophysical parameters. All source data are provided as a Source Data file.

chromatin accessibility are classified as Type 2 LADs[36] and resemble the range of low to intermediate chromatin-lamina affinities we observe. We emphasize that this comparison is an inferred mapping based on qualitative similarity in the predicted spatial distribution of chromatin-lamina affinities and the genomic distribution of Lamin B1-associated domains, rather than a direct measure of interaction strength. This comparison is discussed in detail in Supplementary Information Section 11.

In the subsequent sections, we will examine chromatin reorganization following known pharmacological treatments and biomechanical stimuli. We use the STORM images with our theoretical framework to predict perturbations in the spatial distribution of the histone methylation rate and chromatin-lamina affinity. Our generalized framework remains blind to the specific treatments the nuclei undergo, ensuring that the analysis is not influenced by prior knowledge across datasets. Using numerical simulations, we then confirm that the extracted distributions of the biophysical parameters do indeed drive the observed chromatin reorganization.

## Model captures the differential LAD regulation by distinct pharmacological treatments

Next, we comparatively analyze chromatin organization in control nuclei and those with pharmacologically perturbed biophysical parameters. From our theoretical findings in previous sections, we see that LAD morphologies are regulated synergistically by histone methylation and chromatin-lamina interactions. By targeting the epigenetic reactions, we previously inhibited the activities of enhancer of zeste homologue 2 (EZH2, a histone methyltransferase) via GSK343 treatment (Fig. 3A)[37,38], and histone deacetylase (HDAC) via pan-HDAC inhibitor trichostatin A (TSA, Fig. 4A)[3,28,39]. Both treatments lead to a physical decompaction of chromatin resulting in reduced chromatin density throughout the nucleus[3,14,29]. Since histone deacetylases and methyltransferases contribute to the overall rate of histone methylation in our model (Fig. 1D), we extract the resulting change in $\Gamma_{me}$. Moreover, HDAC3 also interacts with the inner nuclear membrane protein LAP2β in tethering LADs[40]. Hence, we hypothesize that TSA treatment may also alter chromatin-lamina affinity. Thus, the

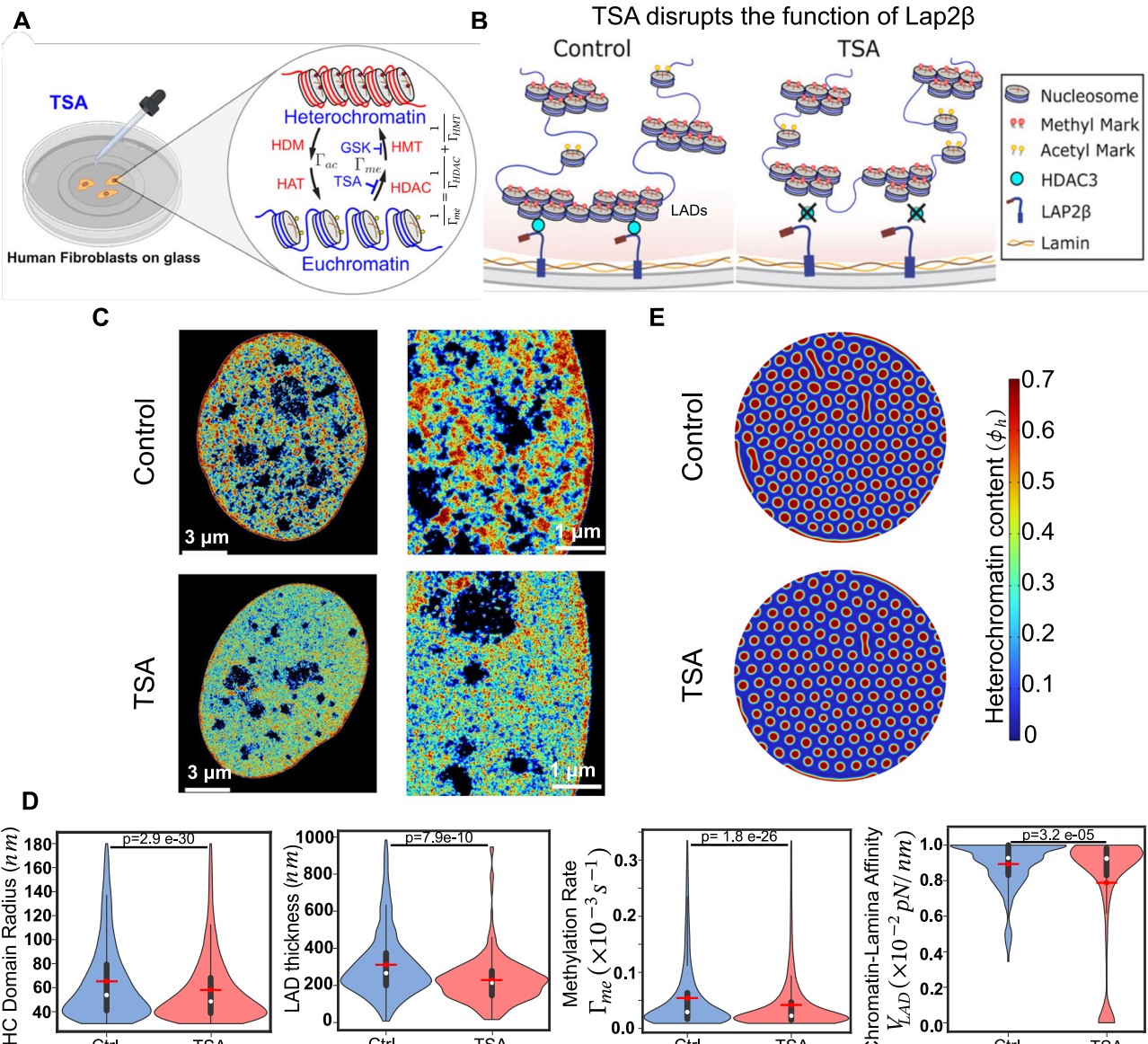

**Fig. 4 | TSA treatment inhibits HDAC, resulting in reduction of methylation rate and chromatin-lamina affinity. A** A schematic showing the role of TSA, which inhibits histone deacetylase (HDAC) and causes chromatin decompaction. Petridish and dropper icons are created in BioRender. Member, S. (2025) https://BioRender.com/3eozx52. **B** A schematic showing the mechanisms of tethering chromatin to the nuclear periphery via HDAC3 and nuclear envelope proteins (e.g., LAP2b). TSA treatment disrupts the association of HADCs with LAP2b and causes the detachment of LADs thus decreasing chromatin-lamina affinity. **C** Superresolution STORM images reveal the reorganization of chromatin upon TSA treatment. (Left panels) Voronoi density plots representing the extent of spatial compaction of chromatin in the control (top) and TSA-treated nuclei (bottom), along with a zoomed-in view (right panels). **D** Our theoretical framework reveals that upon TSA treatment, the interior heterochromatin domain radii, LAD thickness, methylation rate and chromatin-lamina affinity all decrease. (Control: $n = 4223$ loci, 6 nuclei; TSA-treated: 5279 loci, 5 nuclei, unpaired two tail test). All violin plots show a symmetric kernel density estimate (outline), the quartiles (black boxplot), the median (white dot), and the mean (red dot with line). **E** Numerical prediction using the extracted biophysical parameters as input showing smaller HC domains in the interior and periphery in TSA-treated nuclei. All source data are provided as a Source Data file.

comparison of differential biophysical parameter modulations after GSK and TSA treatment serves to validate the accuracy of our parameter extraction procedure.

**HMT inhibition affects only histone methylation and not chromatin-lamina affinity.** EZH2 promotes the transfer of methyl groups to histone H3 at lysine 27 (H3K27)[41,42], which is a mark that is enriched in the interior and also overrepresented at the periphery of nucleus[8,10]. We observe that its inhibition by GSK343 in hMSCs cultured on soft hydrogels results in genome-wide chromatin decompaction compared to untreated control cells cultured on soft hydrogels (Fig. 3B)[29]. From the STORM images of H2B localizations (Fig. 3B), we quantify the

nuclear-wide distribution of heterochromatin domain sizes (as described in Methods section) after control and GSK treatments (Fig. 3C, top panel). Upon GSK treatment the mean interior heterochromatin domain radius $R_d$ decreases by ~18% (Fig. 3C, top panel). Next, by identifying the heterochromatin domains near the nuclear periphery (Methods Section) we obtain the distribution of their average thickness $T_{LAD}$ (as defined in Supplementary Information section 5.2). GSK treatment results in decreased LAD thickness, with a mean LAD thickness approximately 20% lower than that of control-treated nuclei (Fig. 3C, top panel).

Having measured the sizes of the interior domains and LADs, we quantify the distribution of histone methylation rate, $\Gamma_{me}$ and

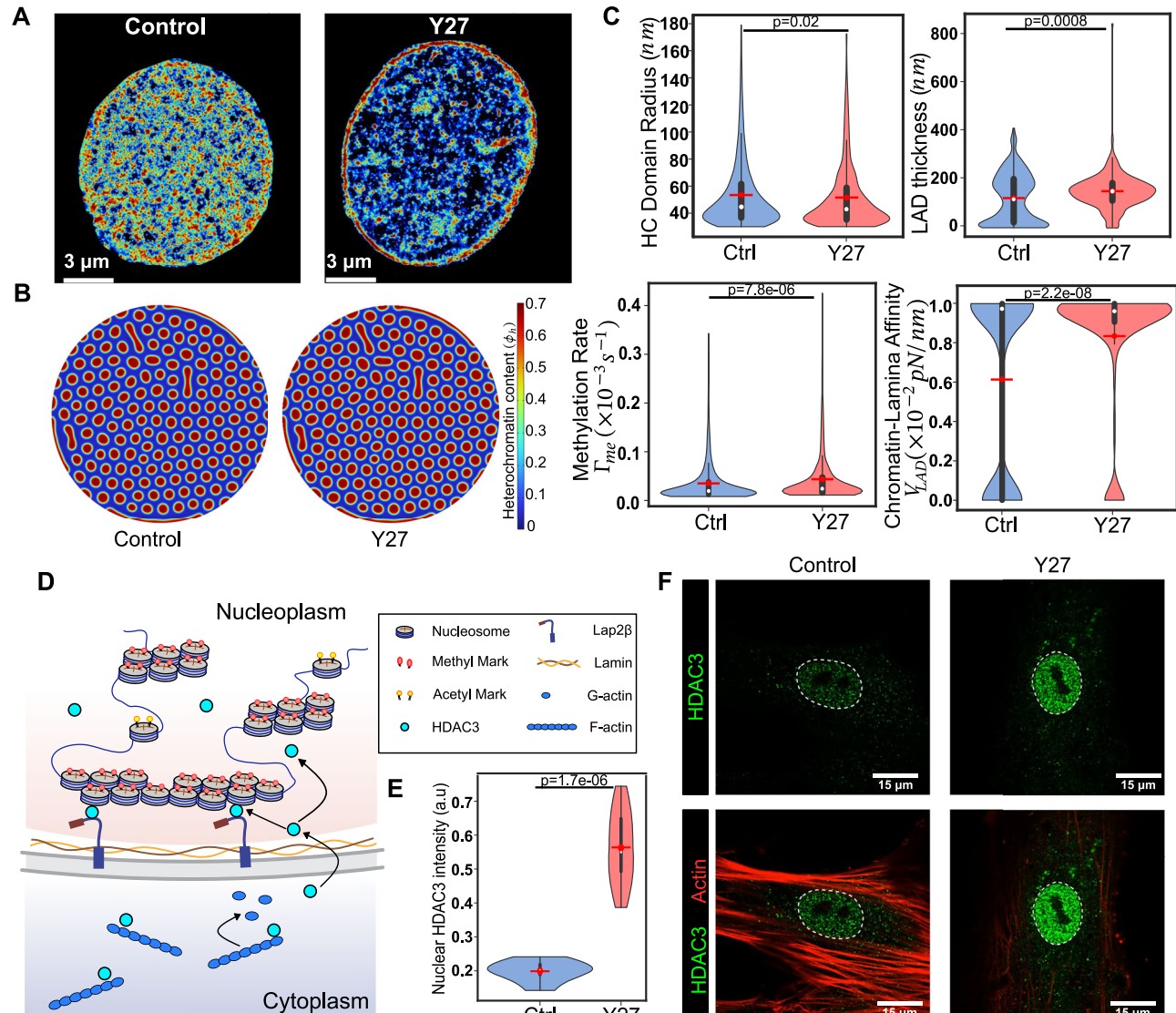

**Fig. 5 | Inhibition of cellular contractility increases the nuclear localization of HDAC3, along with higher methylation rate and chromatin-lamina affinity, consequently resulting in larger LADs.** A reduction in cellular contractility induces the nuclear translocation of HDAC3, leading to chromatin compaction. **A** Voronoi density plots representing the spatial reorganization of chromatin in the control and Y27-treated cells. **B** Numerical simulation showing increased domain size and LAD thickness in Y27-treated nuclei. The methylation rate and chromatin-lamina affinity extracted from the STORM images are used as inputs in the model. **C** Quantitative analysis of STORM images reveals a decrease in the mean heterochromatin domain radius and an increase in LAD thickness, methylation rate, and chromatin-lamina affinity upon Y27 treatment. (Control: $n$ = 3325 loci, 4 nuclei; Y27-

treated: $n$ = 930 loci, 5 nuclei, unpaired two tail test). All violin plots show a symmetric kernel density estimate (outline), the quartiles (black boxplot), the median (white dot), and the mean (red dot with line). **D** A schematic showing the nuclear localization of HDAC3 in response to contractility abrogation. **E** HDAC3 fluorescence intensity quantification, showing increase in nuclear HDAC3 upon Y27 treatment. (Control: $n$ = 10 nuclei; Y27-treated: $n$ = 10 nuclei, unpaired two tail test). All violin plots show a symmetric kernel density estimate (outline), the quartiles (black boxplot), the median (white dot), and the mean (red dot with line). **F** Representative immunofluorescence images for HDAC3 and actin in control and Y27-treated nuclei. The outline of nucleus (white dashed lines) is identified using DAPI. All source data are provided as a Source Data file.

chromatin-lamina affinity, $V_{LAD}$ in GSK and control treated nuclei using our framework (Fig. 3C, bottom panel). Our theoretical framework predicts an approximately 36% lower mean methylation rate in GSK-treated cells than in untreated cells (Fig. 3C, bottom panel). This is expected since GSK inhibits EZH2, a histone methyltransferase that contributes to methylation kinetics in our model (Fig. 1D). However, we do not observe a significant change in chromatin-lamina affinity after GSK treatment (Fig. 3C, bottom panel), indicating that inhibition of EZH2 does not influence the binding of chromatin to lamina according to our predictive framework.

Using the extracted changes in the parameters $\Gamma_{me}$ and $V_{LAD}$ we simulate the change in chromatin organization using Eqs. (3)–(5). The

numerical predictions of chromatin reorganization (Fig. 3D) in GSK-treated and control nuclei closely match the observed in vitro chromatin reorganization (Fig. 3B), validating our parameter extraction framework. The numerically observed mean heterochromatin domain sizes in GSK-treated nuclei decreased by approximately 20% and 30% in the interior and at the periphery of the nucleus, respectively (Supplementary Fig. 14), comparable to the findings in cells.

**HDAC inhibition reduces overall histone methylation and chromatin-lamina affinity.** Similar to HMTs, HDACs play an important role in chromatin organization and gene expression via removal of acetyl groups from histones promoting chromatin condensation

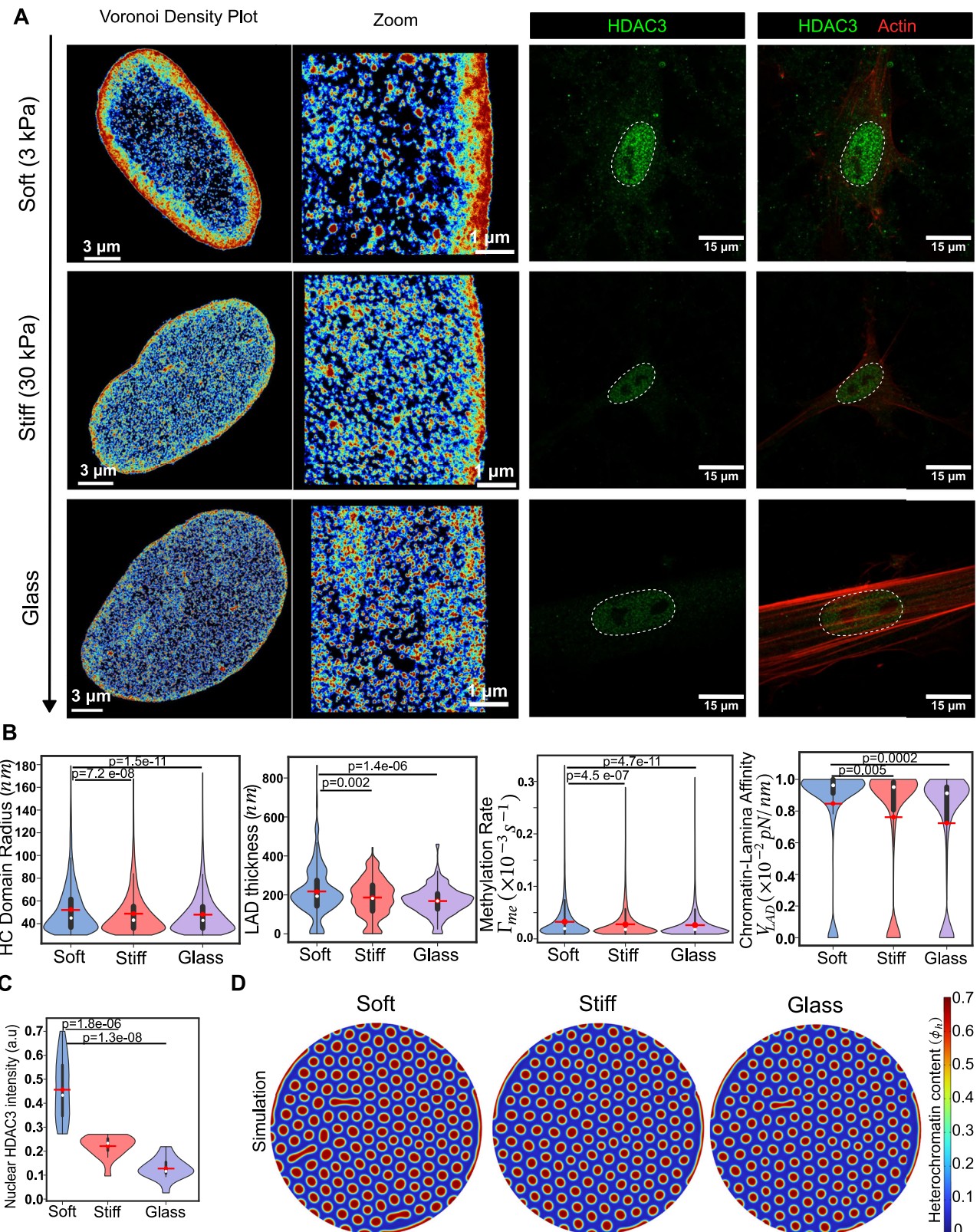

(Fig. 4A)[43]. However, unlike EZH2, HDAC3 also contributes to the tethering of chromatin to the nuclear periphery via interactions with LAP2β, a nuclear envelope protein (Fig. 4B)[40]. Using super-resolution imaging of histones and DNA, we previously showed that inhibition of HDAC via TSA treatment in human fibroblast (hFb) nuclei results in reduced chromatin compaction compared to that in untreated control nuclei (Fig. 4C)[28]. Quantitative analysis of domain sizes after TSA

treatment confirmed these results, revealing reductions of ~11% and 26% in the interior domain sizes and LAD thicknesses, respectively (Fig. 4D). We next extract the changes in histone methylation rate and chromatin-lamina affinity (Fig. 4D) using the theoretical framework developed in previous sections. Our analysis predicts that the histone methylation rate is reduced by ~22% in TSA-treated nuclei compared to that in untreated nuclei (Fig. 4D). The TSA-treatment mediated

**Fig. 6 | Higher substrate stiffness elevates cellular contractility resulting in smaller LADs. A** (Left panels) Voronoi density plots of STORM images of hMSCs representing the spatial compaction of chromatin on soft (3 kPa), stiff (30 kPa) and glass (rigid) substrates, with an enlarged view. Cellular contractility is elevated on stiffer substrates, leading to the translocation of HDAC3 into the cytoplasm from the nucleus, which results in chromatin decompaction. Similarly, there is an increase in HDAC3 and EZH2 levels on softer substrates, which drives chromatin condensation and the attachment of peripheral heterochromatin from the lamina. **A** (Right panels) Representative immunofluorescence images for HDAC3 and actin on varying substrate stiffness. The outline of nucleus (white dashed lines) is identified using DAPI. **B** Quantitative analysis of the STORM images, integrated with theory, showing the distribution of heterochromatin domain radii, LAD thickness,

methylation rates, and chromatin-lamina affinity on substrates of different stiffnesses. (Soft: $n = 1375$, 5 nuclei; stiff: $n = 4365$, 5 nuclei; glass: $n = 11273$, 4 nuclei, unpaired two tail test). All violin plots show a symmetric kernel density estimate (outline), the quartiles (black boxplot), the median (white dot), and the mean (red dot with line). **C** HDAC3 fluorescence intensity quantification, showing decrease in nuclear HDAC3 with increasing substrate stiffness. (Soft: $n = 16$ nuclei; stiff: $n = 15$ nuclei; glass: $n = 15$ nuclei, unpaired two tail test). All violin plots show a symmetric kernel density estimate (outline), the quartiles (black boxplot), the median (white dot), and the mean (red dot with line). **D** Numerical simulation, using the biophysical parameters from the STORM images, showing a decreasing trend in the HC domain (in red) size with increasing substrate stiffness. All source data are provided as a Source Data file.

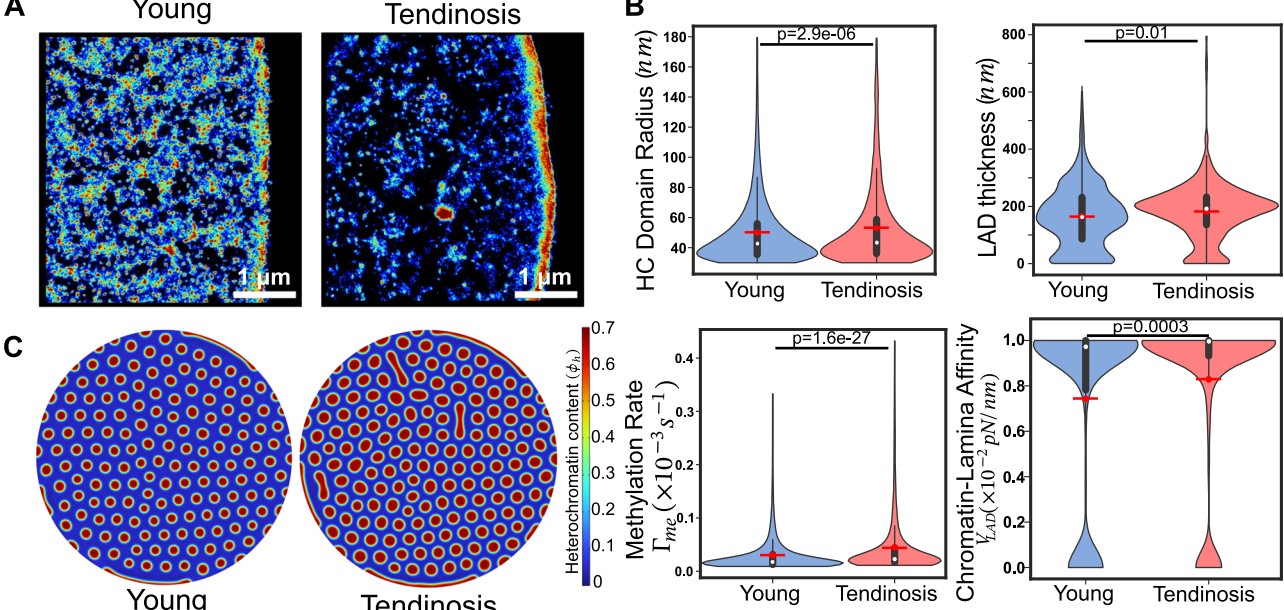

**Fig. 7 | Tendinosis induces soft-like phenotype leading to larger LADs.**
**A** Enlarged images of the Voronoi density plot for young and diseased (tendinosis) human tenocytes (hTCs), showing changes in chromatin condensation.
**B** Quantitative analysis reveals an increase in the mean heterochromatin domain radius, LAD thickness, methylation rate and chromatin-lamina affinity. (Healthy: $n = 14429$ loci, 10 nuclei; Diseased: $n = 1992$, 9 nuclei, unpaired two tail test). All

violin plots show a symmetric kernel density estimate (outline), the quartiles (black boxplot), the median (white dot), and the mean (red dot with line). **C** Numerical simulation, employing biophysical parameters extracted from the integration of STORM images and theory as inputs, shows an increase in both the HC domain size and the LAD thickness. All source data are provided as a Source Data file.

inhibition of histone deacetylation, which is a sub step of the histone methylation reaction in our model (Fig. 1D) explains the predicted reduction in $\Gamma_{me}$. Additionally, the chromatin-lamina affinity in TSA-treated nuclei reduced by approximately 11% compared to that in untreated nuclei (Fig. 4D). The decrease in chromatin-lamina affinity explains why the reduction in sizes of peripheral LADs is more pronounced than that in the interior domains. HDAC3 at the nuclear periphery interacts with chromatin anchoring protein LAP2$\beta$ and mediates gene repression[40]. Abrogation of HDAC enzymatic activity via TSA treatment has been shown to release specific endogenous gene loci from the inner nuclear membrane mediating their movement toward the nucleus interior[44–46]. Thus, the predicted decline in chromatin-lamina affinity, observed using our theoretical framework, is consistent with the known involvement of HDACs, such as HDAC3 in mediating the interactions between heterochromatin and the nuclear lamina[40]. Complementing our LAD-focused imaging and modeling approach, a recent Hi-C study[47] reported widespread TSA-induced changes in chromatin contact patterns and histone acetylation, providing a genome-wide perspective on the multiscale effects of HDAC inhibition. The numerical simulations (Fig. 4E), using the predicted changes in parameters, replicate the chromatin reorganization observed in cells upon TSA-treatment (Fig. 4C) with reductions of ~11%

and 22% in the interior domain and LAD sizes, respectively (Supplementary Fig. 14). Overall, by quantitative analysis of STORM images, our theoretical framework accurately predicts the changes in biophysical parameters governing chromatin organization after GSK and TSA treatments. Notably, we predict that both TSA and GSK treatments reduce histone methylation rates, as expected in our model where deacetylation and methyltransferase activity contribute to the overall histone methylation rate (Fig. 1D). However, TSA treatment, but not GSK treatment, reduces the strength of chromatin-lamina interactions. Indeed, EZH2, which is inhibited by GSK, is not a known contributor to the binding of chromatin to the lamina. Thus, solely based on STORM images, the model is able to account for the distinct actions of biochemical mechanisms altered during the GSK and TSA treatments without any prior knowledge of the specific pharmacological treatments applied. Next, we test our theory on upstream pharmacological perturbations that influence chromatin organization.

## Abrogation of cell contractility alters chromatin-lamina interactions and LAD formation

The epigenetic regulators targeted in previous section, EZH2 and HDAC3 exhibit a mechanosensitive nucleocytoplasmic partitioning[48]. For instance, mechanical stresses such as compressive or fluid shear

stresses have been shown to alter the nuclear availability of EZH2[49]. Furthermore, we have previously shown that cytoskeletal contractility alters the nucleocytoplasmic shuttling of HDAC3 such that under low contractility conditions, the nuclear localization of HDAC3 increases (Fig. 5D)[50]. Thus, to study how upstream cellular perturbations such as cell contractility can alter chromatin organization, we analyzed STORM images of Y27632 (Y27; a specific inhibitor of rho-associated protein kinase II (ROCK II)) treated hMSCs cultured on stiffening hydrogels (30 kPa)[29].

As we previously demonstrated[29], STORM imaging of H2B localization in Y27-treated nuclei reveals global chromatin reorganization (Fig. 5A). The quantification of domain sizes reveals that the mean radius of interior domains in Y27-treated nuclei was approximately 4.3% lower than that in control nuclei (Fig. 5C). On the other hand, the mean LAD thickness increased by ~19.4% after Y27 treatment. The abrogation of contractility upon Y27 treatment enhances the nuclear localization of HDAC3 (Fig. 5E, F), which is consistent with previous observations[50]. The increase in HDAC3 increases the rate of histone deacetylation, thereby amplifying the overall rate of methylation (as defined in Methods Section, Fig. 1D) and therefore the heterochromatin content. However, as discussed in previous sections, significant sequestering of heterochromatin along the periphery leaves less heterochromatin for the nucleus interior, leading to a reduction in the size of the interior domains. To confirm this interplay, we quantify the ratio of peripheral to total localization of heterochromatin in the nuclei from the STORM images. We find that Y27-treated nuclei exhibit significantly more peripheral heterochromatin (Supplementary Fig. 18) than control nuclei, suggesting a stronger sequestration of chromatin to the lamina. Thus, our model explains how a pronounced increase in LAD thickness after Y27 treatment drives a reduction in the size of interior heterochromatin domains, despite an increase in global chromatin condensation.

Using our theoretical framework, we next extract the distributions of histone methylation rate and chromatin-lamina affinity. We observe that the mean histone methylation rate in Y27-treated nuclei increases by ~12% compared to untreated nuclei (Fig. 5C). Moreover, the chromatin-lamina affinity of Y27-treated nuclei was predicted to be approximately 26% higher compared to untreated nuclei (Fig. 5C). The observed increase in nuclear HDAC3 due to the loss of contractility (Fig. 5E) correlates with increased histone deacetylation, which in turn is associated with a higher predicted histone methylation rate in our model. The implications of HDAC3 nuclear localization for chromatin-lamina interactions are further discussed in the Discussion section. Numerical simulations (Fig. 5B) incorporating the spatially heterogeneous biophysical parameters extracted from the theoretical framework recapitulate the in vitro chromatin reorganization (Fig. 5A). We observe an ~7% and 15% increase in domain sizes in the nuclear interior and at the periphery, respectively (Supplementary Fig. 14).

Altogether, our theoretical framework is able to predict the changes in histone methylation rate and chromatin-lamina affinity even when chromatin organization is perturbed via upstream modulations such as abrogation of contractility. As a further examination of the framework and to demonstrate its robustness beyond pharmacological treatments, we next perturb the cytoskeletal contractility via biophysical mechanisms such as altering the biomechanical environment of the cell.

### Chromatin-lamina affinity decreases with the stiffness of the culture substrates

The biomechanical environment, which is altered after disease, injury, or age-dependent tissue-degeneration, drives cytoskeletal remodeling in cells. For instance, in stiff environments, cell contractility increases due to stable substrate adhesions and stress fibers[51]. We have previously demonstrated the regulatory role of in-vitro substrate stiffness on chromatin compaction in the interior and periphery of hMSC

nuclei[29]. The heatmaps of H2B localization densities, obtained via STORM imaging of hMSCs cultured on glass (~70 GPa), stiff (30 kPa), and soft (3 kPa) hydrogel substrates are shown in Fig. 6A. Substrate-dependent chromatin remodeling affects both interior domains and LADs (Fig. 6A, left panels). Quantitative analysis reveals an ~6.8% and 8% decrease in the mean interior domain radius in nuclei on stiff and glass substrates, respectively, compared to those on soft substrates (Fig. 6B). At the nuclear periphery, compared to nuclei on soft substrates in nuclei on stiff and glass substrates the thickness of LADs is reduced by ~14% and 21.5%, respectively (Fig. 6B). Thus, an increase in substrate stiffness results in reduced heterochromatin domain sizes in the interior and periphery of the nucleus.

Next, using our theoretical framework, we extract the nucleus-wide distribution of histone methylation rate and chromatin-lamina affinity. Compared to nuclei cultured on soft substrates, we predict that those on stiff substrates exhibited a 17% decrease in the mean methylation rate. In the nuclei on glass the mean methylation rate was further reduced, showing a 19.9% decrease compared to that in the nuclei on soft substrates (Fig. 6B). Furthermore, we observe a 10% and 14.5% lower chromatin lamina affinity on stiff and glass substrates, respectively, than on soft substrates (Fig. 6B). The increase in stiffness from soft to stiff to glass substrates drives a progressive increase in cell contractility thereby reducing the intranuclear HDAC3 (Fig. 6A, right panels). The observed decrease in nuclear HDAC3 with increasing substrate stiffness (Fig. 6C) is associated with reduced histone deacetylation, which is reflected in a lower predicted $\Gamma_{me}$ in our model. The potential role of HDAC3 in mediating chromatin-lamina interactions and LAD organization is discussed in the Discussion section.

Finally, the distribution of the extracted methylation rate and chromatin-lamina affinity is used in our numerical simulations to predict the changes in chromatin reorganization (Fig. 6D). The numerical simulations show an ~7% (10%) and 15% (25%) decrease in the interior and LAD sizes on stiff (glass) substrate respectively, compared to those on soft substrate (Supplementary Fig. 14), in agreement with the cell measurements (Fig. 6B).

Thus, our theoretical framework accurately predicts changes in biophysical parameters associated with chromatin reorganization in cells cultured in vitro on substrates of varying stiffness. This validation leads us to enquire whether the in vivo chromatin reorganization due to tissue degenerative diseases such as tendinosis can be assessed with our theoretical framework to extract the modulation of biophysical parameters.

### Tendinosis induces chromatin reorganization by altering microenvironmental stiffness

Injury and disease cause pathological degeneration of the fibrous extracellular matrix, altering the chemo-mechanical microenvironment of cells. Previous studies have shown that tissue degeneration in tendinosis reduces local tissue stiffness[52,53]. We have demonstrated that human tenocyte (hTC) nuclei isolated from patients with tendinosis exhibit aberrant chromatin reorganization akin to that of healthy tenocytes cultured in a soft matrix[29]. Super-resolution STORM images of H2B localization densities in diseased hTC nuclei exhibit a global chromatin organization distinct from healthy nuclei (Fig. 7A). We observe approximately 5.8% larger interior domains and 10.2% higher LAD thickness in diseased hTCs than in healthy cells (Fig. 7B). Thus, tendinosis correlates with increased heterochromatin domain sizes both in the nucleus interior and periphery, resembling cells on soft substrates.

We next quantify the distributions of histone methylation rate and chromatin lamina affinity in healthy and diseased tenocytes. Our theoretical framework reveals that in nuclei with tendinosis the mean histone methylation rate increases by ~30% while the chromatin-lamina affinity rises by 11.6% compared to that in healthy counterparts (Fig. 7B). Note that although the methylation rate increased

significantly, the increase in the size of interior heterochromatin domains was relatively low. This is due to the significant sequestering of heterochromatin to the nuclear periphery driven by the increase in chromatin-lamina affinity, as discussed in previous sections. This is further discussed in detail in the Supplementary Information (Supplementary Section 10). Thus, the changes in biophysical parameters after in-vivo tissue degeneration mirror those in cells cultured in vitro on soft substrates. The induction of a soft phenotype in the tenocytes can be attributed to tendinosis driven collagen degeneration resulting in a reduction in tissue stiffness, as discussed in previous section. Incorporating the extracted changes in the histone methylation rate and chromatin-lamina affinity in our numerical simulations, we observe that the mean interior domain radius increases by 16%, while the mean LAD thickness increases by 28% (Supplementary Fig. 14) comparable to what was observed in vitro (Fig. 7B).

Altogether, we find that our theoretical framework offers a methodology to accurately extract the changes in biophysical parameters in a versatile range of nuclear perturbations, not only in vitro but also after in vivo pathological degeneration, such as in tendinosis.

## Discussion

In this study, we integrate theoretical and numerical modeling with super-resolution microscopy to introduce a multimodal framework for analyzing chromatin organization images and extracting the distribution of chromatin-lamina interaction strengths along the nuclear periphery. We begin by developing a coarse-grained mesoscale mathematical model predicting the nucleus-wide chromatin organization in response to changes in the cell microenvironment, epigenetic drugs, and pharmacological treatments affecting cell contractility. Our model captures the emergence of densely compacted heterochromatin domains of characteristic sizes in both the nucleus interior and at the periphery. Leveraging these insights, we predict the spatial chromatin reorganization due to diverse cues such as disease-driven extracellular matrix degradation. Building upon our theoretical insights, our predictive framework utilizes the morphology of peripheral heterochromatin domains, observed via super-resolution microscopy, as a measure of the distribution of chromatin-lamina interaction strengths. Our integrated framework enables the evaluation of the roles of molecular mechanisms mediating the chromatin-lamina interactions in diverse cellular microenvironments.

Chromatin-lamina interactions are pivotal in regulating spatial chromatin organization and gene expression during cell development, differentiation, and disease[7,54]. The roles of such interactions in determining the three-dimensional genome organization, especially in the vicinity of the nuclear periphery, have been experimentally explored using sequencing-based techniques such as DamID[10,55–57] and lamin B1 ChIP-seq[36,58,59] as well as super-resolution imaging-based techniques such as STORM imaging coupled with fluorescence in-situ hybridization (FISH)[13,60] and transmission electron microscopy[61]. These observations highlight the role of nuclear envelope proteins, particularly lamins and associated factors such as LBR and LAP2β, as well as specific histone modifications like H3K9me2/3 and H3K27me3, in the formation and maintenance of heterochromatin domains that interact with lamina to enable gene silencing. Polymer-physics based computational models as well as data-driven modeling approaches have previously been used to infer the quantitative roles chromatin-lamina energetic interactions in peri-laminar genome compartmentalization[23,62–67]. However, these models rely on bulk sequencing techniques like ChIP-seq, which miss single-cell variations, resulting in averaged-out effects. These methods lack the ability to capture cell-to-cell heterogeneity in chromatin-lamina interactions. Our work leverages super-resolution imaging of lamina-associated domains (LADs) to capture the single-cell distribution of chromatin-lamina interactions, providing a more detailed view of how these interactions vary across individual cells.

Also, these polymer models do not incorporate the non-equilibrium kinetics arising from active epigenetic regulation and diffusion kinetics, which we show critically regulate chromatin organization. Moreover, our coarse-grained continuum model, unlike the previous fine-grained polymer models, can concurrently characterize the nucleus-wide chromatin organization, i.e. both in the nucleus interior and the periphery. This presents a significant advantage by facilitating direct comparison with super-resolution imaging and quantitatively extracting the biophysical parameters governing the chromatin organization, which has previously been a challenge.

Our theoretical model predicts that LADs can display a spectrum of morphologies, ranging from small, isolated clusters to elongated formations along the lamina depending on (i) the chromatin-lamina affinity, and (ii) epigenetic reaction rates. While higher chromatin-lamina affinity causes spreading of LADs into elongated lamellar domains, histone methylation rate prominently impacts the LAD size. Hence, the average thickness of the peripheral heterochromatin domains is a composite outcome of both these parameters, as shown in the theoretically predicted phase diagram (Supplementary Fig. 10). Our model further predicts that the size-scale of interior heterochromatin domains is uniquely regulated by the epigenetic reaction kinetics. Thus, given a distribution of size-scales of the interior and peripheral heterochromatin domains, obtained from STORM imaging, our predictive framework can extract the nucleus-wide distribution of epigenetic reaction rates and chromatin-lamina affinity (Fig. 2F). The key predictions of our multimodal framework applied to nuclei undergoing pharmacological and biophysical perturbations are:

- The distribution of chromatin-lamina affinity is bimodal, suggesting presence of regions along the lamina with very strong chromatin binding interspersed with regions with minimal LAD sequestering.
- The predictive framework, without any prior knowledge, distinguishes between TSA and GSK treatments, identifying that while both affect epigenetic kinetics, only TSA alters chromatin-lamina affinity, in agreement with the known role of HDAC3 in LAD formation at the lamina.
- The framework identifies upstream pharmacological perturbations in cell contractility alter both epigenetic regulation and chromatin-lamina affinity, consistent with observed nucleo-cytoplasmic HDAC3 shuttling.
- An increase in substrate stiffness alters chromatin organization by reducing the overall histone methylation rate and decreasing chromatin-lamina affinity, mediated by cytoskeletal contractility driven HDAC3 efflux from the nucleus.
- As a pathological application of the predictive framework, we predict that tendinosis-mediated tissue degeneration alters both histone methylation rates and chromatin-lamina affinity, mimicking the effect of soft substrates.

A notable outcome of our predictive framework is the observation that chromatin-lamina interaction strength exhibits a bimodal distribution, characterized by two distinct peaks (Fig. 8B). This spatial heterogeneity likely stems from the variability in the composition of the lamina meshwork and the presence of diverse nuclear membrane proteins such as ion channels, nuclear pores, and chromatin-tethers like LBR and LAP2β[68]. While proteins like LBR, LAP2β, emerin, and HDAC3 are actively involved in tethering heterochromatin, other nuclear membrane components like ion channels, pumps and nuclear pore complexes only interact very weakly with chromatin. Our model effectively captures this inherent variability, predicting the bimodal distribution of chromatin-lamina affinity in each nuclei analyzed (Fig. 2F, Supplementary Fig. 17). The first peak, representing strong chromatin-lamina interactions, typically correlates with dense H2B localizations in the STORM images, manifesting as large elongated peripheral heterochromatin domains. Conversely, regions along the

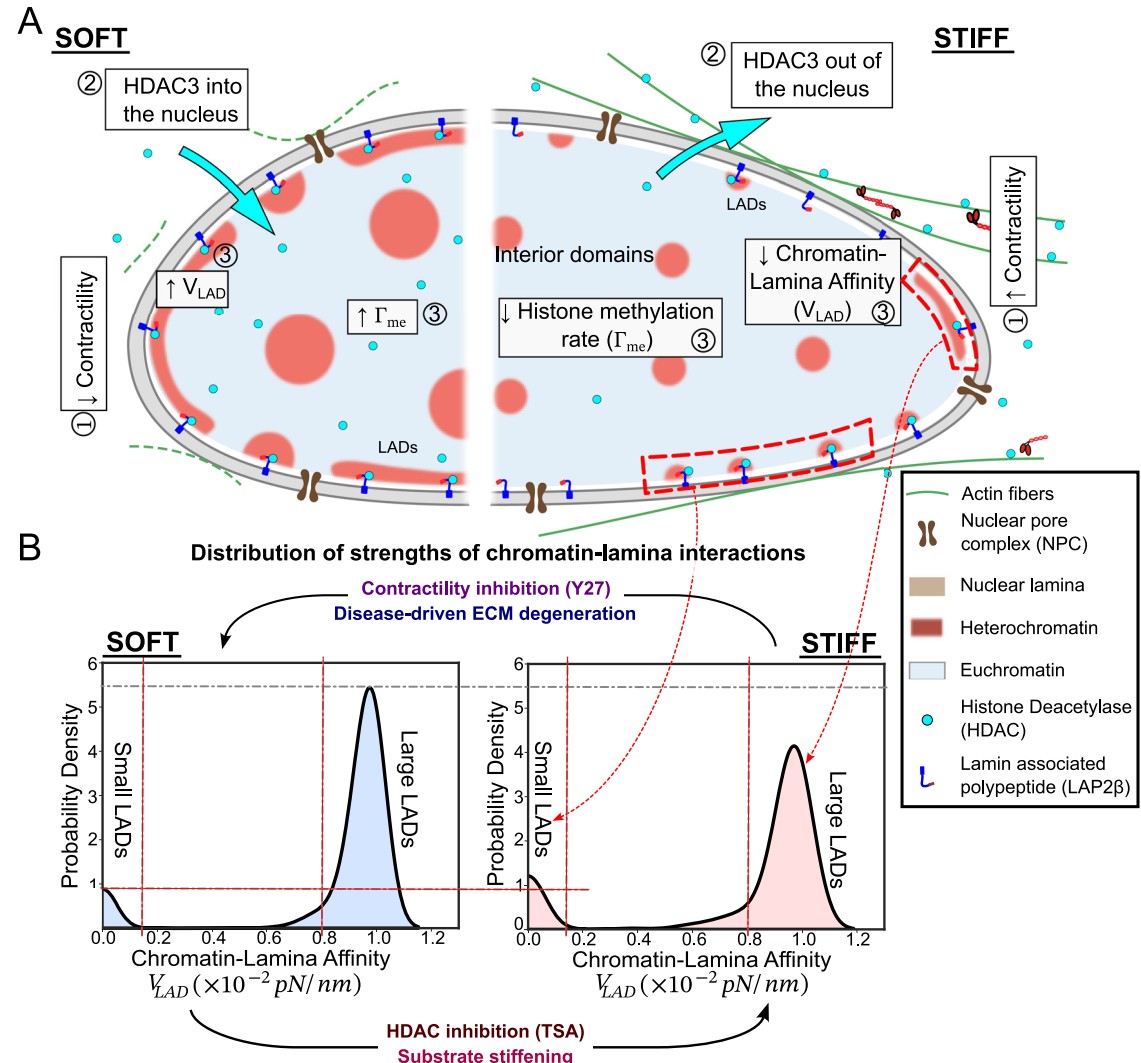

**Fig. 8 | A summary of the proposed model for synergistic regulation of LAD organization by epigenetic reactions and the strength of chromatin-lamina interactions. A** Schematic depicting the shuttling of HDAC3 between the cytoplasm and the nucleus depending on the cellular contractility. The right panel shows the organization of chromatin on stiff substrates, where higher contractility leads to reduced intranuclear HDAC3 levels and hence also lower chromatin-lamina affinity. The left panel shows chromatin organization in cells on soft substrates with higher methylation levels and lamina affinity. **B** The nucleus-wide distribution of the strength of chromatin-lamina affinity extracted from our framework shows a bimodal distribution with one peak at vanishing chromatin-lamina interactions with small discrete LADs (labelled small LADs) and another at strong chromatin-lamina interactions, comparable to the chromatin-chromatin interactions, with large continuous LADs (labelled large LADs). On a stiff substrate reduced nuclear HDAC3 contributes to decreasing the chromatin-lamina interaction strength. Note that the peak corresponding to large LADs decreases. **B** Also summarizes the effects of pharmacological perturbations (TSA, Y27 treatments), change in substrate stiffness and in-vivo tendinopathic ECM degeneration on the distribution of the chromatin-lamina interaction strengths.

periphery corresponding to the second peak, characterized by minimal chromatin-lamina interactions, exhibit little to no presence of peripheral heterochromatin domains. Between these extremes, a spectrum of intermediate chromatin-lamina interactions is observed, with small and thin peripheral heterochromatin domains indicative of regions with relatively weak but still perceptible affinity for heterochromatin. Our model predictions closely align with the heterogeneity observed in Lamin B1 enrichment in genomic regions across multiple cell types, as unveiled by LB1 ChIP-seq[36]. The peak of high chromatin-lamina affinity predicted by our model likely corresponds to genomic regions with high Lamin B1 occupancy (classified as Type 1 LADs). Conversely, the predicted range of weaker but nonzero chromatin-lamina affinities may resemble genomic regions exhibiting relatively lower Lamin B1 occupancy (Type 2 LADs) in LB1 ChIP-seq[36]. Thus, by capturing this distribution of chromatin-lamina affinities, our multi-modal framework offers insights into the distinct mechanisms driving peripheral chromatin sequestration.

The role of distinct chromatin-sequestering mechanisms can be effectively probed by carrying out pharmacological perturbations and using our predictive framework to enquire the changes in the strength of chromatin-lamina interactions. For instance, we have demonstrated that treatment of nuclei with HDAC inhibitor TSA results in an overall reduction in the chromatin-lamina affinity, while treatment with GSK, inhibiting EZH2, causes no significant change. Consistent with our predictions, it has been previously reported that HDAC inhibition through TSA or RNA-induced inhibition detaches gene loci from the lamina and relocates them to the nuclear[44–46], implicating HDAC3 in tethering segments of chromatin to the nuclear periphery, where it interacts with LAP2β and Emerin to suppress gene expression[40]. Previously, it has been shown that via anchoring proteins such as emerin, Lamin A/C is implicated in heterochromatin tethering to the lamina[69,70] and that stiff tissue environment drives an increase in Lamin A/C[71]. However, we observe a lower chromatin lamina affinity on stiffer substrates. Thus, our model suggests HDAC3 whose nuclear levels are

reduced by cytoskeletal tension[49,50], plays a relatively more prominent role in the formation of LADs as compared to Lamin A/C, at least in the cell-culture conditions described in Methods Section. In agreement with this conclusion, previous work showed that knockdown of Lamin proteins does not lead to changes in LADs in mouse embryonic stem cells[72]. While our data and previous work point to a mechanistic role for HDAC3, other HDAC isoforms may also contribute to chromatin-lamina tethering. Our model is agnostic to HDAC subtype and functionally interprets nuclear HDAC levels through their influence on histone deacetylation and methylation kinetics. Thus, the observed changes in chromatin-lamina affinity may reflect the combined activity of multiple HDAC family members. In the future, more specific inhibition of individual HDACs as well as targeting enzymes responsible for alternative repressive histone methylation marks such as H3K9me2/3 coupled with our predictive modeling will help elucidate the individual contributions of these epigenetic mechanisms to the formation of LADs.

Furthermore, our predictive framework links the cytoskeletal contractility to nuclear chromatin organization revealing an increase both in epigenetic reaction rates and chromatin-lamina affinity on soft substrates, where nuclear HDAC3 levels are higher, compared to stiff. These findings align with our prior reports that on stiff substrates, increased cytoskeletal contractility results in the shuttling of HDAC3 from the nucleus into the cytoplasm in mouse fibroblasts[50,73]. This nuclear export of HDAC3 correlates with reduced chromatin-lamina interactions and altered chromatin organization. To validate that the loss of contractility on soft substrates indeed contributes to the altered chromatin organization, we also pharmacologically disrupted cell contractility using Y27 treatment, after which we observed that both the histone methylation rate and chromatin-lamina affinity increase—similar to nuclei on soft substrates. However, after Y27 treatment the increase in chromatin-lamina affinity is more pronounced than on a soft substrate resulting in excessive sequestering of chromatin to the lamina reducing the size of the interior domains (as discussed in Supplementary Information Section 10). This may be mediated by altered chromatin-binding after Y27 treatment due to mechanisms in addition to HDAC3[6,74]. While this requires further investigation, it can be easily incorporated into the model by suitably adapting the mechanosensitive chromatin-lamina interactions as a function of such binding mechanisms. Thus, our observations strongly suggest that via the shuttling of epigenetic factors like HDAC3, cytoskeletal contractility chemo-mechanically transduces environmental stiffness stimuli to the nucleus and drives chromatin reorganization.

Our predictive framework integrates image quantification with parameter-extraction algorithms and is broadly applicable beyond STORM to other super-resolution techniques capable of visualizing heterochromatin domains. It enables unbiased inference of nucleus-wide biophysical parameters without requiring prior knowledge of nuclear stimuli. This makes it particularly valuable in contexts where chromatin remodeling mechanisms remain elusive, allowing the model to illuminate how epigenetic reactions and chromatin-lamina interactions contribute to the formation of heterochromatin domains both in the nuclear interior and at the periphery. To maintain computational tractability and interpretability, we employ a minimal set of spatially varying molecular parameters—namely, epigenetic reaction rates and chromatin-lamina affinity. Despite its simplicity, the model captures the stochastic and heterogeneous organization of chromatin observed at the single-cell level, with predictive accuracy validated by strong agreement between simulations and experimental observations.

The current formulation, however, is based on simplifying assumptions. Specifically, it attributes spatial heterogeneity in chromatin organization primarily to variability in molecular parameters, while omitting other plausible contributors such as genomic domain length differences, loop extrusion dynamics, and three-dimensional domain-domain interactions. Incorporating these features in future models could enhance both descriptive power and predictive accuracy. Additionally, future refinements could replace the average chromatin–lamina affinity with protein-specific interaction coefficients (e.g., LAP2β, LBR, emerin, PRR14), leveraging multiplexed imaging and knockout experiments to delineate the roles of individual nuclear envelope proteins in chromatin organization. While our current analysis focuses on steady-state chromatin architecture, the underlying reaction-diffusion model can be extended to simulate time-dependent dynamics. This opens avenues to study transient chromatin remodeling events in response to temporally varying mechanical or biochemical cues, particularly when paired with dynamic imaging experiments.

Looking ahead, coupling this modeling framework with polymer simulations could enable high-resolution depictions of genome organization at the level of individual chromatin compartments and gene loci. When integrated with imaging techniques such as FISH, Oligopaint, and other forms of super-resolution microscopy[75,76], the framework can help identify genes regulated via epigenetic mechanisms or chromatin-lamina interactions. These capabilities hold diagnostic and therapeutic promise, offering routes to modulate gene expression in response to environmental changes by targeting chromatin structure near the periphery or within the nuclear interior. Overall, our multi-modal parameter extraction approach lays a foundational platform for investigating how environmental stimuli influence chemo-mechanical nucleo-cytoskeletal signaling and drive mesoscale chromatin reorganization throughout the nucleus.

## Methods

### Description of chromatin organization in terms of the hetero and euchromatic phases, and the nucleoplasm

At the mesoscale, chromatin is organized as transcriptionally active euchromatin alongside tightly compacted domains of heterochromatin where genes are predominantly silenced. The heterochromatin domains can occur both in the interior of the nucleus as well as along its periphery as lamina-associated domains (LADs)[7] (Fig. 1C). We mathematically model the interior and peripheral compartmentalized organization of chromatin as a far-from-equilibrium dynamic phenomenon governed by the coupled energetics of chromatin-chromatin and chromatin-lamina interactions. The spatiotemporal evolution of chromatin is governed by the kinetics of nucleoplasm diffusion and spatially varying active interconversion of acetylated and methylated histones mediated by epigenetic factors such as histone deacetylases (HDACs).

The nucleus comprises three constituents, namely, the two forms of chromatin – hetero (prominently methylated repressive form) and euchromatin (prominently acetylated) – and the nucleoplasm. Since our primary focus is on the key features of chromatin organization, all other nuclear constituents are included in the nucleoplasm. At any given location $\mathbf{x}$ and time $t$, we define the volume fraction of the three nuclear constituents as $\phi_h(\mathbf{x}, t)$, $\phi_e(\mathbf{x}, t)$, and $\phi_n(\mathbf{x}, t)$, such that they add up to unity, $\phi_h + \phi_e + \phi_n = 1$. Thus, the nuclear composition at any point can be fully described by two independent variables. The local nuclear composition at each point can be equivalently defined in terms of (i) the volume fraction of the nucleoplasm $\phi_n$ and (ii) the difference between the volume fractions of heterochromatin and euchromatin $\phi_d = \phi_h - \phi_e$. These variables enable a natural definition of the evolution of nucleoplasm and the epigenetic marks between the two forms of chromatin. Furthermore, $\phi_d$ plays the role of an order parameter, such that $\phi_d > 0$ corresponds to the heterochromatic phase, while when $\phi_d < 0$, the euchromatic phase.

### Driving forces for the spatiotemporal evolution of chromatin

The evolution of chromatin phases is determined by energetic driving forces along with the epigenetic reaction kinetics of histone

methylation and acetylation. To capture the energetic driving forces, we construct the local free energy density at any point in space and any instant of time, in terms of the independent variables $\phi_n(x,t)$ and $\phi_d(x,t)$ as,

$$f = \underbrace{\frac{c}{2}\left[\phi_e^2 + \phi_h^2(\phi_{h0}-\phi_h)^2\right]}_{\text{chromatin–chromatin interactions}} + \underbrace{\frac{\kappa}{2}\left[|\nabla\phi_n|^2 + |\nabla\phi_d|^2\right]}_{\text{Interfacial energy}} - \underbrace{\frac{V(\phi_h)}{d_0}e^{-\frac{d}{d_0}}}_{\text{chromatin–lamina interactions}}$$

(3)

Here the first term represents a Flory–Huggins type of description for the free-energy of chromatin organization and arises from the competition between the enthalpy of nucleosome interactions and the entropy of mixing of the two phases. The enthalpic interactions between proximal nucleosomes could include electrostatic and hydrogen-bonding contributions associated with their epigenetic state. For instance, histone posttranslational modifications can alter the positive charge on histones, thereby modifying chromatin compaction[77]. Furthermore, cross-linking of methylated histones mediated by HP1α[78,79] can increase chromatin-chromatin interactions and reduce overall enthalpy, leading to an energy minimum corresponding to compacted water-poor heterochromatin phase. On the other hand, the water-rich, loosely packed euchromatin phase increases the local entropy giving rise to second energy minimum. Moreover, differential levels of activity between the transcriptionally active euchromatin phase and the silenced heterochromatic phase have also been shown to alter the depth of the energy wells, thus contributing to energetic chromatin-chromatin interactions. Thus, the resulting double-well energy landscape comprises two minima (Fig. 1E) corresponding to the euchromatin $(\phi_h = 0)$ and heterochromatin $(\phi_h = \phi_{h0})$ phases. Physiologically the strength of chromatin-chromatin interactions is order of $\sim 2 - 4k_BT \sim 8 - 16pN.nm$[80] (details Supplementary Information Section 9). The parameter $c$, which characterizes these chromatin-chromatin interactions per unit volume, is on the order of $\sim 8\times10^{-3} - 16\times10^{-3}pN/nm^2$.

The second set of terms penalizes the formation of sharp gradients in the spatial organization of chromatin (refer Supplementary Information Section 1.2) which opposes the bulk energetic contributions from chromatin-chromatin interactions that try to segregate the dissimilar phases. The term $\kappa$ ($\sim 0.8pN$, estimated in Supplementary Information section 9) is the increase in the energy due to formation of a unit width of the interface. As $\kappa$ increases, there is a greater penalty on formation of sharp interfaces, resulting in more smooth interfaces. The width of the interface $l_{int}$ is determined by the balance of the interfacial penalty and the bulk energy contributions, such that $l_{int}$ ($\sim \sqrt{\kappa/c}$) $\sim 10nm$ (see Supplementary Information Section 9 for details). We have previously reported that the experimentally observed change in chromatin density between hetero and euchromatin is indeed smooth[17].

The last term captures the interaction between chromatin and the nuclear lamina, which leads to the formation of LADs along the nuclear periphery. In-vivo, the interactions between chromatin and the nuclear lamina are orchestrated by a diverse set of chromatin anchoring proteins, such as LAP2β, LBR, emerin and PRR14, associated with the inner nuclear membrane and the nuclear lamina[7,81]. While these interactions can be individually incorporated by capturing the strength of each anchoring protein associated with the lamina and the nuclear membrane, as a first approximation we assume these to be the same.

The function $V(\phi_h)$ measures the chromatin-lamina interaction energy per unit surface area of the nuclear envelope and considers all tethering mechanisms collectively contributing to the spatial organization of LADs. The magnitude of strength of chromatin-lamina interactions are dependent on the local chromatin state. In euchromatic phase $(\phi_h = 0)$ the interactions are mediated by proteins in the lamina that may tether euchromatin and have a strength $V_{EC} = V(0)$. On the other hand, the tethering of methylated histones, mediated by proteins such as LAP2β has an interaction strength $V_{HC} = V(\phi_{h0})$. We define the preferential interactions of the lamina with heterochromatin over euchromatin as $V_{LAD} = V_{HC} - V_{EC}$. As shown in the Supplementary Information (Supplementary Section 5.1), the specific choice of $V(\phi_h)$ only weakly impacts $V_{LAD}$. Furthermore, the strength of chromatin-lamina interactions decays exponentially away from the nuclear periphery over a length-scale $d_0$ comparable to the size of anchoring proteins ($\sim 2.5nm$)[82]. Physically, the term $V/d_0$ in Eq. (3) measures the chromatin-lamina interactions per unit volume of region where the LADs are formed.

## Spatiotemporal dynamics of chromatin evolution

To reduce the total free energy of the system, any random initial configuration (white dot) on the energy landscape (shown via a contour plot in Fig. 1E) will be driven toward the two energy wells (red and blue dots). The temporal dynamics of energy-driven evolution are determined by the magnitudes of the gradients of the energy landscape, defined as chemical potentials $\mu_{n(d)}(\mathbf{x},t) = \frac{\delta f}{\delta\phi_{n(d)}}$. Here, the functional derivative $\delta$ is a measure of the change in free energy density with respect to the volume fraction. The spatiotemporal evolution of nucleoplasm is diffusively driven by spatial heterogeneities in its chemical potential $\mu_n$ and can be written as,

$$\frac{\partial\phi_n}{\partial t} = \underbrace{M_n\nabla^2\mu_n}_{\text{diffusion}}$$

(4)

where $M_n$ denotes the mobility of nucleoplasm in the nucleus. Thus, the passive diffusion of the nucleoplasm over time, via Eq. (4), reduces the overall free energy of chromatin organization while conserving the net amount of nucleoplasm in the nucleus unless there is exchange of water with the cytosol (as shown in Supplementary Fig. 1) via appropriate boundary conditions, as discussed in the Supplementary Information Section 1.4.

The diffusion of nucleoplasm via Eq. (4) captures only the movement of water, without changing the relative locations of acetylated and methylated marks on chromatin, as shown in Supplementary Fig. 1. Epigenetic marks can be actively added to or removed from histone tails via epigenetic reactions mediated by epigenetic factors such as HDACs, allowing interconversion between the heterochromatin and euchromatin (Supplementary Fig. 1). The conversion of euchromatin into heterochromatin phase requires two steps – removal of acetyl group via deacetylation followed by addition of methyl group via methyltransferase (HMT) activity. Thus, the overall rate of histone methylation can be written as $\frac{1}{\Gamma_{me}} = \frac{1}{\Gamma_{HDAC}} + \frac{1}{\Gamma_{HMT}}$, as shown in Fig. 1D. Similarly, the conversion of heterochromatin into euchromatin phase, which involves the activities of histone demethylase (HDM) and histone acetyltransferase (HAT) is $\frac{1}{\Gamma_{ac}} = \frac{1}{\Gamma_{HDM}} + \frac{1}{\Gamma_{HAT}}$. As we discuss next, the second term in Eq. (5) represents these nonconservative reaction kinetics.

Additionally, the reaction kinetics can also be influenced by chromatin-chromatin interactions between spatially proximal nucleosomes. For instance, conversion of heterochromatin into euchromatin might be energetically less favorable in a heterochromatin-rich neighborhood than in a euchromatin-rich neighborhood (further discussion in the Supplementary Information Section 1.4). The reaction-kinetic contributions arising from these neighborhood interactions can be shown to aptly mimic (see Supplementary Information Section 1.4 for the theoretical derivation) a diffusion-like evolution of epigenetic marks determined by neighborhood-dependent reaction kinetics. The diffusion-like and reactive kinetics together determine

the spatiotemporal evolution of epigenetic marks as,

$$\frac{\partial \phi_d}{\partial t} = \underbrace{M_d \nabla^2 \mu_d}_{\text{Diffusion of epigenetic marks}} + \underbrace{2(\Gamma_{me}\phi_e - \Gamma_{ac}\phi_h)}_{\text{epigenetic reaction kinetics}} \quad (5)$$

where $M_d$ is the effective mobility of epigenetic marks in the nucleus (see Supplementary Information Section 1.3). Note that $\Gamma_{me}$ and $\Gamma_{ac}$ are not constant but spatially dependent on the non-homogeneous distribution of epigenetic regulators[29,83] such that $\Gamma_{me} = \Gamma_{me}(\mathbf{x},t)$ and $\Gamma_{ac} = \Gamma_{ac}(\mathbf{x},t)$.

Equations (3)–(5) can be numerically solved to determine the spatiotemporal organization of chromatin in the nucleus. For the numerical solution, we rescale the equations using the intrinsic length and time scales, as described in the Supplementary Information (Supplementary Section 1.5). A description of the model parameters, as well as the initial and boundary conditions used for the numerical solution is provided in the Supplementary Information Section 8.

### Super-resolution STORM imaging and localization analysis of chromatin structures

The 2D-STORM images were obtained from previously published work[28,29] and re-analyzed using the theoretical framework developed here.

**Cell culture and immunostaining.** Human mesenchymal stromal cells (hMSCs) were isolated from fresh bone marrow obtained from human donors[84]. The cells were then plated and expanded on tissue-culture plastic in α-modified essential medium (α-MEM) supplemented with 10% fetal bovine serum (FBS), 1% penicillin–streptomycin, and 5 ng/ml basic fibroblast growth factor at 37 °C and 5% $CO_2$ until the colonies reached 80% confluency. Subsequently, the cells were stored in liquid nitrogen using a freezing medium composed of 95% FBS and 5% dimethylsulfoxide (DMSO). Throughout the expansion process, all hMSCs were cultured in standard growth medium consisting of α-MEM supplemented with 10% FBS and 1% penicillin–streptomycin. Human Tenocytes (hTCs) were isolated from the finger flexor tendon tissues of both young individuals and patients diagnosed with tendinosis[85]. The cells were then plated in basal growth medium, which consisted of high-glucose DMEM medium supplemented with 10% penicillin–streptomycin, L-glutamine, and 10% FBS.

hMSCs and hTCs were fixed using methanol-ethanol (1:1) at −20 °C for 6 min, followed by blocking with a solution of 10%(w/v) bovine serum albumin (BSA) in phosphate-buffered saline (PBS) for 1 h[3]. Subsequently, the cells were subjected to overnight incubation at 4 °C with a 1:50 dilution of histone H2B anti-rabbit antibody (ProteinTech, #15857-1-AP). After thorough PBS washing, the samples were incubated with secondary antibodies labelled with Alexa Fluor 405 - Alexa Fluor 647[3].

Human fibroblasts (hFB) were grown in growth media (DMEM medium, 10% FBS, 1× non-essential amino acids, 1× Penicillin/Streptomycin and 1× GlutaMax) supplemented with 5 μM ethynil-deoxycytidine (EdC) for 96 h[28]. The cells were fixed with 4% paraformaldehyde (PFA) diluted in PBS for 15 min at room temperature. Subsequently, they were permeabilized with 0.3% (v/v) Triton X-100 in PBS for 15 min at room temperature. Afterward, the cells were blocked using a solution containing 10% BSA (w/v) and 0.01% (v/v) Triton X-100 in PBS.

**Confocal Airyscan imaging.** Multi-channel confocal Airyscan images of hMSCs were acquired using the Zeiss LSM 900 with Airyscan 2 commercial super-resolution imaging platform (×63 oil immersion objective). The HDAC3 (Cell Signaling Technology, #3949, dilution 1:50) reporter dye Alexa Fluor 647 (Abcam, Cat: ab150115, dilution 1:50) was excited using 640 nm laser set at 5% power. The selective staining

of DNA and F-actin were done by DAPI (Thermo Fisher Scientific, Cat: D1306) and conjugated Alexa Fluor 568 (Thermo Fisher Scientific, Cat: A12380). The DAPI and Phalloidin reporter dye were excited with 405 nm, and 560 nm laser at 2% power, respectively.

**Super-resolution imaging.** Super-resolution imaging-based visualization of immunostained H2B was utilized to observe the detailed mesoscale organization of chromatin in the cell nuclei. Super-resolution images were acquired using the ONI (Oxford NanoImager) commercial STORM imaging platform. The imaging system was equipped with a ×100, 1.4 numerical aperture oil-immersion objective and a sCMOS Hamamatsu Orca Flash camera. To ensure optimal photo switching of Alexa Fluor 647, the imaging buffer followed standard guidelines[86], consisting of 10 mM cysteamine MEA in GLOX Solution: 0.5 mg ml⁻¹ l-glucose oxidase, 40 mg ml⁻¹ l-catalase and 10% glucose in PBS. The reporter dye (Alexa Fluor 647) was excited using a 640 nm laser set at 40% power. Gradually, the power of the 405 nm laser was increased to reactivate Alexa Fluor 647. The camera's exposure settings were configured to 15 ms, and a total of 30,000 frames were collected for each image using the ONI software. Additional details can be found in ref. 29.

Super-resolution imaging of EdC labeled DNA in hFb cells were acquired using a commercial N-STORM microscope (Nikon) equipped with a CFI HP Apochromat TIRF 100 × 1.49 oil objective, an iXon Ultra 897 camera (Andor), and a Dual View system (Photometrics DV2 housing with a T647lpxr dichroic beam splitter from Chroma). A 647 nm laser was employed to excite the DNA labeled with AlexaFluor 647, with a power density of ~3 kW/cm. Alexa 647 was progressively reactivated with increasing 405 nm laser power during acquisition, up to a maximal power density of 0.020 kW/cm². The imaging buffer consisted of 100 mM Cysteamine MEA, 5% glucose, 1% Glox, and 0.75 nM Imager strand (I2-560 Ultivue) in Ultivue Imaging Buffer. Additional details can be found in ref. 28.

The subsequent post-analysis of STORM image localization was performed using custom-written MATLAB codes[29].

**Identification and quantification of heterochromatin domains.** MATLAB was utilized for the analysis of STORM images using an adapted Voronoi tessellation-based segmentation method[87] to construct Voronoi polygons of each fluorophore. This approach assigns a Voronoi polygon to each localization, where the polygon size is inversely proportional to the local localization density[28]. For each nucleus, the spatial distribution of localizations is represented by a collection of Voronoi polygons, where smaller polygon areas indicate higher density regions. The nuclear area is computed by summing the Voronoi polygon areas, excluding edge polygons due to their disproportionately large size resulting from edge effects. Localization density (total localizations per nuclear area) varies across nuclei due to differing mechanical and chemical treatments. To compare chromatin organization changes from different mechanical/chemical treatments, the localization density of each nucleus is normalized to its mean Voronoi polygon area, yielding a standard density unit. The inverse of the polygon area forms the reduced Voronoi density. Reduced Voronoi densities across nuclei and treatments were pooled and visualized as cumulative distribution plots. A threshold was set so that the 0–30 percentile of the density distribution represents sparse chromatin, while the 31–70 percentile represents dense chromatin. The Density-Based Spatial Clustering of Applications with Noise (DBSCAN) algorithm was then used to cluster neighboring and connected Voronoi polygons into distinct domains, ensuring at least three localizations per cluster.

**Identification and quantification of lamina-associated domains.** We further distinguished LADs from interior heterochromatin domains based on their proximity to the nuclear lamina. For each heterochromatin cluster, we quantified the distance of each localization to nuclear envelope. If any localization lies within the predefined

threshold distance, we classified that cluster as a LAD. The threshold was set at 2.5% of nuclear radii from nuclear envelope based on the approximate size and localization of nuclear lamina proteins and their interaction zones (details in Supplementary Information Section 11).

To quantify the thickness of LADs, we divided the nuclear boundary into n segments. Within each segment, we computed the cumulative area covered by LADs and divided it by the segment length to determine the thickness of the LAD domain. In our analysis, we selected $n = 50$ as the number of segments. However, we also conducted tests up to $n = 100$ and observed no significant changes in the results.

### Reporting summary

Further information on research design is available in the Nature Portfolio Reporting Summary linked to this article.

## Data availability

The data generated in this study are provided in the Source Data file. Additionally, the processed STORM imaging data used in this study have been deposited on Zenodo and are publicly available at https://doi.org/10.5281/zenodo.16763306. Source data are provided with this paper.

## Code availability

The code used for measurement of sizes of heterochromatin domain obtained from STORM imaging is freely available through github (https://github.com/ShenoyLab/STORM_Analysis_Parameter_Extraction)[88].

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

## Acknowledgements

This work was supported by NIH Award U54CA261694 (V.B.S.); NCI Awards R01CA232256 (V.B.S.); NSF CEMB Grant CMMI-154857 (V.B.S.); NSF Grants MRSEC/DMR-1720530(V.B.S) and DMS-2347834 (V.B.S.); NIBIB Awards R01EB017753 (V.B.S) and R01EB030876 (V.B.S.); National Institute of General Medical Sciences award R01GM155943 (V.B.S). We gratefully acknowledge the valuable comments and suggestions from Dr. Rajan Jain.

## Author contributions

M.D., Z.G., A.K. and V.B.S. conceived the project and conducted the theoretical and numerical analyses. S.C.H., R.L.M. and M.L. performed the STORM imaging. M.D., Z.G. and A.K. carried out the quantitative analysis of STORM images. Z.G., R.B. and R.J. conducted the immunofluorescence imaging. V.V. led the sequencing analyses. M.D., A.K. and V.B.S. led the drafting of the manuscript, with Z.G., V.V., S.C.H. and M.L. contributing to revisions and edits.

## Competing interests

The authors declare no competing interests.
