## [Peer Review file · Nature Communications]

Revealing the Biophysics of Lamina-Associated Domain Formation by Integrating Theoretical Modeling and High-Resolution Imaging

Corresponding Author: Professor Vivek Shenoy

Version 0:

Reviewer comments:

Reviewer #1

(Remarks to the Author)

Dhankhar et al develops a theoretical framework to investigate how heterochromatin density within the nucleus and at its periphery may depend on molecular processes and interactions, and use it to analyze super-resolution images in various conditions to extract (and discuss) the corresponding distribution of molecular parameters.

In particular, authors have extended their previous Cahn-Hilliard-like framework in order to integrate interaction with the nuclear lamina. An insightful analysis of the model allows them to relate analytically some structural properties (like the radius of heterochromatin domains or the LAD thickness) of heterochromatin with model parameters that capture underlying molecular processes. They then use their analytical formulas to extract parameter values from experimentally-measured structural properties. Finally, this allows them to discuss the effect of epigenetic drugs and of cell microenvironment on these parameters.

The work is serious and well done and the article is overall well written. In particular, the original idea of exploiting STORM images through a mechanistic model to infer some molecular information is great. Furthermore, their findings seem adequate with the general understanding of heterochromatin and lamina-associated domains formation. However, I have some major & minor concerns to be discussed/addressed before acceptance.

Major

- 1) Authors based their analysis on the assumption that the heterogeneities of HC domain sizes and LAD thicknesses observed experimentally in a cell are due to spatial heterogeneities in the model – molecular – parameters. This is a strong hypothesis that should be clearly motivated and discussed. In particular, there are many intrinsic sources of noise that may naturally lead to heterogeneous HC domain sizes or LAD thicknesses without the necessity to invoke heterogeneity in parameters. For example, along the genome, HC domains have various genomic lengths, thus their 3D folding could naturally lead to different sizes, moreover these domains may interact in the 3D space (via phase separation for example) and thus forming larger – also heterogeneous – foci.
- 2) On Figure 4 and Figure 6, results are shown for hMSC cells as control. However, the distributions of LAD thicknesses (panel C top right) seem rather different. While I can understand that experimentally there might be some discrepancies, this questions the validity of the approach to extract robustly interpretable parameters from experiments.
- 3) It is unclear how the parameters Γ_{me} and V_{LAD} are actually inferred from Eqs 5, 6 & 8 as there are other – a priori unfixed – parameters (D_h , Γ_{ac} , θ , etc.) in these equations.
- 4) Authors claim that ‘the predicted spectrum of chromatin-lamina affinities align with in multiple cell lines’. A figure showing such a result from ChIP should be shown.
- 5) The introduction of the manuscript consists in a small introductory paragraph with generalities followed by a kind of extended summary of the work. This is clearly not enough to really introduce the question, the state-of-the-art and the current challenges to motivate their work. In particular, the current biological and physical knowledge on HC and LAD formation should be discussed. For example, several papers (purely theoretical and in dialog with experiments) already addressed the interplay between both (Falk et al, Nature 2019; Ulianov et al, Nat Com 2019; Sati et al, Mol Cell 2020; Chiang et al, Cell Rep 2020; Tolokh et al, Epigen. & Chrom. 2023; etc.). A fair presentation also on the use of Cahn-Hilliard-like framework to study chromatin organization including authors’ previous work but also from others (Awazu et al, PRE 2015 ; Potoyan’s

group work; etc.) would be welcomed to highlight the originality of the presented model.

6) Similarly, the Discussion/Conclusion section is more an extended summary of their results than an actual discussion compared to the state-of-the-art and of the limitations/caveats of their approach and interpretations.

7) TSA is affecting all HDACs and is likely to be a huge perturbation of the system. Authors relates the observed changes in parameters to the role of HDAC3. Is there a way to specifically target HDAC3 to experimentally validate this hypothesis?

Minor

- In Nat Coms, normally, Materials and Methods are localized after the Conclusion/Discussion section.

- Page 4, the citation of Ref[14] may seem inappropriate.

- Page 5, Eq.3 is cited before it is written.

- It is unclear if super-resolution images are 2D or 3D.

- To infer LAD from STORM images, authors use a threshold (2.5% of nuclear radius). What is the rationale for this choice?

Are the results (and following parameter inference) robust?

- Just after Eq.4: area -> volume, no?

- The paragraph 'As a second step, we validate ...' seems to claim some statements but do not refer to any figures.

- Figs 4- 8, while the differences between both conditions are clearly visible on the STORM images, this is quite subtle on the simulations. Can the authors comment? Maybe showing a zoom of the simulation as in Fig2C would help.

- Figs4-8: authors made simulations using the average parameter values. What would happen if instead they input the actual inferred spatial distribution of parameters?

- Fig.S1: the terms cis and trans diffusion should be more clearly defined.

- Eqs S5 and S6, the meaning of 'epigen' and 'cons' is unclear.

- Fig.S11: it seems that there is a misleading caption.

- Fig.S13: how to explain the very high heterogeneities in HC repartition among the different cells (entire regions seems depleted in HC)? Is it the nucleolus?

- The paper is written towards a physics/biophysics audience. It might be very difficult for a biologists interested in LAD formation to grasp the methodology. It is not a problem per se but the authors may want to consider this point if they also aim their results at biologists.

(Remarks on code availability)

Reviewer #2

(Remarks to the Author)

Dhankhar et al.

In this manuscript, Dhankhar et al examine biophysical properties ("size and shape") of LADs by integrating analysis of microscopy data and results from a theoretical modeling framework they have implemented. Experimental manipulations such as inhibition of EZH2, TSA, various cell culture surface hardnesses and use of tendinosis patient-derived cells provide some functionality to their analyses and modeling framework. Using this combined approach, the authors make inferences from their modeling on critical parameters driving LAD morphology, and overall "underscore the pivotal role of the microenvironment in shaping genome organization".

Overall, I find the manuscript too complicated, confusing, with a constant and forth between analysis of imaging data and "theoretical modeling" making it unclear which contributes to which and where conclusions are drawn from. It is difficult to follow a thread and the main messages are unclear (which are they?). The novelty of their findings, if any, is not apparent and possibly erroneous because the rationale and parameters they focus on throughout the paper are too simplistic. The conclusions drawn from modeling are consequently withdrawn from many other biological observations in the literature. It is to me completely unclear (and possibly erroneous) how the various parameters tested affect "methylation rates" and what this has to do with LAD properties (see also below on methylation). Reading (and understanding) the manuscript with either a biology mindset is to say the least extremely challenging. This is also the case reading and evaluating it from a physics/mathematics/modeling perspective: while the mathematics behind it are a priori correct, the design, set up and presentation throughout the paper is simply too overwhelming. The nomenclature / abbreviations chose is not intuitive, hard to recall from one experiment to the next, and overall hard to follow.

In short, the paper is contains too much data hiding the main purpose, conclusion and novelty (if any) of the findings, too long and too confusing. The interplay between analysis of microscopy images and use of the "theoretical framework" in drawing conclusions is unclear to me.

I don't understand the rationale for manipulation EZH2 activity (and conclusions on "methylation reactions") since it is already known that H3K27me3, catalyzed by EZH2, plays essentially no role in the establishment or maintenance of LADs. In fact, the authors find the GSK inhibitor they are using has no effect on LADs – this could be predicted from previous studies, so I don't quite see the point of these experiments. I don't see how the authors can make implications of manipulating HDAC activity with TSA on " the overall rate of histone methylation " in the models. In fact, I simply do not see how the parameters tested possibly affect "methylation reactions". This seems to be result from their modeling framework, but is not demonstrated by the biology and therefore of limited, if at any, value.

If the authors do want to consider histone methylation in their modeling, then they should instead consider H3K9me2 or H3K9me3 methylation in (perhaps) driving (or contributing to) LAD properties in their models, as these modifications are largely found in constitutive bona fide LADs (in contrast to H3K27me3), in particular in the cell types they are considering.

Microscopy analyses: I am surprised by the very low number of nuclei analyzed (n=4-10 depending on the experiment and figure); how can the authors get statistics from so few nuclei? A lot more nuclei going into analyses would be warranted. The actual differences claimed in the plots are at times very minute and it is doubtful that they are biologically relevant. Voronoi plots resulting from STORM images shown are at times inconsistent between experiments or figures, showing significant variations in chromatin distribution in controls (e.g, Fig 4B control and Fig 3C control). Along the same lines, the GSK image in Fig 5B is very similar to the control in Fig 4B in terms of density. This should be clarified. Fig 4C is labeled "Quantification of STORM images", but the two lower plots are modeling results on methylation rate. This should be clarified.

That HDAC3 also interacts with lamin A/C is ignored in the paper. In fact, lamin A/C, which together with LBR is key protein involved in tethering heterochromatin at the nuclear lamina, is completely ignored in the manuscript.

From the experiments done (HDAC activity manipulation with TSA, surface hardness, healthy vs tendinosis nuclei), implications made by the authors on the importance of "methylation reactions" in driving LAD morphology are completely unclear to me and possibly erroneous.

Fig 7 and related text: the authors imply difference in "methylation rates" from increase surface stiffnesses which are very hard to believe. I don't see from Fig 7D how they modeling show differences in HC densities. This is also the case for tendinosis nuclei vs controls later in the paper.

(Remarks on code availability)
see above.

Version 1:

Reviewer comments:

Reviewer #1

(Remarks to the Author)

The reviewer acknowledges the authors for improving their manuscript by addressing most of my concerns.

There are still points that was eluded in the first round of revision which should be discussed in the main text:

- On comment 1 of the previous review: authors don't fully address the reviewer's comment on the source of stochasticity.

The point made by the reviewer should be discussed in the Discussion as a limitation of the current model.

- Comment 6: while the authors have improved a lot the Discussion part, it lacks a part where they discuss the limitations of their approach and the (sometimes strong) assumptions made along the paper.

- Comment 7: at several places in the Results section, authors claim that HDAC3 is driving the change lamina-chromatin interactions. These claims or statements should be relocated in the Discussion as authors just show correlation (levels of HDAC 3 change & lamina-chromatin interactions change, but the causal relationship is not demonstrated, it is therefore fair to add it to the discussion but not in Results).

- Authors have moved the Material & Methods section at the end of the paper (which is the standard format in Nature Com). A small paragraph at the beginning of Part 2.2 explaining briefly the model would be useful.

- There are still misleading or unclear statements:

o "in the interior of the nucleus shows a characteristic mean (FigS11), in excellent agreement..." the first part of the sentence refers to interior HC while Fig.S11 deals with Tlad

o "... is the far-field heterochromatin concentration...": please define far-field

o "...using the theoretical framework are accurate despite": what accurate means here ?

o "...spatially sampled from a normal distribution...": it is unclear if authors use the average parameter values (inferred from the STORM data) or draw them from a normal distribution.

o "The predicted spectrum of....aligns qualitatively....": experimentally, chromatin-LaminB1 interaction strengths are not observed using ChIP. Here the authors make the assumption that strong predicted interactions correspond to Type 1 LAD and intermediate predictions to Type 2 LAD, this is an assumption (more or less justified) but it is not a demonstration or a qualitative alignment.

o A recent Hi-C work on TSA was released (Paldi et al., Cavalli, bioRxiv 2024). Could be interesting to discuss it.

o "our numerical simulations can also capture the time-dependent dynamics...": not dynamical results are shown in the paper, so unclear if this statement is justified.

(Remarks on code availability)

Reviewer #2

(Remarks to the Author)

The authors have made a very significant effort to address my comments by providing detailed and documented explanations and by revising the manuscript where relevant. I have no further comments.

(Remarks on code availability)
N/A

Version 2:

Reviewer comments:

Reviewer #1

(Remarks to the Author)

The authors have addressed my concerns.
Revisions lead to a very nice paper.

(Remarks on code availability)

We would like to thank the referees for their careful review of our manuscript and for finding our work interesting. We found the comments/suggestions of the referees very useful. We have carefully considered each comment and have provided a detailed point-by-point response to them. We feel that the additional information we have now included in the revised manuscript and the supplementary information addresses all the concerns raised by the referees. We firmly believe that the revised manuscript is stronger because of the reviewers' suggestions

(The reviewer comments are in blue, our response in plain text, and the edits made to the manuscript are highlighted in yellow)

Reviewer #1

Dhankhar et al develops a theoretical framework to investigate how heterochromatin density within the nucleus and at its periphery may depend on molecular processes and interactions, and use it to analyze super-resolution images in various conditions to extract (and discuss) the corresponding distribution of molecular parameters.

In particular, authors have extended their previous Cahn-Hilliard-like framework in order to integrate interaction with the nuclear lamina. An insightful analysis of the model allows them to relate analytically some structural properties (like the radius of heterochromatin domains or the LAD thickness) of heterochromatin with model parameters that capture underlying molecular processes. They then use their analytical formulas to extract parameter values from experimentally-measured structural properties. Finally, this allows them to discuss the effect of epigenetic drugs and of cell microenvironment on these parameters.

The work is serious and well done and the article is overall well written. In particular, the original idea of exploiting STORM images through a mechanistic model to infer some molecular information is great. Furthermore, their findings seem adequate with the general understanding of heterochromatin and lamina-associated domains formation. However, I have some major & minor concerns to be discussed/addressed before acceptance.

General Response: We sincerely thank the reviewer for their thoughtful and constructive feedback on our manuscript. We are pleased that they find our work to be serious and well-executed, with a well-written presentation. We greatly appreciate their recognition of the originality of our approach and the relevance of our findings to the current literature.

We value the reviewer's comments and have carefully considered their major and minor concerns. In particular, we are grateful for their suggestions on streamlining the introduction and discussion sections to better contextualize our contributions. We also appreciate their recommendation to compare our imaging approach with Lamin ChIP-seq and DamID, which we have now incorporated into the revised manuscript. Our findings indicate that these methods provide complementary information, and their combined use offers a more comprehensive understanding of chromatin-lamina interactions.

In the revised manuscript, we have made significant updates to address all these points, as detailed below.

Major

Comment 1: Authors based their analysis on the assumption that the heterogeneities of HC domain sizes and LAD thicknesses observed experimentally in a cell are due to spatial heterogeneities in the model – molecular – parameters. This is a strong hypothesis that should be clearly motivated and discussed. In particular, there are many intrinsic sources of noise that may naturally lead to heterogeneous HC domain sizes or LAD thicknesses without the necessity to invoke heterogeneity in parameters. For example, along the genome, HC domains have various genomic lengths, thus their 3D folding could naturally lead to different

sizes, moreover these domains may interact in the 3D space (via phase separation for example) and thus forming larger – also heterogeneous – foci.

Our response: We agree with the reviewer that there can be multiple sources of heterogeneities that can lead to the experimentally observed spatial variations in the chromatin organization. In our analysis, the primary source of heterogeneities that we account for is the local availability of epigenetic factors that mediate the epigenetic post-translational modifications of histones. For instance, due to the presence of nuclear structures such as nucleolus, or biomolecular condensates such as the transcriptional factories or the Cajal bodies, local availability of epigenetic factors may be restricted. This is evidenced by our experimental measurements of the distribution of EZH2 [1], reproduced here as Figure R1, right] and HDAC3 [Figure 7A, reproduced here as Figure R1, left], which demonstrate spatial variations in their local concentrations. Thus, within our framework, the epigenetic reaction parameters, which denote the availability of the epigenetic factors, are presumed to be heterogeneously distributed spatially.

Figure R1: Representative immunofluorescence images showing heterogeneous distribution of HDAC3 (left) EZH2 (Right) in hMSC nuclei cultured on glass.

However, we agree that this is only one possible source of variability in the distribution of epigenetic marks, rather than the sole explanation.

As the reviewer rightly points out, variations in genomic lengths and different extents of 3D folding due to different strengths of interactions between neighboring chromatin segments could independently lead to spatially heterogeneous chromatin organization. These interactions can be incorporated into our model by introducing heterogeneities in other model parameters. For example, in the current model we could add spatial heterogeneities in the strengths of chromatin-chromatin interactions (in addition to the heterogeneities in chromatin-lamina interactions already incorporated), at the cost of adding more parameters. However, to meaningfully extract the physiological variations in the biophysical parameters, here we have considered a minimal parameter model with heterogeneities in – (a) epigenetic reaction rates, and (b) chromatin-lamina affinity. Both our theory and simulations show that these two parameters have a greater impact on the quantitative features of chromatin organization we extract than other parameters, as we explain next. From our theoretical analysis, we find that the size of heterochromatin domains in the interior of the nucleus scales as (Eq S19, SI),

$$R_d = \sqrt{\frac{3D_h}{\Gamma_{ac}\phi_{h0}} \left(\frac{\Gamma_{me}}{\Gamma_{ac} + \Gamma_{me}} \right)}$$

This scaling relationship, derived in Section S3 of the SI, incorporates roles played by the epigenetic reactions (Γ_{ac}, Γ_{me}), diffusion of epigenetic marks (D_h) as well as the chromatin-chromatin interaction energies (and hence the variations in lengths of interacting segments) which determine how compacted the nucleosomes are in the heterochromatin phase (ϕ_{h0}). However, as we see from our experiments (after GSK and TSA treatments and for cells on substrates of different stiffness), the density of the histones within the red heterochromatin clusters does not vary much spatially (Figure R2). Thus, the spatially heterogeneous energetic interactions-such as chromatin-chromatin interaction energy, do not significantly affect the parameter ϕ_{h0} , thereby not significantly impacting the size scales of the heterochromatin domains. On the other hand, we see that the epigenetic reactions strongly determine the heterochromatin domain sizes – which is further evidenced by the pharmacological perturbation of HDAC3 (TSA treatment) and EZH2 (GSK treatment) as shown in Figure R2. Furthermore, our numerical sensitivity analysis reinforces this: a 10% increase in the methylation rate leads to greater domain size changes than the same increase in interaction energies (Figure R3). Similarly, at the nuclear periphery we see that the epigenetic reactions

and the chromatin-lamina interactions are prominent determinants of the peripheral LAD thickness.

Figure R2: Effect of pharmacological treatments on histone density and methylation rates within heterochromatin domains. (Left) Comparison of histone density within heterochromatin domains before and after GSK (EZH2 inhibitor) and TSA (HDAC inhibitor) treatments, showing no significant spatial variation. The red dots represent the mean values for each condition. (Right) In contrast, methylation rates exhibit significant changes following GSK and TSA treatments, highlighting the strong role of epigenetic reactions in determining heterochromatin organization.

In response to the reviewer’s suggestion, and to further support this statement, we have now performed numerical simulations to show that changes in other model parameters (e.g. chromatin-chromatin interactions, mobility of epigenetic marks) do not prominently affect the size-scale of heterochromatin domains. These discussions have now been added to the SI section S8.

Figure R3: Sensitivity analysis of heterochromatin domain size to model parameters. Numerical simulations showing the relative impact of changes in methylation rate and chromatin-chromatin interaction energies on heterochromatin domain size. A 10% increase in methylation rate significantly alters the domain size, while a similar change in chromatin-chromatin interaction energies has a negligible effect. This demonstrates that heterochromatin domain sizes are far more sensitive to epigenetic reaction rates than to chromatin interaction strengths.

Comment 2: On Figure 4 and Figure 6, results are shown for hMSC cells as control. However, the distributions of LAD thicknesses (panel C top right) seem rather different. While I can

understand that experimentally there might be some discrepancies, this questions the validity of the approach to extract robustly interpretable parameters from experiments.

Our response: We appreciate the reviewer’s attention to the differences in LAD thickness distributions for hMSC controls in Figures 3 and 5 (Figure 4 and 6 in older version of manuscript), which helps us further reinforce the robustness of our predictive framework, as we discuss below.

We clarify that while the cells used in GSK and Y27 treatments, as well as those cultured on soft and stiff substrates, are all hMSCs, they were grown in distinct biomechanical environments specific to each experiment. Therefore, we have avoided cross-experiment comparisons. In Figures 3 and 5, although both feature hMSCs, the differing biomechanical conditions influenced chromatin organization. The following table outlines these differences for better comparison:

	Culture Condition	LAD thickness
Figure 3 (GSK)	hMSCs were cultured on soft substrates (3 kPa) with a sub-population treated with GSK.	$T_{LAD}^{control} = 211 \text{ nm}$
Figure 5 (Y27)	hMSCs were cultured on stiffened hydrogels (30 kPa) with a sub-population treated with Y27	$T_{LAD}^{control} = 123 \text{ nm}$

The choice of substrates (soft for GSK and stiff for Y27) was made to highlight key aspects of chromatin reorganization. Soft substrates, which promote high chromatin condensation, effectively illustrate GSK’s role in chromatin opening. Conversely, stiff hydrogels, where cell contractility is high, are ideal for studying Y27, which inhibits contractility; its impact on softer substrates would be less pronounced.

As the reviewer correctly noted, “the distribution of LAD thicknesses seems rather different” due to these culture conditions. However, this trend aligns with our observations: LAD thickness is greater on soft substrates (GSK controls) than on stiff substrates (Y27 controls). To further validate our framework, we performed consistency checks:

- **GSK controls on soft substrates** were compared to hMSCs cultured on soft substrates (Figure 4 vs. Figure 7, 3 kPa).
- **Y27 controls on stiff substrates** were compared to hMSCs cultured on stiff substrates (Figure 5 vs. Figure 6, 30 kPa).

Figure R4: Comparison of LAD thickness distributions under similar biomechanical conditions. (Left) LAD thickness distributions for hMSC controls cultured on soft substrates (3 kPa) as shown in Figure 3 (GSK controls) and Figure 6 (soft hydrogel, 3 kPa). (Right) LAD thickness distributions for hMSC controls cultured on stiff substrates (30 kPa) as shown in Figure 5 (Y27 controls) and Figure 6 (stiff hydrogel, 30 kPa).

These comparisons demonstrate a similar LAD thickness distribution with comparable mean LAD thickness (Figure R4), - reinforcing the robustness of our parameter extraction methodology when similar biomechanical conditions are used.

Comment 3: It is unclear how the parameters Γ_{me} and V_{LAD} are actually inferred from Eqs 5, 6 & 8 as there are other – a priori unfixed – parameters (D_h , Γ_{ac} , θ , etc.) in these equations.

Our response: To infer Γ_{me} and V_{LAD} from Eqs 1, 2, S36 (Eqs 5, 6 and 8 in older version of manuscript) we used fixed values for D_h and Γ_{ac} , which were determined based on data from the literature.

Figure R5: Numerical simulations showing the impact of 10% change in diffusion coefficient on heterochromatin domain size.

Specifically, we adopted parameter values that are experimentally validated or computationally derived, ensuring consistency within the context of our model. For example, D_h , diffusion coefficient of nucleosomes, is $\sim 10^{-3} \mu m^2/s$, based on experimental measurements of chromatin mobility [2] and Γ_{ac} , histone acetylation rate, is $\sim 10^{-2} s^{-1}$, derived from robust studies on histone modification kinetics under diverse conditions [3]. Moreover, note that even significant ($\pm 10\%$) change in the diffusion coefficient D_h does not significantly impact the heterochromatin domain size. Guided by the sensitivity analysis (Fig. R5), we fixed D_h to reduce system uncertainty, allowing us to focus on inferring the most sensitive parameters, Γ_{me} and V_{LAD} , which directly govern heterochromatin size and LAD thickness, respectively.

The choice to infer Γ_{me} rather than Γ_{ac} , was guided by biological relevance of these parameters in the context of chromatin organization. The perturbations analysed in this study, such as HDAC inhibition and EZH2 modulation, primarily affect the overall rate of histone methylation Γ_{me} (Figure 1D (Fig 1B in older version of manuscript), reproduced here as Figure R6). For

Figure R6: Epigenetic regulation of chromatin phases. HDAC and HMT, regulate the interconversion between heterochromatin and euchromatin phases through histone acetylation and methylation reactions.

heterochromatin size selection. This makes Γ_{me} a more sensitive and biologically relevant parameter for understanding the processes governing chromatin organization. Furthermore, literature values for Γ_{ac} are relatively well-characterised across different cell types and experimental contexts, allowing us to fix this parameter with confidence [3, 5]. In contrast, Γ_{me} exhibits greater variability across biological contexts, with fewer studies providing quantitative estimates of methylation rates [6]. Given this variability, inferring Γ_{me} directly from chromatin size scales observed in our STORM imaging data ensured that the derived values are specific and relevant to the chromatin organization in our study. A detailed table of parameter values, their physical interpretation, and associated references is provided in Table S3 of the SI. In this table, we explicitly outline the origins of these parameter values and how they were incorporated into calculations to ensure robustness and consistency within the modeling framework. We have also clarified this point in the revised manuscript (Section 2.3) to improve transparency regarding the derivation of Γ_{me} and V_{LAD} .

Change in Manuscript: The obtained interior domain and LAD sizes are locally related to the methylation rate Γ_{me} via Eqs 2 and Fig 2E (S36), allowing us to extract the nucleus-wide distribution of Γ_{me} . Note that to infer Γ_{me} and V_{LAD} we used fixed values for D_h and Γ_{ac} , which were determined based on data from the literature (discussed in SI section S9).

Comment 4: Authors claim that ‘the predicted spectrum of chromatin-lamina affinities align with in multiple cell lines’. A figure showing such a result from ChIP should be shown.

Our response: We appreciate the reviewer’s feedback with respect to comparing our predicted chromatin-lamina affinity spectrum with LaminB1-ChIP-seq (LB1 ChIP-seq) data.

In section 2.4 we describe how both methods consistently capture the distinct aspects of chromatin-lamina interactions. Specifically, the spectrum of chromatin-lamina affinities predicted by our framework aligns qualitatively with the classification of LADs characterized by LB1 ChIP-seq, as shown in Fig. R4. To recapitulate the key similarities,

- High-affinity regions in our framework correspond to T1 LADs from ChIP-seq, which exhibit strong chromatin-lamina interactions and transcriptional repression.
- Low-to-intermediate-affinity regions correspond to T2 LADs, with weaker interactions and higher chromatin accessibility.

This consistency illustrates the ability of both methods to distinguish regions of strong and weak chromatin-lamina interactions. However, the methodologies fundamentally differ in their (i) approach and (ii) interpretation of information they provide, as we explain next.

Firstly, in terms of their methodological approach (Fig R7 A), LB1 ChIP-seq identifies genome-wide interactions of the chromatin with Lamin B1, accounting for the interactions of specific genome regions in individual cells. Using this data over millions of cells LB1 ChIP-seq provides the genome-wide LB1 attachment probability maps. In contrast, our STORM-based framework

instance, EZH2 is a methyltransferase that catalyses H3K27 methylation, a key modification driving heterochromatin compaction and spatial segregation. Similarly, HDAC inhibition alters the histone deacetylation, indirectly affecting methylation rate and chromatin reorganization. Since these perturbations significantly alter the overall histone methylation rate, Γ_{me} is the parameter most directly linked to the experimental outcomes, making it a natural choice for inference.

Additionally, literature values for Γ_{me} , is significantly smaller ($\sim 10^{-3} \text{ s}^{-1}$ [4]) than Γ_{ac} value (10^{-2} s^{-1} [3]), meaning that variations in methylation have a more pronounced effect on

provides a single-cell, single-time-point snapshot of LAD organization, capturing dynamic, probabilistic interactions and spatial heterogeneity at the moment of cell fixation.

As an example, to illustrate this distinction, say LB1 ChIP-seq resulted in identification of 100 T1 LADs (high-affinity) and 100 T2 LADs (low-affinity). While T1 LADs are nearly always attached to the lamina (probability~1), T2 LADs may attach less frequently (say probability~0.25). In a single-cell snapshot, our framework observes only the LADs physically attached at the time of fixation. Thus, we may observe most of the T1 LADs (high-affinity) but only about 25% of the T2 LADs identified by ChIP-seq. This results in fewer observed T2 LADs, reflecting their weaker interaction strengths and lower attachment probabilities (Fig R7 B). Each approach has its strengths – while LB1 ChIP-seq provides population-averaged genomic coverage over time, offering comprehensive genome-wide coverage, our framework provides a single-cell snapshot of LAD organization, capturing dynamic, probabilistic interactions and spatial heterogeneity. When the two sets of data is integrated, a more holistic understanding of LAD organization can be achieved.

Figure R7: Comparison of chromatin-lamina interactions as observed in STORM-based single-cell analysis versus LB1 ChIP-seq

As an example of such a cooperative approach, we can use our predictive framework with ChIP-seq data to estimate attachment probabilities for specific LAD regions. From the ChIP-seq data [7], which is averaged over a large population ($\sim 10^6$ cells) we see that nearly 48% of LADs are of type T1 (very high probability of attachment, say P_{T1}) while the remaining (52%) are T2 type LADs (which attach to the lamina with a probability P_{T2}), as visualized in Fig. R8 (solid color bars, left side). Comparing this with the STORM-based theoretical framework for a single cell, we can find the values of the attachment probabilities P_{T1} and P_{T2} which will give the distribution of chromatin-lamina affinity observed in Fig. R7. For this specific cell we find that the attachment probability for T1 LADs is $P_{T1} = 0.848$, and that for T2 LADs is $P_{T2} = 0.152$, as shown in Fig. R8 (hatched bars, right side). By combining theoretical analysis and sequencing results with STORM imaging of specific epigenetic marks like H3K9me3 and H3K27me3, we can explore correlations between heterochromatin states (facultative vs constitutive) and strength of lamina attachment extracting more biophysical parameters such as chromatin-lamina attachment probabilities. This example provides a predictive framework that bridges population-level genomic data with single-cell imaging and theoretical analysis.

Figure R8: The fraction of LADs of type T1 and T2 based on population-averaged ChIP-seq data (solid color bars, left side) and single-cell STORM-based data (hatched bars, right side)

The second point of distinction is in terms of the interpretation of the information that is obtained via the two methodologies. LB1 ChIP-seq identifies “where” along the chromatin polymer are the LADs located but does not resolve their 3D spatial arrangement within the nucleus. In contrast, our framework reveals “how” LADs are organized spatially, visualizing the 3D clusters of the nucleosomes along the nuclear lamina, identifying the physics governing the size scaling of these clusters, and as well as the variabilities across cells. Together, these methods provide complementary insights: LB1 ChIP-seq identifies which genomic regions interact with Lamin B1, while our framework reveals how these interactions occur spatiotemporally at the single-cell level. This integration of genomic and biophysical perspectives provides a more complete understanding of chromatin-lamina interactions and their role in nuclear architecture.

We have now included this comparison in the SI section S12. Additionally, we have clarified this alignment in the revised manuscript (Section 2.3), emphasizing qualitative alignment of our model’s predictions with LaminB1-ChIP-seq data.

Change in Manuscript: The predicted spectrum of chromatin-lamina affinities aligns qualitatively with a range of chromatin-LaminB1 interaction strengths observed experimentally using LaminB1-chromatin immunoprecipitation (LB1 ChIP) in multiple cell types (Fig S21) [7]. Regions along the chromatin polymer exhibiting strong LaminB1 association are classified as Type 1 LADs, resembling the peak of high chromatin-lamina affinity we observe. Furthermore, as we predict, these LADs have highly compacted chromatin apparent from their very low chromatin accessibility [7]. Regions of chromatin polymer exhibiting a weaker (but nonzero) LaminB1 association, with marginally increased chromatin accessibility are classified as Type 2 LADs [7] and resemble the range of low to intermediate chromatin-lamina affinities we observe. This comparison is discussed in details in SI section S11.

Comment 5: The introduction of the manuscript consists in a small introductory paragraph with generalities followed by a kind of extended summary of the work. This is clearly not enough to

really introduce the question, the state-of-the-art and the current challenges to motivate their work. In particular, the current biological and physical knowledge on HC and LAD formation should be discussed. For example, several papers (purely theoretical and in dialog with experiments) already addressed the interplay between both (Falk et al, Nature 2019; Ulianov et al, Nat Com 2019; Sati et al, Mol Cell 2020; Chiang et al, Cell Rep 2020; Tolokh et al, Epigen. & Chrom. 2023; etc.). A fair presentation also on the use of Cahn-Hilliard-like framework to study chromatin organization including authors' previous work but also from others (Awazu et al, PRE 2015 ; Potoyan's group work; etc.) would be welcomed to highlight the originality of the presented model.

Our response: We thank the reviewer for this comment which helped us set a better perspective of our manuscripts contributions in understanding the mechanisms of chromatin reorganization. We have extensively revised the introduction to address the reviewer's suggestions and better establish the biological and physical foundations of heterochromatin (HC) and lamina-associated domain (LAD) formation. The revised text now includes a discussion of HC-LAD interactions, referencing key findings from sequencing (e.g., DamID, lamin B1 ChIP-seq) and imaging techniques (e.g., STORM), highlighting the roles of histone modifications (e.g., H3K9me2/3) and lamina proteins (e.g., LBR, LAP2 β) in chromatin-lamina tethering. We have also expanded the discussion on existing theoretical frameworks to better contextualize our approach and emphasizing how previous models lack non-equilibrium considerations like histone modification dynamics.

This revised introduction emphasizes the novelty of our multimodal framework by highlighting the following unique features:

- Inclusion of non-equilibrium considerations via active processes including histone acetylation/methylation kinetics.
- Mesoscale approach linking sub-chromosome features (e.g., LADs) to genome-wide chromatin organization.
- Integration of super-resolution imaging techniques with theoretical modelling to predict the changes in specific biophysical parameters which dictate chromatin reorganization.
- Incorporation of cell-to-cell heterogeneities via single cell imaging

Change in Manuscript: Recent advances in sequencing and imaging techniques, such as DamID, lamin B1 ChIP-seq, and super-resolution microscopy, have provided key insights into chromatin-lamina interactions. These methods have revealed the roles of specific DNA sequences, chromatin states, nuclear envelope proteins (such as lamins, LBR, and LAP2 β), and histone modifications in tethering heterochromatin to the nuclear periphery [8]. Previous studies showed that heterochromatin marks including H3K9me2 and H3K27me3 are enriched at the nuclear periphery [9, 10] and these marks are also over-represented within LADs, with H3K27me3 being particularly enriched at LAD borders [11, 12]. In addition, previous work showed a role for both H3K9me2 and H3K27me3 for the formation and maintenance of lamina-proximal positioning of loci [13, 14]. However, sequencing-based approaches often obscure cell-to-cell heterogeneity by averaging data across large populations. In contrast, single-cell sequencing and super-resolution imaging techniques like STORM and FISH offer insights at a single-cell resolution, providing a more granular view of chromatin-lamina interactions. These experiments have indeed shown cell to cell heterogeneity in chromatin folding and spatial organization. For example, STORM imaging showed that chromatin folds into heterogeneous groups of nucleosome groups called nucleosome clutches [15]. In addition, chromatin tracing studies showed heterogeneity in the boundaries of topologically associating domains (TADs) at the single cell level[16].

To integrate and help interpret information across single cell and population averaged studies, experimental techniques need to be complemented by first-principles theories of chromatin-lamina interactions along with computational approaches to infer the principles underlying chromatin organization. Computational approaches based on polymer modeling, ranging from individual nucleosomes to whole-genome simulations, as well as continuum-scale simulations

based on the Cahn-Hilliard equation, have enhanced our understanding of the principles governing chromatin organization[17-23]. While polymer models treat chromatin as a flexible sequence of beads, focusing on its folding and interactions with binding proteins [24], continuum models have been used to describe the nucleus-wide segregation of distinct chromosome territories via global free-energy minimization [25], without resolving sub-chromosome features such as chromatin compartments or nucleosome clutches. Other approaches, like the strings and binders switch (SBS) model [26], focus on chromatin-protein interactions, while the loop extrusion model explains the formation of topologically associating domains (TADs) via cohesin complexes [27]. Polymer models for interactions of chromatin with the nuclear lamina have been developed [28, 29], but they do not account for non-equilibrium contributions, such as the dynamics of methylation and acetylation reactions. As a result, they are unable to predict the shapes and sizes of LADs. The absence of these dynamic considerations limits our understanding of how molecular interactions, such as chromatin-lamina affinity, regulate the morphology of LADs and adapt to changes in the microenvironment or cell signaling.

Here we introduce a multimodal framework to decipher the spatially heterogeneous biophysical mechanisms governing chromatin organization, with a specific focus on the strength of chromatin-lamina tethering and its modulation in response to the cell's microenvironment. Our framework integrates super-resolution microscopy with a mesoscale mathematical model that accounts for interactions between chromatin segments and chromatin-lamina, as well as the kinetics of histone methylation and acetylation. This approach allows us to quantitatively extract key biophysical parameters, such as chromatin-lamina interaction strengths and histone methylation rates, from single-cell imaging data, thereby offering insights into how chromatin-lamina interactions vary across individual cells and are modulated by changes in extracellular stiffness and pharmacological treatments targeting epigenetic regulators. We show that chromatin-lamina interactions, combined with histone methylation and acetylation, mechanistically shape the morphology of heterochromatin both in the nuclear interior and periphery. Our findings suggest that softening of extracellular matrix, which occurs in diseases such as tendinosis, regulates chromatin-lamina interactions and epigenetic regulation, resulting in nucleus-wide spatial chromatin reorganization. We test this hypothesis by using our predictive framework to analyze cells from healthy donors and patients with tendinosis. Our multifaceted integrative framework, which combines theoretical, and imaging approaches, provides insights into specific mechanosensitive molecular alterations driving genome organization, and has wider implications for translating these findings into understanding pathological conditions impacting the extracellular environment.

Comment 6: Similarly, the Discussion/Conclusion section is more an extended summary of their results than an actual discussion compared to the state-of-the-art and of the limitations/caveats of their approach and interpretations.

Our response: We again thank the reviewer for the feedback on better discussing the implications of our contributions. We have substantially revised the Discussion section to move beyond summarizing results, providing a more critical analysis of our findings within the context of current research on chromatin-lamina interactions and chromatin organization models. Specifically, we now:

1. Compare our model with existing models, highlighting how our framework differs from and improves upon prior models by incorporating non-equilibrium dynamics of epigenetic reactions and accounting for cell-to-cell variability—limitations that have previously restricted the field.
2. Contextualize our findings within recent experimental advances, such as sequencing and imaging techniques (e.g., STORM), to emphasize how our model provides previously inaccessible insights into chromatin-lamina interactions.

3. Acknowledge limitations and future directions, discussing areas where further experimental validation and model extensions could enhance its applicability.

Additionally, to improve clarity and accessibility, we have also organized key predictions and implications into bullet points for better readability. These revisions provide a more balanced and comprehensive discussion of our contributions, their implications, and their limitations within the state-of-the-art research landscape.

Change in Manuscript: We begin by developing a coarse-grained mesoscale mathematical model predicting the nucleus-wide chromatin organization in response to changes in the cell microenvironment, epigenetic drugs, and pharmacological treatments affecting cell contractility. Our model captures the emergence of densely compacted heterochromatin domains of characteristic sizes in both the nucleus interior and at the periphery. Leveraging these insights, we predict the spatial chromatin reorganization due to diverse cues such as disease-driven extracellular matrix degradation. Building upon our theoretical insights, our predictive framework utilizes the morphology of peripheral heterochromatin domains, observed via super-resolution microscopy, as a measure of the distribution of chromatin-lamina interaction strengths. Our integrated framework enables the evaluation of the roles of molecular mechanisms mediating the chromatin-lamina interactions in diverse cellular microenvironments.

Chromatin-lamina interactions are pivotal in regulating spatial chromatin organization and gene expression during cell development, differentiation, and disease [8, 30]. The roles of such interactions in determining the three-dimensional genome organization, especially in the vicinity of the nuclear periphery, have been experimentally explored using sequencing-based techniques such as DamID [11, 31-33] and lamin B1 ChIP-seq [7, 34, 35] as well as super-resolution imaging-based techniques such as STORM imaging coupled with fluorescence in-situ hybridization (FISH) [14, 36] and transmission electron microscopy [37]. These observations highlight the role of nuclear envelope proteins, particularly lamins and associated factors such as LBR and LAP2 β , as well as specific histone modifications like H3K9me_{2/3} and H3K27me₃, in the formation and maintenance of heterochromatin domains that interact with lamina to enable gene silencing. Polymer-physics based computational models as well as data-driven modeling approaches have previously been used to infer the quantitative roles chromatin-lamina energetic interactions in peri-laminar genome compartmentalization [38-43]. However, these models rely on bulk sequencing techniques like ChIP-seq, which miss single-cell variations, resulting in averaged-out effects. These methods lack the ability to capture cell-to-cell heterogeneity in chromatin-lamina interactions. Our work leverages super-resolution imaging of lamina-associated domains (LADs) to capture the single-cell distribution of chromatin-lamina interactions, providing a more detailed view of how these interactions vary across individual cells. Also, these polymer models do not incorporate the non-equilibrium kinetics arising from active epigenetic regulation and diffusion kinetics, which we show critically regulate chromatin organization. Moreover, our coarse-grained continuum model, unlike the previous fine-grained polymer models, can concurrently characterize the nucleus-wide chromatin organization, i.e. both in the nucleus interior and the periphery. This presents a significant advantage by facilitating direct comparison with super-resolution imaging and quantitatively extracting the biophysical parameters governing the chromatin organization, which has previously been a challenge.

Our theoretical model predicts that LADs can display a spectrum of morphologies, ranging from small, isolated clusters to elongated formations along the lamina depending on (i) the chromatin-lamina affinity, and (ii) epigenetic reaction rates. While higher chromatin-lamina affinity causes spreading of LADs into elongated lamellar domains, histone methylation rate prominently impacts the LAD size. Hence the average thickness of the peripheral heterochromatin domains is a composite outcome of both these parameters, as shown in the theoretically predicted phase diagram (Fig S10). Our model further predicts that the size-scale of interior heterochromatin domains is uniquely regulated by the epigenetic reaction kinetics.

Thus, given a distribution of size-scales of the interior and peripheral heterochromatin domains, obtained from STORM imaging, our predictive framework can extract the nucleus-wide distribution of epigenetic reaction rates and chromatin-lamina affinity (Fig 2F). The key predictions of our multimodal framework applied to nuclei undergoing pharmacological and biophysical perturbations are:

- The distribution of chromatin-lamina affinity is bimodal, suggesting presence of regions along the lamina with very strong chromatin binding interspersed with regions with minimal LAD sequestering.
- The predictive framework, without any prior knowledge, distinguishes between TSA and GSK treatments, identifying that while both affect epigenetic kinetics, only TSA alters chromatin-lamina affinity, implicating HDAC3 in LAD formation at the lamina.
- The framework identifies upstream pharmacological perturbations in cell contractility alter both epigenetic regulation and chromatin-lamina affinity, consistent with observed nucleo-cytoplasmic HDAC3 shuttling.
- An increase in substrate stiffness alters chromatin organization by reducing the overall histone methylation rate and decreasing chromatin-lamina affinity, mediated by cytoskeletal contractility driven HDAC3 efflux from the nucleus.
- As a pathological application of the predictive framework, we predict that tendinosis-mediated tissue degeneration alters both histone methylation rates and chromatin-lamina affinity, mimicking the effect of soft substrates.

A notable outcome of our predictive framework is the observation that chromatin-lamina interaction strength exhibits a bimodal distribution, characterized by two distinct peaks (Fig 8B). This spatial heterogeneity likely stems from the variability in the composition of the lamina meshwork and the presence of diverse nuclear membrane proteins such as ion channels, nuclear pores, and chromatin-tethers like LBR and LAP2 β [44]. While proteins like LBR, LAP2 β , emerin, and HDAC3 are actively involved in tethering heterochromatin, other nuclear membrane components like ion channels, pumps and nuclear pore complexes only interact very weakly with chromatin. Our model effectively captures this inherent variability, predicting the bimodal distribution of chromatin-lamina affinity in each nuclei analyzed (Fig 2F, S17). The first peak, representing strong chromatin-lamina interactions, typically correlates with dense H2B localizations in the STORM images, manifesting as large elongated peripheral heterochromatin domains. Conversely, regions along the periphery corresponding to the second peak, characterized by minimal chromatin-lamina interactions, exhibit little to no presence of peripheral heterochromatin domains. Between these extremes, a spectrum of intermediate chromatin-lamina interactions is observed, with small and thin peripheral heterochromatin domains indicative of regions with relatively weak but still perceptible affinity for heterochromatin. Our model predictions closely align with the heterogeneity observed in Lamin B1 enrichment in genomic regions across multiple cell types, as unveiled by LB1 ChIP-seq [7]. The peak of high chromatin-lamina affinity predicted by our model likely corresponds to genomic regions with high Lamin B1 occupancy (classified as Type 1 LADs). Conversely, the predicted range of weaker but nonzero chromatin-lamina affinities may resemble genomic regions exhibiting relatively lower Lamin B1 occupancy (Type 2 LADs) in LB1 ChIP-seq [7]. Thus, by capturing this distribution of chromatin-lamina affinities, our multimodal framework offers new insights into the distinct mechanisms driving peripheral chromatin sequestration.

The role of distinct chromatin-sequestering mechanisms can be effectively probed by carrying out pharmacological perturbations and using our predictive framework to enquire the changes in the strength of chromatin-lamina interactions. For instance, we have demonstrated that treatment of nuclei with HDAC inhibitor TSA results in an overall reduction in the chromatin-lamina affinity, while treatment with GSK, inhibiting EZH2, causes no significant change. Consistent with our predictions, it has been previously reported that HDAC inhibition through

TSA or RNA-induced inhibition detaches gene loci from the lamina and relocates them to the nuclear interior [38-40, 56], implicating HDAC3 in tethering segments of chromatin to the nuclear periphery, where it interacts with LAP2 β and Emerin to suppress gene expression [21, 43].

Furthermore, via HDAC3 signaling our predictive framework links the cytoskeletal contractility to nuclear chromatin organization revealing an increase both in epigenetic reaction rates and chromatin-lamina affinity on soft substrates compared to stiff. These findings align with our prior reports that on stiff substrates, increased cytoskeletal contractility results in the shuttling of HDAC3 from the nucleus into the cytoplasm in mouse fibroblasts [45, 46]. To validate that the loss of contractility on soft substrates indeed contributes to the altered chromatin organization, we also pharmacologically disrupted cell contractility using Y27 treatment, after which we observed that both the histone methylation rate and chromatin-lamina affinity increase—similar to nuclei on soft substrates. However, after Y27 treatment the increase in chromatin-lamina affinity is more pronounced than on a soft substrate resulting in excessive sequestering of chromatin to the lamina reducing the size of the interior domains (as discussed in Section S10). This may be mediated by altered chromatin-binding after Y27 treatment due to mechanisms in addition to HDAC3 [47, 48]. While this requires further investigation, it can be easily incorporated into the model by suitably adapting the mechanosensitive chromatin-lamina interactions as a function of such binding mechanisms. Thus, our observations strongly suggest that via the shuttling of epigenetic factors like HDAC3, cytoskeletal contractility chemo-mechanically transduces environmental stiffness stimuli to the nucleus and drives chromatin reorganization.

Our predictive framework, encompassing image quantification and subsequent parameter-extraction algorithms, is not limited to STORM and can be easily adapted to other super-resolution techniques enabling the visualization of heterochromatin domains.

Comment 7: TSA is affecting all HDACs and is likely to be a huge perturbation of the system. Authors relates the observed changes in parameters to the role of HDAC3. Is there a way to specifically target HDAC3 to experimentally validate this hypothesis?

Our response: We agree with the reviewer that TSA, as a pan-HDAC inhibitor, affects multiple HDACs, not just HDAC3, and therefore represents a broad perturbation to the system. However, literature evidence suggests a specific role for HDAC3 in mediating chromatin binding at the nuclear lamina. For instance, Somech et al. (2005) demonstrated that HDAC3 is enriched at the nuclear periphery and plays a key role in facilitating chromatin-lamina interactions. Based on these findings, we attribute the observed changes in chromatin-lamina affinity primarily to HDAC3, while broader changes in methylation levels are likely due to the combined inhibition of multiple HDACs by TSA.

To experimentally validate this hypothesis and isolate HDAC3's role, future studies could employ HDAC3-specific inhibitors such as RGFP966, which selectively inhibits HDAC3 without affecting other HDACs. Alternatively, HDAC3 knockdown via siRNA could be used to confirm its specific contribution to chromatin-lamina interactions. These experiments would help refine our understanding of HDAC3's role in mediating the observed changes in chromatin-lamina affinity and methylation but remain outside the scope of our current work.

Minor

Comment 1: In Nat Coms, normally, Materials and Methods are localized after the Conclusion/Discussion section.

Our response: We thank the reviewer for pointing this out. In the revised manuscript, we have rearranged the sections to place the Materials and Methods section after the Discussion, aligning with the format requirements of *Nature Communications*.

Comment 2: Page 4, the citation of Ref[14] may seem inappropriate.

Our response: We thank reviewer for bringing this to our attention. Upon revisiting the citation of Ref [14] on page 4, we agree with the reviewer's observation that it is not directly relevant to the context of chromatin organization and the role of energetic interactions between euchromatin and heterochromatin. While our initial reasoning for including this reference was to draw a parallel to theoretical frameworks describing energy landscapes and nonequilibrium systems, we acknowledge that the paper does not specifically address the depth of energy wells in chromatin organization. To improve clarity and ensure relevance, we have removed Ref [14] from page 4.

Comment 3: Page 5, Eq.3 is cited before it is written.

Our response: We thank reviewer for pointing this out. We apologize for the oversight. In the revised version, we have corrected the placement of Eq. 3 so that it is introduced in the proper order. We have also reviewed the manuscript carefully to avoid similar errors.

Comment 4: It is unclear if super-resolution images are 2D or 3D.

Our response: The super-resolution images used in this study are 2D [15]. To address this, we have clarified this detail explicitly in the revised manuscript.

Change in Manuscript:

1. ...to extract the epigenetic reaction rates of methylation Γ_{me} and the strength of the chromatin-lamina interactions relative to chromatin-chromatin interactions V_{LAD} from 2D STORM images.
2. The 2D-STORM images were obtained from previously published work.

Comment 5: To infer LAD from STORM images, authors use a threshold (2.5% of nuclear radius). What is the rationale for this choice? Are the results (and following parameter inference) robust?

Our response: We thank the reviewer for this insightful comment. The threshold of 2.5% of the nuclear radius used to infer LADs from STORM images is based on the approximate size and localization of nuclear lamina proteins and their interaction zones. Lamina-associated proteins, such as Lamin B1 and associated chromatin interactions typically localize within a region of ~100–300 nm from the nuclear envelope [32, 49]. For nuclei with radii ranging from 5000-9000 nm, this corresponds to approximately 2-3% of the nuclear radius, making 2.5% threshold biologically relevant.

To ensure the robustness of this threshold, we had performed sensitivity analyses by testing slightly lower and higher thresholds for which data was not shown. These tests showed that the key features of LAD organization and the inferred parameters (Γ_{me} and V_{LAD}) do not alter significantly within this range (Fig R9), confirming the robustness of our conclusions.

Figure R9: Sensitivity analysis of LAD parameters to the search region threshold. Violin plots showing the distribution of LAD thickness (left), methylation rate (Γ_{me} , middle), and chromatin-lamina affinity (V_{LAD} , right) for varying LAD search region thresholds (2.0%, 2.5%, and 3.0% of the nuclear radius).

However, thresholds that were too low failed to capture chromatin regions near the lamina that visually correspond to LADs, while thresholds that were too high began misclassifying some interior chromatin domains as LADs (Fig R10). This analysis confirmed that the 2.5% threshold strikes an optimal balance, accurately capturing LADs while avoiding misclassification of interior domains (Fig. R10).

Figure R10: The figure shows the impact of varying thresholds on LAD identification, with the 2.5% threshold (optimal) accurately capturing LADs, while higher or lower thresholds lead to misclassification (pink arrows) or missed (pink arrows) LADs (bottom row).

Additionally, we visually inspected the STORM images and confirmed that LAD regions identified using the 2.5% threshold closely align with high-density chromatin regions (red in Fig R10) expected to be associated with the nuclear lamina (Fig R10).

This explanation has been added to SI Section S11 to clarify the rationale behind this choice and demonstrate its robustness.

Comment 6: Just after Eq.4: area -> volume, no?

Our response: All derivations were done in 2D to be consistent with the 2D STORM images used in our analysis. Hence the theoretical derivations involved changes in the area of the heterochromatin domains. However, we have also shown in the SI that the scaling relationship derived theoretically is consistent in both 3D and 2D.

Comment 7: The paragraph ‘As a second step, we validate ...’ seems to claim some statements but do not refer to any figures.

Our response: We thank the reviewer for raising this point. The paragraph beginning with "As a second step, we validate..." is primarily a methodological description, explaining the numerical simulations used to validate the extracted parameter distributions and predict chromatin organization. As such, this section does not directly refer to any specific figure, as it outlines the conceptual and computational approach rather than presenting new results.

To ensure clarity, we have carefully reviewed the text and confirmed that it serves as an explanation of methodology, with figures referenced where appropriate elsewhere in the manuscript to illustrate the results of these simulations (e.g., Fig. 4-8).

Change in Manuscript: Further, we ensure that the distributions of Γ_{me} and V_{LAD} extracted using the theoretical framework are accurate despite the simplifying assumptions involved in the theoretical derivation of Eqs 1-2 and S36 (numerically depicted in Fig 2E). The mean values of the extracted parameters Γ_{me} and V_{LAD} are used as inputs to simulate chromatin organization in the nucleus via numerical solutions of phase-field equations (Eqs 3-5) using COMSOL Multiphysics. The pharmacological or biophysical perturbation driven changes in the distributions of biophysical parameters, extracted from the theoretical framework, are incorporated into the simulation by a proportionate change in the mean of the parameters Γ_{me} and V_{LAD} . By quantifying the simulation predicted chromatin reorganization due to changes in parameters we assess the relative changes in the LAD and interior domain size-scales, validating these changes against those observed in cells (Fig 3D, 4C, 5B, 6D, 7C).

Comment 8: Figs 4- 8, while the differences between both conditions are clearly visible on the STORM images, this is quite subtle on the simulations. Can the authors comment? Maybe showing a zoom of the simulation as in Fig2C would help.

Our response: We thank the reviewer for this observation. We agree that while the differences between the conditions are striking in the STORM images, the differences in the simulations may appear subtler due to the spatial averaging inherent in the simulation approach. Simulations are based on extracted mean values of the parameters Γ_{me} and V_{LAD} , and spatial heterogeneities are sampled from normal distributions. This can result in less pronounced visual contrasts compared to the single-cell resolution of the STORM images, which directly capture experimental variability.

Figure R11: Numerical predicted change in heterochromatin domain size (A) and LAD thickness (B) in nuclei undergoing chemical-mechanical alterations.

To address this, we have already quantified the differences in chromatin organization predicted by the simulations. These quantitative metrics, including changes in LAD thickness and interior domain sizes, are presented in SI Fig S14 (Fig S12 in older version of manuscript) reproduced here as Fig R11, providing a clearer and more objective comparison between the conditions. We believe this quantification sufficiently highlights the differences predicted by the simulations, complementing the experimental data shown in the STORM images.

Comment 9: Figs4-8: authors made simulations using the average parameter values. What would happen if instead they input the actual inferred spatial distribution of parameters?

Our response: We thank the reviewer for this insightful comment. In the current simulations (Figs. 4–8), we used the average parameter values for Γ_{me} and V_{LAD} , with spatial heterogeneities sampled from normal distributions. This approach captures the key statistical features of chromatin organization while maintaining computational feasibility.

While incorporating the actual inferred spatial distributions of these parameters could provide finer detail, it would significantly increase the complexity of the simulations without substantially altering the results or providing substantial new insights into the system's behavior. We believe the current approach strikes a balance between computational efficiency and capturing the key statistical features of the data. To clarify this point, we have included this explanation in SI section S8.

Change in Manuscript: While the current simulations use spatially averaged parameter values supplemented with noise, incorporating the actual inferred spatial distributions of $\tilde{\Gamma}_{me}$ and \tilde{V}_{LAD} could enable finer spatial detail by directly mapping experimentally derived spatial profiles into the model. This would account for localized variations in methylation reaction rates and chromatin-lamina interactions across individual nuclei. However, such an approach would significantly increase computational complexity without significantly altering the key results, as the current method already reproduces the main experimental observations. For example, the simulated change in heterochromatin domain sizes and LAD thicknesses align well with the in-vitro measurements obtained from STORM imaging after treatments (Fig. S14, Table S4).

Comment 10: Fig.S1: the terms cis and trans diffusion should be more clearly defined.

Our response: We agree that clarifying the terms cis and trans diffusion will enhance reader understanding. In response, we have added a more detailed description of these terms in SI section S1.4, specifically explaining:

Note that during the sequence of events described here, and depicted in Fig S1d, the epigenetic marks (i.e. red and blue beads in Fig S1d) are effectively diffusing along the chain of the chromatin polymer. This is an effective cis-diffusion and is also depicted in the schematic cartoon shown in Fig S1c, where the red-marked heterochromatic nucleosomes moved along the polymer to compact together to form the heterochromatic domain. However, in three-dimensions this effective diffusion need not be limited to along the polymer chain. For example, consider three red-marked heterochromatic nucleosomes which are not neighbors along the polymer chain, but are spatially close due to the three-dimensional arrangement of the chromatin polymer (as depicted in schematic cartoon in Fig S1c). Analogous to the 1D example in Fig S1d, the epigenetic reactions may (in a two-step process) convert heterochromatin and euchromatin such that the epigenetic marks effectively diffuse across the chain, which we called a trans-diffusion of epigenetic marks.

These clarifications have also been referenced in the caption for Fig. S1 for additional clarity.

Comment 11: Eqs S5 and S6, the meaning of 'epigen' and 'cons' is unclear.

Our response: We thank the reviewer for highlighting this issue. We agree that clearer definitions will enhance the readers' understanding of these terms. To address this, we have added a more detailed explanation in SI section 1.3, clarifying:

- Epigen represents the non-conservative contributions from epigenetic reactions, specifically capturing the effects of methylation and acetylation rates on transitions between heterochromatin and euchromatin phases.
- Cons denotes the conservative contributions, associated with the diffusion-like evolution of epigenetic marks.

Together, these terms describe the overall rate of chromatin state changes, where epigen reflects reaction-driven transformations, and cons represents the gradual redistribution of epigenetic marks via diffusion.

Change in Manuscript: The overall rate of acetylation reaction (at a rate Γ_{ac}) is a combination of its substeps such that $\frac{1}{\Gamma_{ac}} = \frac{1}{\Gamma_{HDM}} + \frac{1}{\Gamma_{HAT}}$ (Fig 1D). Similarly, the euchromatin phase is converted into heterochromatin via an overall methylation reaction (at a rate Γ_{me}) incorporating its sub steps of histone deacetylation and methyltransferase activity such that $\frac{1}{\Gamma_{me}} = \frac{1}{\Gamma_{HDAC}} + \frac{1}{\Gamma_{HMT}}$ (Fig 1D). The reaction kinetics driven by the epigenetic regulation of overall acetylation and methylation result in the conversion of euchromatin phase into heterochromatin phase at a rate,

$$\left. \frac{\partial \phi_d}{\partial t} \right|_{\text{epigen}} = 2(\Gamma_{me} \phi_e - \Gamma_{ac} \phi_h) \quad (\text{S1})$$

However, the rates of reactions (Γ_{me} and Γ_{ac}) in Eq. S5 are not constant, but dependent on the strength of energetic interactions between neighboring nucleosomes, as we clarify next.

The chromatin-chromatin energy landscape has two stable wells as shown in Fig 1E. The heterochromatic well (red in Fig 1E) is marked by high ϕ_h (~ 0.7), which indicates that a high volume of heterochromatic nucleosomes, i.e. tightly compacted heterochromatin, is energetically favored. The other well (blue in Fig 1E) is rich in water with euchromatic nucleosomes, with a minimal presence of heterochromatic nucleosomes. Thus, the energy landscape results in like-marked (heterochromatin-heterochromatin or euchromatin-euchromatin) neighbors being more stable than unlike-marked (heterochromatin-euchromatin) neighbors. When reactions convert heterochromatin into euchromatin, or vice-versa, the reactions that lead to the formation of more like-marked nucleosome neighbors will be preferred over the reactions that lead to more intermingling of heterochromatin and euchromatin. We have shown previously [19] and again schematically in Section S1.4 below that such neighborhood dependent reaction kinetics effectively emulates a diffusion-like conservative evolution of epigenetic marks. In other words, the epigenetic marks are effectively spatially diffusing, resulting in coarsening of the heterochromatin and euchromatin phases (as shown schematically in Fig S1 and explained in Section S1.4). This effectively conservative evolution can be written as,

$$\left. \frac{\partial \phi_d}{\partial t} \right|_{\text{cons}} = M_d \nabla^2 \mu_d \quad (\text{S2})$$

Here, M_d is the effective mobility of epigenetic marks in nucleus, the physical value of which we have previously estimated from experimentally observed mean-squared displacement curves

Comment 12: Fig.S11: it seems that there is a misleading caption.

Our response: We thank the reviewer for pointing this out. We agree that the caption for Fig. S13(Fig S11 in older version on manuscript) was misleading. In response, we have revised

the caption to ensure it accurately reflects the content of the figure and improves clarity for the readers.

Change in Manuscript: Figure S13: Size distribution of the heterochromatin domains obtained numerically in the interior (left) and at the periphery (right) of the control nucleus.

Comment 13: Fig.S13: how to explain the very high heterogeneities in HC repartition among the different cells (entire regions seems depleted in HC)? Is it the nucleolus?

Our response: We thank the reviewer for this insightful observation. The high heterogeneity in HC distribution observed among different cells in Fig. S17(Fig S13 in older version of manuscript) likely corresponds to distinct nuclear regions with reduced DNA density. In addition to euchromatin, these depleted regions may indeed represent nucleoli, RNA-rich compartments, or other nuclear substructures that are typically low in heterochromatin.

Additionally, such variability could also arise from technical factors, such as what part of the nucleus is captured within the focal plane during imaging. We have clarified this in the SI section S10 to address this point.

Change in Manuscript: These heatmaps show the spatial variability in chromatin distribution, including regions of low heterochromatin density that may correspond to nucleoli, RNA-rich compartments, or other nuclear substructures, as noted in Fig. S17. Such regions are typically low in heterochromatin and contribute to the observed heterogeneity. Additionally, certain technical factors, such as what part of the nucleus is captured within the focal plane, may amplify this variability.

Comment 14: The paper is written towards a physics/biophysics audience. It might be very difficult for a biologists interested in LAD formation to grasp the methodology.

It is not a problem per se but the authors may want to consider this point if they also aim their results at biologists.

Our response: We agree and appreciate the reviewers point about the importance of making our work accessible to a broader audience, including biologists interested in LAD formation. To enhance accessibility, we have revised the manuscript to include more intuitive explanations, ensuring that the methodology and results are understandable across disciplines. In particular, we have made specific adjustments in Section 2.1 and 2.2 (presenting theoretical results) to enhance clarity for a broader audience. For example, complex mathematical derivations have been moved to the SI replacing them in the main text with simplified explanations and conceptual descriptions, using intuitive analogies to convey the principles governing LAD formation. These revisions frame the theoretical results in a biologically relevant context, minimizing the need for advanced physics knowledge while maintaining scientific rigor. We believe these adjustments improve the manuscript's accessibility without compromising its appeal to the physics/biophysics audience.

Change in Manuscript: 2.1 Chromatin Organization Features Interior Heterochromatin Domains and LADs with Distinctive Sizes

To understand how chromatin is spatially segregated into heterochromatin and euchromatin, we used STORM microscopy [15, 50] to visualize the distribution of histone H2B in the nuclei of human mesenchymal stromal cells (hMSCs) cultured on a glass substrate at nanoscale resolution (~20 nm). To gain quantitative insights into the spatial chromatin organization, we used Voronoi tessellation-based segmentation to identify regions of high H2B density (defined as inverse of the Voronoi polygon area [50]) as shown in Fig 1A. These high H2B density regions correspond to tightly packed heterochromatin (HC, red in Fig 1A), while low-density regions indicate euchromatin (EC) (discussed in detail in Material and Method section 4.4). These images reveal that localizations of H2B form distinct clusters, that can be visualized

both at the periphery of the nucleus and away from the periphery in the nucleus interior (Fig 1A, zoom). We used a density-based spatial clustering algorithm to identify the HC clusters from the H2B localizations and classified them as peripheral or interior HC domains depending on their distance from the nuclear periphery in the STORM images.

Further, we observed that interior HC domains exhibit a characteristic size distribution. On quantifying the sizes of the interior domains, by estimating their area to measure their radii (Fig 1A, Bottom), we observed that the interior domain radii follow a statistical distribution about a characteristic mean radius of ~ 50 nm (Fig 1B, Top). Similarly, for the peripheral heterochromatin domains (herein referred to as lamina associated domains, LADs), we measured their thickness from the nuclear periphery (Fig 1A, Right) as described in Material and Method section 4.4. We find that the distribution of LAD thicknesses along the nuclear periphery also exhibits a characteristic mean of ~ 200 nm (Fig 1B, Bottom). Notably, in addition to their characteristic thickness, LADs exhibit diverse shapes ranging from small bead-like to elongated domains (Fig 1A, Right).

Despite the literature focus on the spatial organization of chromatin, its quantification in terms of the characteristic sizes and shapes of the HC domains as well as the underlying biophysical origins of such size selection are not yet known. To uncover the key biophysical determinants of the chromatin organization in terms of the morphologies of interior HC domains and LADs, we next discuss the theoretical framework we have developed based on first-principles of non-equilibrium thermodynamics and active epigenetic reactions.

2.2 Theory uncovers the biophysical determinants of heterochromatin shape and size

Our mathematical model incorporates the energetics of chromatin-chromatin and chromatin-lamina interactions coupled with the kinetics of nucleoplasm diffusion and spatially inhomogeneous interconversion of acetylated and methylated histones via epigenetic regulation (Fig 1C). The coupled energetic and kinetic contributions lead to a far-from-equilibrium segregation of chromatin into compacted heterochromatin domains in the interior as well as at the periphery of the nucleus. We integrate our theoretical predictions with the experimentally observed chromatin organization quantified via super-resolution STORM imaging to extract the genome-wide distributions of key biophysical parameters, including the strength of chromatin-lamina interactions and methylation rates. This integrative framework allows us to evaluate changes in the distributions of these biophysical parameters, shedding light on the mechanisms underlying the chromatin reorganization due to pharmacological treatments, biophysical signaling perturbations and disease conditions. A comprehensive description of the modeling framework, experimental protocols, and imaging techniques is provided in the Materials and Methods section.

Sizes of the interior heterochromatin domains are determined by the balance between epigenetic reactions and diffusion kinetics:

To understand chromatin organization in the nucleus, we first consider the formation, growth and stabilization of heterochromatin domain in the nucleus's interior, away from the nuclear periphery. The initial homogeneous state of chromatin with HC and EC volume fractions, $(\bar{\phi}_h, \bar{\phi}_e)$, is determined by the reactions and do not lie in either of the energy wells (white circle in Fig 1E). However, the free energy can be lowered by nucleating heterochromatin domains (red circle in Fig 1E) surrounded in the immediate vicinity by euchromatin (blue circle in Fig 1E). The resulting concentration gradient (as shown in Fig S2 SI) drives a flow of methyl marks toward the domain, promoting its growth (Fig S2 SI). At the same time, within the growing domains, methylated histones are epigenetically modified into acetylated euchromatin. These newly acetylated histones are pushed out of the domain, due to preferential interactions of like-marked histones. This epigenetic reaction-driven outflux of euchromatin marks opposes the diffusive influx of methyl marks, establishing a kinetic balance that leads to a quasi-periodic

distribution of heterochromatin domains stable against further growth (Fig 2B). A detailed description of the growth and stabilization of the heterochromatin domains is given in the SI (Section S3). The kinetic balance gives the characteristic steady-state radius R_d for interior heterochromatin domain, away from the nuclear periphery as,

$$R_d = \sqrt{3 \frac{D_h \bar{\phi}_h^*}{\Gamma_{ac}}} \quad (2)$$

Here, D_h is the diffusion constant, $\bar{\phi}_h^*$ is the far-field heterochromatin concentration, and Γ_{ac} is the acetylation rate. The $D_h \bar{\phi}_h^*$ term in Eq 2 corresponds the heterochromatin influx which is opposed by the acetylation driven outflux (corresponding to the Γ_{ac} term), determining the size of the interior heterochromatin domains. Next, we determine the size scaling of the LADs at the nuclear periphery with respect to the histone methylation rate and the strength of chromatin-lamina interactions.

..Increase in chromatin-lamina interactions lead to preferential localization of histones modified with repressive methylation marks along the nuclear periphery, thereby affecting the shape of the LADs, which we quantify via their thickness T_{LAD} (Fig 2C). The dependence of LAD thickness T_{LAD} on the chromatin-lamina affinity, relative to chromatin-chromatin interaction strength V_{LAD} determined numerically is shown in Fig 2D (refer SI section S6 for detailed numerical methodology). At low chromatin-lamina affinity relative to chromatin-chromatin interactions, methylated histones preferentially interact with other methylated histones rather than with the lamina leading to the formation of bead-like peripheral heterochromatin domains, with minimal effective LAD thickness (Fig 2B, 2C) with contact angles, θ close to 90° . A slight increase in chromatin-lamina affinity within this regime (Regime I, Fig 2D) does not significantly affect LAD morphology, as chromatin-chromatin interactions dominate over chromatin-lamina interactions. However, as the chromatin-lamina affinity continues to increase the HC domains spread with smaller contact angles and elongate along the lamina forming more lamellar LADs. This decreases the spacing between the LADs effectively increasing the LAD thickness (Regime II, Fig 2D). In Regime III, high chromatin-lamina affinity results in a near-continuous HC layer along the lamina (Fig 2D), with the maximal LAD thickness.

In addition to the chromatin-lamina interactions, the epigenetic reactions also contribute to the increase in LAD size. The stable size of the peripheral heterochromatin domains is determined by the kinetic balance between the diffusion-driven influx of methylated histones (due to their preferential interactions with each other) and the acetylation-driven outflux of histones as in the case of interior domains. Thus, as the overall histone methylation rate increases (or acetylation rate decreases) the size of LADs increases, effectively increasing their thickness. Numerically (as depicted in Fig 2E), we find that for all levels of chromatin-lamina interactions, an increase in methylation rate enhances heterochromatin content in the nucleus, amplifying LAD thickness. Further we theoretically and numerically confirm that the thickness of the LADs follows distinct scaling relationship with both epigenetic reactions and chromatin-lamina affinity as discussed in detail in the SI (Section S7). Thus, the chromatin-lamina interactions and the epigenetic reaction rates respectively determine the shapes and the sizes of the peripheral heterochromatin domains, effectively determining the LAD thickness.

Reviewer #2

In this manuscript, Dhankhar et al examine biophysical properties (“size and shape”) of LADs by integrating analysis of microscopy data and results from a theoretical modeling framework they have implemented. Experimental manipulations such as inhibition of EZH2, TSA, various cell culture surface hardnesses and use of tendinosis patient-derived cells provide some functionality to their analyses and modeling framework. Using this combined approach, the authors make inferences from their modeling on critical parameters driving LAD morphology, and overall “underscore the pivotal role of the microenvironment in shaping genome

organization”.

Overall, I find the manuscript too complicated, confusing, with a constant and forth between analysis of imaging data and “theoretical modeling” making it unclear which contributes to which and where conclusions are drawn from. It is difficult to follow a thread and the main messages are unclear (which are they?). The novelty of their findings, if any, is not apparent and possibly erroneous because the rationale and parameters they focus on throughout the paper are too simplistic. The conclusions drawn from modeling are consequently withdrawn from many other biological observations in the literature. It is to me completely unclear (and possibly erroneous) how the various parameters tested affect “methylation rates” and what this has to do with LAD properties (see also below on methylation). Reading (and understanding) the manuscript with either a biology mindset is to say the least extremely challenging. This is also the case reading and evaluating it from a physics/mathematics/modeling perspective: while the mathematics behind it are a priori correct, the design, set up and presentation throughout the paper is simply too overwhelming. The nomenclature / abbreviations chose is not intuitive, hard to recall from one experiment to the next, and overall hard to follow. In short, the paper contains too much data hiding the main purpose, conclusion and novelty (if any) of the findings, too long and too confusing. The interplay between analysis of microscopy images and use of the “theoretical framework” in drawing conclusions is unclear to me.

General Response:

We sincerely thank the reviewer for their thoughtful feedback and detailed evaluation, which has provided us valuable opportunities to critically refine and improve our manuscript. We have now undertaken significant revisions to improve the clarity and the structure of the manuscript and have addressed all the concerns from the reviewer.

To begin with, we wish to clarify key aspects of our framework’s predictive capabilities and its novel contributions. We have identified four major points the reviewer is focusing on in the general and specific comments. To avoid repetitive discussions of these points in our responses to the specific comments, we address these here in detail.

1. Clarity of the novelty:

Our study addresses a critical gap in obtaining the strength of chromatin-lamina interactions, which regulate gene activity by forming lamina-associated domains that preserve cellular identity through gene repression, but cannot be directly measured, even with high-resolution sequencing methods. While advanced imaging techniques like STORM can reveal the size and spatial organization of LADs, these observations alone are insufficient to quantify the lamina interaction strengths and their distribution in healthy and diseased cells. Our work bridges this gap by first developing a theoretical framework that relates the shape and sizes to LADs to the interactions, which when compared with high resolution images of LADs enables the prediction of chromatin-lamina interaction strengths. This predictive capability allows us to reveal new biological insights, including (a) the uncovering specific pathways (e.g., tissue stiffness driven cell contractility, nucleo-cytoplasmic shuttling of epigenetic factors, and LAP2 β -HDAC3 driven chromatin-lamina anchoring) that govern chromatin reorganization, and (b) how different epigenetic perturbations differentially contribute to and influence LAD morphology. For instance, while GSK treatment (EZH2 inhibition) selectively reduces histone methylation, TSA treatment (HDAC inhibition) alters both chromatin-lamina interactions and methylation levels, highlighting HDAC3’s role in mediating chromatin tethering to the nuclear periphery. Furthermore, diseased tenocyte nuclei exhibit increased LAD thickness and heterochromatin size, driven by elevated histone methylation and chromatin-lamina affinity, mirroring chromatin remodeling under soft mechanical cues. This suggests a compensatory epigenetic response to extracellular matrix softening, linking tissue degeneration to nuclear organization.

2. Presentation Complexity and Accessibility:

The reviewer raised concerns that “Reading (and understanding) the manuscript with either a biology mindset is to say the least extremely challenging...”. In response, we have streamlined the manuscript to emphasize the biological motivation behind the theory, presenting the physics concepts in an accessible manner for a broad audience. Detailed derivations and equations have been relocated to the Supplementary Information, with their biological relevance explained in the main text. This approach ensures that the main text remains concise, accessible to a wider audience, and scientifically rigorous.

3. Integration of experimental and theoretical components:

The reviewer comments that “The interplay between analysis of microscopy images and use of the “theoretical framework” in drawing conclusions is unclear to me”. Integrating modeling and experimental data allows us to identify the specific mechanisms predicted by the theory that drive variations in LAD morphology (the distribution of its shape and size) in response to microenvironmental cues. Using STORM imaging, we quantify LAD shapes and sizes, while the theoretical framework uses this data to reveal the biophysical parameters underlying LAD reorganization. We validate this framework with known pharmacological treatments to ensure biological fidelity and predict the parameters influenced by changes in substrate stiffness (in vitro) and tissue degeneration (in vivo). For example, on stiffer substrates, quantification of STORM imaging revealed a reduction in LAD thickness and interior heterochromatin domain size. By integrating these experimental measurements, the theoretical framework identified this reorganization as being driven both by a decrease in methylation rate and chromatin-lamina affinity. This prediction aligns with known mechanotransduction pathways, where increased stiffness leads to higher cytoskeletal contractility resulting in shuttling of HDAC3 out of the nucleus, reducing histone deacetylation and weakening chromatin-lamina interactions. This highlights how the model connects measurable LAD properties to the underlying biological mechanisms driving chromatin reorganization in response to mechanical cues.

4. Role of methylation rate Γ_{me} :

In our theory, the overall rate of methylation represents the rate of conversion of acetylated histones into methylated histones. It is a multistep process that encompasses both histone deacetylation (by HDACs) and methyltransferase activity (by HMTs), as shown in Fig R12. Changes in either HDACs or HMTs activity can affect this rate. For example, TSA inhibits HDAC activity, reducing the availability of deacetylated lysines required for methylation, thereby lowering the overall methylation rate. Similarly, substrate stiffness influences cell contractility and nuclear HDAC3 localization via well-documented mechanotransduction

Figure R12: Epigenetic regulation of chromatin phases. HDAC and HMT, regulate the interconversion between heterochromatin and euchromatin phases through histone acetylation and methylation reactions.

pathways [45, 51-53], thus affecting the overall methylation rate. The reviewer’s comments, such as “I don’t see how the authors can make implications of manipulating HDAC activity with TSA on “the overall rate of histone methylation”” suggest a possible misunderstanding of this parameter’s generalized nature in our framework. We have revised the manuscript to provide additional context in the results section (Section 2.2) and eliminate potential misinterpretations of methylation rate Γ_{me} .

We are confident that our responses to the reviewer’s comments, along with revisions we have now made to manuscript, address all concerns raised and reinforce our work’s novelty, rigor, and significance in the field of chromatin biology and nuclear mechanics. We thank the reviewer again for their comments, which have helped us improve the manuscript.

Reconciling Reviewer Comments with Established Evidence in the Literature

We would also like to explicitly point out what we consider to be a potential misinterpretation in the reviewer's comments (Comments 6 and 8) regarding substrate stiffness and methylation, as well as Lamin proteins and chromatin organization. These topics are extensively supported by existing literature and central to our study.

Figure R13: HDAC3 fluorescence intensity and its quantification, showing decrease in nuclear HDAC3 with increasing substrate stiffness

nucleo-cytoplasmic HDAC3 shuttling is well supported by several studies [45, 46, 54-56] performed in our lab as well as others. These results show that substrate stiffness alters cell contractility as well as nuclear tension, in turn affecting the nuclear localization of histone-modifying enzymes like HDAC3. To further justify this, we have experimentally shown that HDAC3 is enriched into the hMSC nuclei on a soft substrate, whereas its levels are significantly reduced in the nuclei on a stiff substrate (Figure 6A (Fig 7A in older version of manuscript), reproduced here as Figure R13). As explained in point 4 above, HDAC3 availability influences methylation dynamics.

2. Lamin Proteins and Chromatin Organization (Comment 6)

The reviewer suggests that “HDAC3 also interacts with lamin A/C is ignored in the paper”. On the other hand, our study explicitly considers chromatin-lamina interactions, specifically via proteins such as lamin A/C, LAP2 β (which mediate the HDAC3-chromatin-lamina anchoring). These interactions are captured by the chromatin-lamina affinity parameter in our framework, which determines the shape of LADs.

Specifically, we show that weak chromatin-lamina interactions result in bead-like LADs, while stronger interactions produce lamellar LADs spread along the nuclear lamina. This demonstrates the importance of lamin-mediated chromatin organization, a topic addressed in detail in the manuscript.

Comment 1: I don't understand the rationale for manipulation EZH2 activity (and conclusions on “methylation reactions”) since it is already known that H3K27me3, catalyzed by EZH2, plays essentially no role in the establishment or maintenance of LADs. In fact, the authors find the GSK inhibitor they are using has no effect on LADs – this could be predicted from previous studies, so I don't quite see the point of these experiments.

Our response: We thank the reviewer for their comment.

In this work, we present a novel theoretical framework that enables the direct measurement of chromatin-lamina interaction strength and epigenetic reaction rates from image-derived data, without the need for prior knowledge of experimental conditions or biological details. To validate the predictive power of our framework, we first apply it to two well-characterized scenarios: GSK343 and TSA treatments. By showing that the framework accurately predicts “the known” biological effects of GSK343 and TSA, we establish confidence in its predictive ability. With this validated model, we then examine more complex scenarios, such as the impact of microenvironment stiffness on chromatin reorganization.

1. Substrate Stiffness and Epigenetic Regulation (Comment 8)

The reviewer questions the role that “surface hardnesses” play in epigenetic regulation, stating that “difference in “methylation rates” from increase surface stiffnesses which are very hard to believe”. The connection between substrate stiffness, cell contractility and

While the role of H3K27me3 in LAD formation is not well established and potentially underappreciated, several studies including our own work and others have shown a relationship between H3K27me3 and LADs. Specifically:

- Luo et. al. [9] demonstrated enrichment of H3K27me3 at the nuclear periphery.
- Guelen et. al. [11] found that H3K27me3 is overrepresented within LADs, especially at LAD borders.
- Harr et. al. [13] showed that targeted recruitment of YY1 protein facilitated ectopic LAD formation in a manner dependent on H3K27me3 and H3K9me2/3, as well as Lamin A/C. This study supports the role of H3K27me3 in maintaining lamina-proximal chromatin positioning.
- Our own work [1] demonstrated that substrate stiffness-dependent chromatin localization to the nuclear periphery is mediated by EZH2 and H3K27me3. Inhibiting EZH2 with GSK343 abolishes chromatin relocalization to the periphery on soft substrates.

In light of this evidence, the use of GSK343 in our current study serves a dual purpose: (1) validating our theoretical framework against a known perturbation, and (2) further elucidating the mechanistic role of methylation in LAD dynamics.

Importantly, even within the well-established contexts of GSK343 and TSA treatments, our analysis uncovers new insights into how chromatin-lamina interactions and epigenetic processes jointly regulate LAD thickness (Fig. R14). This highlights the framework's utility in

Figure R14: Quantification of STORM Images in control and GSK treated nuclei showing the distribution of LAD thickness (Left). Numerical simulation showing the effect of methylation reaction on LAD thickness (Right).

unraveling the distinct contributions of coupled biological processes. Specifically, the methylation rate alters the population of methylated histones available for interaction with the nuclear lamina, thereby affecting the size of the LADs (Fig. R14, right). These findings are experimentally validated (Fig. R14, left). Contrary to the reviewer's remark that “In fact, the authors find the GSK inhibitor they are using has no effect on LADs”, our model as well as experiments (Section 2.5, Figure 3(Figure 4 in older version of manuscript), reproduced here as Fig. R14) demonstrate that GSK343 treatment does indeed impact LAD organization by reducing LAD thickness.

Comment 2(i): I don't see how the authors can make implications of manipulating HDAC activity with TSA on “ the overall rate of histone methylation “ in the models. In fact, I simply do not see how the parameters tested possibly affect “methylation reactions”. This seems to be result from their modeling framework, but is not demonstrated by the biology and therefore of limited, if at any, value.

Our response: We thank the reviewer for their comment regarding the implications of HDAC activity manipulation with TSA on histone methylation and its representation in our modeling framework. We appreciate the opportunity to clarify the rationale behind this approach.

As explained in multiple sections of the main manuscript and reiterated in the general response, our model defines methylation rate as the rate of conversion of acetylated histones into methylated histones. This process involves the removal of acetyl groups (via deacetylation by enzymes such as HDAC) followed by the addition of methyl marks (via HMTs). HDAC inhibition by TSA reduces deacetylation, thereby lowering the methylation rate in our model.

Physiologically, HDAC inhibition limits the availability of unacetylated lysines, which are substrates for methylation. Previous studies have demonstrated that HDAC inhibition leads to a reduction in methylated histones [57, 58]. Additionally, we have experimentally shown that indirect modulation of HDAC3 levels through factors such as substrate stiffness, cell shape, and mechanical forces alters the levels of H3K9ac, H3K9me3, and H3K27me3 [45, 54].

Given this evidence, we respectfully disagree with the reviewer's assertion that this observation "seems to be a result of their modeling framework, but is not demonstrated by the biology." Our findings are supported by both experimental data and existing literature, which align well with the predictions of our modeling framework.

Comment 2(ii): If the authors do want to consider histone methylation in their modeling, then they should instead consider H3K9me2 or H3K9me3 methylation in (perhaps) driving (or contributing to) LAD properties in their models, as these modifications are largely found in constitutive bona fide LADs (in contrast to H3K27me3), in particular in the cell types they are considering.

Our response: We thank the reviewer for their insightful comment regarding the role of H3K9me2/3 in LAD organization. We would like to clarify that the STORM images analyzed in this study primarily rely on H2B labeling, which reflects the density of histones—particularly high in condensed LAD phases. Therefore, these images are agnostic to the specific epigenetic marks defining heterochromatin subtypes.

That said, we have confirmed that labeling with H3K9me2 and H3K27me3 produce visually similar results (as shown in Fig. R15). This consistency

demonstrates that inferring LAD properties from H2B labeling provides a reliable approximation of LAD organization associated with H3K9me2 or H3K27me3. Consequently, this supports the robustness of our generalized framework.

Our current model represents the collective contribution of all methylated histones, rather than focusing on specific marks like H3K9me2/3 or H3K27me3, ensuring applicability across different epigenetic contexts. This is important because while H3K9me2/3 are known LAD-specific marks [8], previous studies have also reported that H3K27me3 plays roles in peripheral chromatin organization [59-61]. In the future, we plan to explicitly model important epigenetic modifications as distinct species to yield deeper insights and carry out further experiments to deplete H3K9me2/3 to dissect the differential contribution of these distinct repressive modifications to LAD organization. At present, even without this distinction our

Figure 15: Representative images of LADs labeled using H3K9me2 and H3K27me3, confirms that analyzing the density of H2B can be reliably used to identify LAD domains.

generalized framework enables robust, treatment-specific inferences about chromatin-lamina interactions without requiring detailed experimental quantification of individual histone modifications.

Comment 3: Microscopy analyses: I am surprised by the very low number of nuclei analyzed ($n=4-10$ depending on the experiment and figure); how can the authors get statistics from so few nuclei? A lot more nuclei going into analyses would be warranted. The actual differences claimed in the plots are at times very minute and it is doubtful that they are biologically relevant.

Our response: We thank the reviewer for raising concerns regarding the number of nuclei analyzed and the biological relevance of the observed differences. We would like to clarify that all imaging data used in this study was obtained from previously published datasets in *Nature Biomedical Engineering* (Fig 3, 5-7 (Fig 4, 6-8 in older version of manuscript)) [54] and *Nucleic Acids Research* (Fig 4 (Fig 5 in older version of manuscript)) [50]. Consequently, we did not have the flexibility to increase the number of nuclei analyzed in this study. The previously published datasets have been peer-reviewed and deemed robust for similar quantitative analyses. In fact, since we are considering individual HC domains as a source of a single datapoint, the sample size for the statistical analyses we perform is of the order of 10^4 .

Secondly, the reviewer states that the “actual differences claimed in the plots are at times very minute”. On the other hand, our study consistently shows changes of 5% or higher in nuclear interior heterochromatin sizes and 10% or higher in LAD thickness (Table S4). Apart from being statistically significant, these magnitudes of change are well within the range of biologically meaningful variations in chromatin organization.

Comment 4: Voronoi plots resulting from STORM images shown are at times inconsistent between experiments or figures, showing significant variations in chromatin distribution in controls (e.g., Fig 4B control and Fig 3C control). Along the same lines, the GSK image in Fig 5B is very similar to the control in Fig 4B in terms of density. This should be clarified.

Our response: We thank the reviewer for their observations regarding the consistency across the figures in the manuscript. However, such inconsistencies are expected since these are different experiments performed on different cell types under different mechanobiological conditions, chosen to capture the specific biological effects of each perturbation. For instance, soft substrates (e.g., Fig. 3 (Fig 4 in older version of manuscript), GSK treatment) with highly condensed chromatin are ideal for GSK treatment to more clearly demonstrate the effect of chromatin decompaction. Similarly, stiff substrates (e.g., Fig 5 (Fig 6 in older version of manuscript), Y27 treatment), where cell contractility is high, were selected to examine the impact of Y27, which inhibits contractility.

Firstly, the reviewer compares *Fig 3B (Fig 4B in older version) control* and *Fig 4C (Fig 5C in older version) control* (we believe 3C is not relevant to the context and is a typo):

- **Figure 3B control:** Represents hMSCs cultured on soft substrates (3 kPa), imaged using super-resolution microscopy of H2B-labeled chromatin.
- **Figure 4C control:** Represents hFb cells cultured on glass substrate, imaged using super-resolution microscopy of EdC-labeled DNA.

To further the reviewer’s idea of cross-experiment comparison, when controls are compared within the same cell type and under similar experimental conditions, the chromatin distributions are indeed consistent. Here are two examples within our manuscript:

- In *Fig. 5*, LAD thickness distributions for hMSCs cultured on soft substrates (3 kPa) align closely with those presented in *Fig. 6* for hMSCs under the same biomechanical conditions.
- Similarly, LAD distributions for hMSCs cells cultured on stiff substrates (30 kPa) are consistent within their respective experimental contexts (*Fig. 5 vs. Fig. 6*).

Secondly, the reviewer compares *Fig. 3B (GSK-treated hMSCs)* and *Fig. 4C control (untreated hFb)*. To reiterate, these are different cell types under different conditions.

In conclusion, we want to stress that these figures are for distinct cell-lines, distinct biological contexts and distinct labeling (H2B v/s EdC) and therefore should not be directly compared.

Comment 5: Fig 4C is labeled “Quantification of STORM images”, but the two lower plots are modeling results on methylation rate. This should be clarified.

Our response: We agree that the current label could lead to confusion, as the two lower plots represent results derived from our theoretical framework rather than direct quantification of STORM images. We have now revised the figure label and the corresponding text in the manuscript as:

"Fig. 3C (Fig 4C in older version): (Top) Quantification of STORM image. (Bottom) Biophysical parameters inferred by the framework."

Additionally, in the main text (Section 2.5), we now explicitly clarify that the upper panels of Fig. 3C show direct image-based quantifications, whereas the lower panels represent model-inferred parameters.

Comment 6: That HDAC3 also interacts with lamin A/C is ignored in the paper. In fact, lamin A/C, which together with LBR is key protein involved in tethering heterochromatin at the nuclear lamina, is completely ignored in the manuscript.

Our response: We respectfully disagree with the reviewer. Far from being ignored, the interactions between HDAC3 and the lamin A/C in tethering heterochromatin at the nuclear lamina form the core of the paper and are quantitatively predicted via the chromatin-lamina interactions.

The chromatin-lamina affinity parameter accounts for the cumulative effects of chromatin-anchoring proteins such as LAP2 β , LBR, and HDAC3 in mediating the anchoring of methylated histones to the nuclear lamina. While our simulations are agnostic to the exact roles played by the individual proteins, none of these are ignored. For instance, our prediction that TSA treatment alters chromatin-lamina affinity is likely due to the HDAC3 inhibition resulting in reduction of its interactions with lamin A/C thus impairing the tethering of chromatin. Similarly, we consider the HDAC3-lamin A/C interactions to be the cause of changes in chromatin-lamina affinity upon changes in substrate stiffness in-vitro and tissue stiffness in tendinosis.

Comment 7: From the experiments done (HDAC activity manipulation with TSA, surface hardness, healthy vs tendinosis nuclei), Implications made by the authors on the importance of “methylation reactions” driving LAD morphology are completely unclear to me and possibly erroneous.

Comment 8 (i): Fig 7 and related text: the authors imply difference in “methylation rates” from increase surface stiffnesses which are very hard to believe.

Our response: As explained in the main manuscript and reiterated in general response above (Fig R9), our framework models overall histone methylation rate as the combined effects of histone methyltransferases (HMTs) and histone deacetylases (HDACs).

Perturbations in HDAC activity either directly via pharmacological treatments like TSA or indirectly via mechanobiological factors like substrate stiffness thus have a direct effect on the methylation rate. While TSA directly inhibits HDAC, substrate stiffness and diseases like tendinosis which soften the extracellular matrix have been previously shown to alter HDAC shuttling between the cytoplasm and the nucleus [53, 62], thus influencing the intranuclear HDAC availability.

Secondly, we theoretically show that LAD morphology is affected by both chromatin-lamina affinity (shape of LADs) as well as the overall methylation rate (size of LADs). For instance, Figure 6 specifically deals with changes in substrate stiffness. Softer substrates show more

HDAC3 within the nucleus (Fig 6A), which has two effects on LADs – (a) increase in the pool of methylated histones that can localize at the nuclear periphery, and (b) increase in the HDAC3 mediated interactions between chromatin and lamin A/C. Via both these pathways the LAD thickness is predicted to increase, exactly as observed experimentally. TSA which inhibits HDAC3 has the opposite effect which is also validated experimentally.

Comment 8(ii): I don't see from Fig 7D how they modeling show differences in HC densities. This is also the case for tendinosis nuclei vs controls later in the paper.

Our response: We thank the reviewer for this comment which is also highlighted by the first reviewer (minor comment 8).

In Fig. 6D (Fig 7D in older version of manuscript), the model predicts that the size of interior domains reduced by 7% and 15% on stiff and glass substrates as compared to soft substrates. From the experiments, the observed changes in sizes of interior domains were 6.8% and 8% respectively. Similarly, the model predicts that LAD thickness reduced by 15% and 25% on stiff and glass with respect to soft substrates. The experiments showed a respective change of 14% and 21.5%. Hence, there are significant differences between the sizes of interior and peripheral domains from the model predictions, although they may not be visible very clearly in the images.

To clarify this, we have more explicitly mentioned this comparison between experiments and simulations. Further, we have added a table in the SI (Table S4) to enumerate this comparison for each treatment. Moreover, the raw data associated with each figure has been provided in the file "Source_data.xlsx" for further cross-verification.

REFERENCES

1. Heo, S.J., et al., *Aberrant chromatin reorganization in cells from diseased fibrous connective tissue in response to altered chemomechanical cues*. Nat Biomed Eng, 2023. **7**(2): p. 177-191.
2. Nozaki, T., et al., *Condensed but liquid-like domain organization of active chromatin regions in living human cells*. Science Advances, 2023. **9**(14): p. eadf1488.
3. Waterborg, J.H., *Dynamics of histone acetylation in vivo. A function for acetylation turnover?* Biochemistry and cell biology, 2002. **80**(3): p. 363-378.
4. Haws, S.A., et al., *Intrinsic catalytic properties of histone H3 lysine-9 methyltransferases preserve monomethylation levels under low S-adenosylmethionine*. Journal of Biological Chemistry, 2023. **299**(7).
5. Zheng, Y., P.M. Thomas, and N.L. Kelleher, *Measurement of acetylation turnover at distinct lysines in human histones identifies long-lived acetylation sites*. Nature communications, 2013. **4**(1): p. 2203.
6. Zee, B.M., et al., *In vivo residue-specific histone methylation dynamics*. Journal of Biological Chemistry, 2010. **285**(5): p. 3341-3350.
7. Shah, P.P., et al., *An atlas of lamina-associated chromatin across twelve human cell types reveals an intermediate chromatin subtype*. Genome Biol, 2023. **24**(1): p. 16.
8. Briand, N. and P. Collas, *Lamina-associated domains: peripheral matters and internal affairs*. Genome Biol, 2020. **21**(1): p. 85.
9. Luo, L., et al., *The nuclear periphery of embryonic stem cells is a transcriptionally permissive and repressive compartment*. Journal of cell science, 2009. **122**(20): p. 3729-3737.
10. Poleshko, A., et al., *Genome-Nuclear Lamina Interactions Regulate Cardiac Stem Cell Lineage Restriction*. Cell, 2017. **171**(3): p. 573-587 e14.
11. Guelen, L., et al., *Domain organization of human chromosomes revealed by mapping of nuclear lamina interactions*. Nature, 2008. **453**(7197): p. 948-51.
12. Kind, J. and B. van Steensel, *Genome–nuclear lamina interactions and gene regulation*. Current opinion in cell biology, 2010. **22**(3): p. 320-325.

13. Harr, J.C., et al., *Directed targeting of chromatin to the nuclear lamina is mediated by chromatin state and A-type lamins*. J Cell Biol, 2015. **208**(1): p. 33-52.
14. Poleshko, A., et al., *H3K9me2 orchestrates inheritance of spatial positioning of peripheral heterochromatin through mitosis*. Elife, 2019. **8**: p. e49278.
15. Ricci, M.A., et al., *Chromatin fibers are formed by heterogeneous groups of nucleosomes in vivo*. Cell, 2015. **160**(6): p. 1145-1158.
16. Boettiger, A.N., et al., *Super-resolution imaging reveals distinct chromatin folding for different epigenetic states*. Nature, 2016. **529**(7586): p. 418-422.
17. Chiang, M., et al., *Predicting genome organisation and function with mechanistic modelling*. Trends in Genetics, 2022. **38**(4): p. 364-378.
18. Chiang, M., et al., *Genome-wide chromosome architecture prediction reveals biophysical principles underlying gene structure*. Cell Genomics, 2024. **4**(12).
19. Kant, A., et al., *Active transcription and epigenetic reactions synergistically regulate meso-scale genomic organization*. Nature Communications, 2024. **15**(1): p. 4338.
20. Tiana, G. and L. Giorgetti, *Modeling the 3D conformation of genomes*. 2019: Crc Press.
21. Vinayak, V., et al., *Polymer Model Integrates Super-Resolution Imaging and Epigenomic Sequencing to Elucidate the Role of Epigenetic Reactions in Shaping 4D Chromatin Organization*. bioRxiv, 2024: p. 2024.10.08.617296.
22. Zhang, Y., et al., *Computational methods for analysing multiscale 3D genome organization*. Nature Reviews Genetics, 2024. **25**(2): p. 123-141.
23. Zhou, R. and Y.Q. Gao, *Polymer models for the mechanisms of chromatin 3D folding: review and perspective*. Physical Chemistry Chemical Physics, 2020. **22**(36): p. 20189-20201.
24. Le Treut, G., F. Képès, and H. Orland, *A polymer model for the quantitative reconstruction of chromosome architecture from HiC and GAM data*. Biophysical journal, 2018. **115**(12): p. 2286-2294.
25. Laghmach, R. and D.A. Potoyan, *Liquid-liquid phase separation driven compartmentalization of reactive nucleoplasm*. Physical biology, 2020. **18**(1): p. 015001.
26. Barbieri, M., et al., *Complexity of chromatin folding is captured by the strings and binders switch model*. Proceedings of the National Academy of Sciences, 2012. **109**(40): p. 16173-16178.
27. Fudenberg, G., et al., *Formation of chromosomal domains by loop extrusion*. Cell reports, 2016. **15**(9): p. 2038-2049.
28. Chiang, M., et al., *Polymer modeling predicts chromosome reorganization in senescence*. Cell reports, 2019. **28**(12): p. 3212-3223. e6.
29. Maji, A., et al., *A lamin-associated chromatin model for chromosome organization*. Biophysical Journal, 2020. **118**(12): p. 3041-3050.
30. van Steensel, B. and A.S. Belmont, *Lamina-Associated Domains: Links with Chromosome Architecture, Heterochromatin, and Gene Repression*. Cell, 2017. **169**(5): p. 780-791.
31. Kind, J., et al., *Genome-wide maps of nuclear lamina interactions in single human cells*. Cell, 2015. **163**(1): p. 134-47.
32. Kind, J., et al., *Single-cell dynamics of genome-nuclear lamina interactions*. Cell, 2013. **153**(1): p. 178-192.
33. Rooijers, K., et al., *Simultaneous quantification of protein-DNA contacts and transcriptomes in single cells*. Nature biotechnology, 2019. **37**(7): p. 766-772.
34. Pascual-Reguant, L., et al., *Lamin B1 mapping reveals the existence of dynamic and functional euchromatin lamin B1 domains*. Nature communications, 2018. **9**(1): p. 3420.
35. Sadaie, M., et al., *Redistribution of the Lamin B1 genomic binding profile affects rearrangement of heterochromatic domains and SAHF formation during senescence*. Genes & development, 2013. **27**(16): p. 1800-1808.
36. Nguyen, H.Q., et al., *3D mapping and accelerated super-resolution imaging of the human genome using in situ sequencing*. Nature methods, 2020. **17**(8): p. 822-832.

37. Marin, H., et al., *The nuclear periphery confers repression on H3K9me2-marked genes and transposons to shape cell fate*. bioRxiv, 2024.
38. Amiad-Pavlov, D., et al., *Live imaging of chromatin distribution reveals novel principles of nuclear architecture and chromatin compartmentalization*. Science advances, 2021. **7**(23): p. eabf6251.
39. Brunet, A., N. Destainville, and P. Collas, *Physical constraints in polymer modeling of chromatin associations with the nuclear periphery at kilobase scale*. Nucleus, 2021. **12**(1): p. 6-20.
40. Falk, M., et al., *Heterochromatin drives compartmentalization of inverted and conventional nuclei*. Nature, 2019. **570**(7761): p. 395-399.
41. Laghmach, R., M. Di Pierro, and D. Potoyan, *A liquid state perspective on dynamics of chromatin compartments*. Frontiers in Molecular Biosciences, 2022. **8**: p. 781981.
42. Tolokh, I.S., et al., *Strong interactions between highly dynamic lamina-associated domains and the nuclear envelope stabilize the 3D architecture of Drosophila interphase chromatin*. Epigenetics & Chromatin, 2023. **16**(1): p. 21.
43. Ulianov, S.V., et al., *Nuclear lamina integrity is required for proper spatial organization of chromatin in Drosophila*. Nat Commun, 2019. **10**(1): p. 1176.
44. Turgay, Y., et al., *The molecular architecture of lamins in somatic cells*. Nature, 2017. **543**(7644): p. 261-264.
45. Alisafaei, F., et al., *Regulation of nuclear architecture, mechanics, and nucleocytoplasmic shuttling of epigenetic factors by cell geometric constraints*. Proceedings of the National Academy of Sciences, 2019. **116**(27): p. 13200-13209.
46. Damodaran, K., et al., *Compressive force induces reversible chromatin condensation and cell geometry-dependent transcriptional response*. Molecular biology of the cell, 2018. **29**(25): p. 3039-3051.
47. Czapiewski, R., M.I. Robson, and E.C. Schirmer, *Anchoring a Leviathan: How the Nuclear Membrane Tethers the Genome*. Front Genet, 2016. **7**: p. 82.
48. Manzo, S.G., L. Dauban, and B. van Steensel, *Lamina-associated domains: Tethers and looseners*. Curr Opin Cell Biol, 2022. **74**: p. 80-87.
49. Nmezi, B., et al., *Concentric organization of A- and B-type lamins predicts their distinct roles in the spatial organization and stability of the nuclear lamina*. Proc Natl Acad Sci U S A, 2019. **116**(10): p. 4307-4315.
50. Otterstrom, J., et al., *Super-resolution microscopy reveals how histone tail acetylation affects DNA compaction within nucleosomes in vivo*. Nucleic acids research, 2019. **47**(16): p. 8470-8484.
51. Jain, N., et al., *Cell geometric constraints induce modular gene-expression patterns via redistribution of HDAC3 regulated by actomyosin contractility*. Proc Natl Acad Sci U S A, 2013. **110**(28): p. 11349-54.
52. Mierke, C.T., *Extracellular matrix cues regulate mechanosensing and mechanotransduction of cancer cells*. Cells, 2024. **13**(1): p. 96.
53. Toscano-Marquez, F., et al., *Absence of HDAC3 by matrix stiffness promotes chromatin remodeling and fibroblast activation in idiopathic pulmonary fibrosis*. Cells, 2023. **12**(7): p. 1020.
54. Heo, S.-J., et al., *Biophysical regulation of chromatin architecture instills a mechanical memory in mesenchymal stem cells*. Scientific reports, 2015. **5**(1): p. 16895.
55. Song, Y., et al., *Cell engineering: biophysical regulation of the nucleus*. Biomaterials, 2020. **234**: p. 119743.
56. Wang, P., et al., *Substrate stiffness dominates cell gene expression via regulation of HDAC3 subcellular localization*. Colloid and Interface Science Communications, 2023. **55**: p. 100719.
57. Ganai, S.A., S. Malli Kalladi, and V. Mahadevan, *HDAC inhibition through valproic acid modulates the methylation profiles in human embryonic kidney cells*. Journal of Biomolecular Structure and Dynamics, 2015. **33**(6): p. 1185-1197.

58. Legoff, L., et al., *Histone deacetylase inhibition leads to regulatory histone mark alterations and impairs meiosis in oocytes*. *Epigenetics & Chromatin*, 2021. **14**: p. 1-16.
59. Lochs, S.J., S. Kefalopoulou, and J. Kind, *Lamina associated domains and gene regulation in development and cancer*. *Cells*, 2019. **8**(3): p. 271.
60. Martin, C.J., et al., *Distinct Classes of Lamin-Associated Domains are Defined by Differential Patterns of Repressive Histone Methylation*. *bioRxiv*, 2024: p. 2024.12.20.629719.
61. Wu, F. and J. Yao, *Identifying novel transcriptional and epigenetic features of nuclear lamina-associated genes*. *Scientific reports*, 2017. **7**(1): p. 100.
62. Subramanian, A. and T.F. Schilling, *Tendon development and musculoskeletal assembly: emerging roles for the extracellular matrix*. *Development*, 2015. **142**(24): p. 4191-4204.

We again sincerely thank the referees for their careful review of the manuscript, and their constructive feedback on it. We have now addressed the remaining concerns raised by the referees below. (The reviewer comments are in blue, our response in plain text, and the edits made to the manuscript are highlighted in yellow)

Reviewer #1

The reviewer acknowledges the authors for improving their manuscript by addressing most of my concerns.

There are still points that was eluded in the first round of revision which should be discussed in the main text:

General response: We thank the reviewer for their thoughtful reassessment of our manuscript and for acknowledging our efforts in the previous revision. We greatly appreciate their continued engagement and constructive feedback, which have further improved the clarity and rigor of our work. In this revision, we have addressed the remaining points, including clarifying modeling assumptions and limitations, refining the interpretation of HDAC3's role, and improving the precision of theoretical definitions and figure citations, as detailed below.

Comment 1: On comment 1 of the previous review: authors don't fully address the reviewer's comment on the source of stochasticity. The point made by the reviewer should be discuss in the Discussion as a limitation of the current model.

Our response: We thank the reviewer for revisiting this important point. We acknowledge that the assumption of spatial heterogeneities in molecular parameters—such as epigenetic reaction rates and chromatin-lamina affinities—as the dominant source of variation in HC domain sizes and LAD thicknesses represents a modeling choice that simplifies the complex landscape of nuclear organization. While our experimental data (Figures S15) and sensitivity analyses (Figure S16) support the significant role of epigenetic reactions in shaping chromatin architecture, we agree that other intrinsic sources of stochasticity—such as variability in genomic domain lengths, 3D folding due to loop extrusion, and domain-domain interactions—may also contribute to the observed heterogeneity.

As noted in the previous version of the manuscript, these alternative sources of variability were discussed in SI Section S8. In response to the reviewer's suggestion, we have now explicitly incorporated this discussion into the main text, adding a paragraph to the *Discussion* section. This new paragraph acknowledges the limitations of our current model and emphasizes that our minimal parameter framework is designed to extract mechanistic insights, while recognizing that future model extensions could incorporate additional sources of heterogeneity to capture the full biological complexity.

Change in manuscript: The current formulation, however, is based on simplifying assumptions. Specifically, it attributes spatial heterogeneity in chromatin organization primarily to variability in molecular parameters, while omitting other plausible contributors such as genomic domain length differences, loop extrusion dynamics, and three-dimensional domain-domain interactions. Incorporating these features in future models could enhance both descriptive power and predictive accuracy. Additionally, future refinements could replace the average chromatin–lamina affinity with protein-specific interaction coefficients (e.g., LAP2 β , LBR, emerin, PRR14), leveraging multiplexed imaging and knockout experiments to delineate the roles of individual

nuclear envelope proteins in chromatin organization. While our current analysis focuses on steady-state chromatin architecture, the underlying reaction-diffusion model can be extended to simulate time-dependent dynamics. This opens avenues to study transient chromatin remodeling events in response to temporally varying mechanical or biochemical cues, particularly when paired with dynamic imaging experiments.

Comment 2: Comment 6: while the authors have improved a lot the Discussion part, it lacks a part where they discuss the limitations of their approach and the (sometimes strong) assumptions made along the paper.

Our response: We thank the reviewer for this insightful suggestion. While we had aimed to strengthen the *Discussion* section in the revised manuscript, we agree that a clear and explicit acknowledgment of the limitations of our approach and the underlying modeling assumptions was still missing. In response, we have now added a paragraph which focuses on limitations and future directions at the end of the *Discussion*. This section addresses the following key points:

1. The simplifications made in modeling chromatin organization using a minimal set of parameters—specifically, the assumption that spatial heterogeneity in heterochromatin domain sizes and LAD thicknesses primarily arises from variations in epigenetic reaction rates and chromatin-lamina affinities;
2. The focus on steady-state spatial organization, while noting that the framework is capable of capturing dynamic reorganization -- an aspect that could be explored in future studies alongside experiments capable of measuring the time dependence of chromatin organization;
3. The possibility of extending the model to incorporate gene-level resolution by integrating it with fine-grained polymer models and locus-specific imaging techniques.

Change in manuscript: Our predictive framework integrates image quantification with parameter-extraction algorithms and is broadly applicable beyond STORM to other super-resolution techniques capable of visualizing heterochromatin domains. It enables unbiased inference of nucleus-wide biophysical parameters without requiring prior knowledge of nuclear stimuli. This makes it particularly valuable in contexts where chromatin remodeling mechanisms remain elusive, allowing the model to illuminate how epigenetic reactions and chromatin-lamina interactions contribute to the formation of heterochromatin domains both in the nuclear interior and at the periphery. To maintain computational tractability and interpretability, we employ a minimal set of spatially varying molecular parameters—namely, epigenetic reaction rates and chromatin-lamina affinity. Despite its simplicity, the model captures the stochastic and heterogeneous organization of chromatin observed at the single-cell level, with predictive accuracy validated by strong agreement between simulations and experimental observations.

The current formulation, however, is based on simplifying assumptions. Specifically, it attributes spatial heterogeneity in chromatin organization primarily to variability in molecular parameters, while omitting other plausible contributors such as genomic domain length differences, loop extrusion dynamics, and three-dimensional domain-domain interactions. Incorporating these features in future models could enhance both descriptive power and predictive accuracy. Additionally, future refinements could replace the average chromatin-lamina affinity with protein-specific interaction coefficients (e.g., LAP2 β , LBR, emerin, PRR14), leveraging multiplexed imaging and knockout experiments to delineate the roles of individual nuclear envelope proteins in chromatin organization. While our current analysis focuses on steady-state chromatin

architecture, the underlying reaction-diffusion model can be extended to simulate time-dependent dynamics. This opens avenues to study transient chromatin remodeling events in response to temporally varying mechanical or biochemical cues, particularly when paired with dynamic imaging experiments.

Looking ahead, coupling this modeling framework with polymer simulations could enable high-resolution depictions of genome organization at the level of individual chromatin compartments and gene loci. When integrated with imaging techniques such as FISH, Oligopaint, and other forms of super-resolution microscopy [75,76], the framework can help identify genes regulated via epigenetic mechanisms or chromatin-lamina interactions. These capabilities hold diagnostic and therapeutic promise, offering routes to modulate gene expression in response to environmental changes by targeting chromatin structure near the periphery or within the nuclear interior. Overall, our multi-modal parameter extraction approach lays a foundational platform for investigating how environmental stimuli influence chemo-mechanical nucleo-cytoskeletal signaling and drive mesoscale chromatin reorganization throughout the nucleus.

Comment 3: Comment 7: at several places in the Results section, authors claim that HDAC3 is driving the change lamina-chromatin interactions. These claims or statements should be relocated in the Discussion as authors just show correlation (levels of HDAC 3 change & lamina-chromatin interactions change, but the causal relationship is not demonstrated, it is therefore fair to add it to the discussion but not in Results).

Our response: We thank the reviewer for this important clarification. We agree that our study does not include direct experimental perturbation of HDAC3 (e.g., through specific inhibition or knockdown) to isolate causality. However, we have imaged nuclear HDAC3 levels and observed consistent reductions under conditions that also show decreased LAD thickness and heterochromatin domain sizes. Our theoretical framework mechanistically links reduced HDAC3 to a decrease in methylation rates, which in turn leads to smaller domains — providing a model-based causal pathway connecting HDAC3 levels to chromatin reorganization. Moreover, prior studies have directly implicated HDAC3 in LAD formation through its interactions with lamina-associated proteins such as LAP2 β [1-3] and report global chromatin decondensation, which aligns with our observations, although not at the spatial resolution captured by STORM. Taken together, this body of evidence — combining model-informed predictions and literature precedent — supports a plausible causal role for HDAC3 in regulating chromatin-lamina interactions.

That said, we agree with the reviewer that other HDAC variants may also contribute to chromatin-lamina tethering. Our model is agnostic to specific HDAC variants and interprets nuclear HDAC levels functionally, in terms of their impact on histone deacetylation and methylation kinetics. Therefore, while our model, in conjunction with previous literature suggest a mechanistic role for HDAC3, the observed effects may reflect the collective activity of multiple HDAC family members. In response to the reviewer's concern, we have carefully revised the Results section to avoid any "causal" language. All interpretive discussion regarding HDAC3's potential role has been relocated to the Discussion section, where it is explicitly framed as a supported hypothesis rather than a demonstrated mechanism.

Change in manuscript:

Result section:

..The observed increase in nuclear HDAC3 due to the loss of contractility (Fig 5E) correlates with increased histone deacetylation, which in turn is associated with a higher predicted histone

methylation rate in our model. The implications of HDAC3 nuclear localization for chromatin-lamina interactions are further discussed in the Discussion section...

..The observed decrease in nuclear HDAC3 with increasing substrate stiffness (Fig 6C) is associated with reduced histone deacetylation, which is reflected in a lower predicted Γ_{me} in our model. The potential role of HDAC3 in mediating chromatin-lamina interactions and LAD organization is discussed in the Discussion section...

Discussion Section:

Previously, it has been shown that via anchoring proteins such as emerin, Lamin A/C is implicated in heterochromatin tethering to the lamina [69,70] and that stiff tissue environment drives an increase in Lamin A/C[71]. However, we observe a lower chromatin lamina affinity on stiffer substrates. Thus, our model suggests HDAC3 whose nuclear levels are reduced by cytoskeletal tension [49, 50], plays a relatively more prominent role in the formation of LADs as compared to Lamin A/C, at least in the cell-culture conditions described in Materials and Methods Section 4.4. In agreement with this conclusion, previous work showed that knockdown of Lamin proteins does not lead to changes in LADs in mouse embryonic stem cells[72]. While our data and previous work point to a mechanistic role for HDAC3, other HDAC isoforms may also contribute to chromatin-lamina tethering. Our model is agnostic to HDAC subtype and functionally interprets nuclear HDAC levels through their influence on histone deacetylation and methylation kinetics. Thus, the observed changes in chromatin-lamina affinity may reflect the combined activity of multiple HDAC family members. In the future, more specific inhibition of individual HDACs as well as targeting enzymes responsible for alternative repressive histone methylation marks such as H3K9me2/3 coupled with our predictive modeling will help elucidate the individual contributions of these epigenetic mechanisms to the formation of LADs.

Furthermore, our predictive framework links the cytoskeletal contractility to nuclear chromatin organization revealing an increase both in epigenetic reaction rates and chromatin-lamina affinity on soft substrates, where nuclear HDAC3 levels are higher, compared to stiff. These findings align with our prior reports that on stiff substrates, increased cytoskeletal contractility results in the shuttling of HDAC3 from the nucleus into the cytoplasm in mouse fibroblasts [50,73]. This nuclear export of HDAC3 correlates with reduced chromatin-lamina interactions and altered chromatin organization.

Comment 4: Authors have moved the Material & Methods section at the end of the paper (which is the standard format in Nature Com). A small paragraph at the beginning of Part 2.2 explaining briefly the model would be useful.

Our response: We thank the reviewer for this helpful suggestion. While the original manuscript did include an introductory paragraph outlining the key components of our model at the beginning of Section 2.2, we agree that it can be made clearer and more effective in guiding the reader. In response, we have revised and expanded this paragraph to better highlight the model's biological motivation and its core elements—including chromatin-chromatin and chromatin-lamina interactions, nucleoplasm diffusion, and spatially varying epigenetic kinetics—before presenting the detailed theoretical formulation.

Change in manuscript: To investigate how nuclear-scale chromatin organization emerges from local biophysical and biochemical interactions, we develop a reaction-diffusion-based mesoscale model of chromatin organization. This mathematical model incorporates the energetics of chromatin-chromatin and chromatin-lamina interactions coupled with the kinetics of nucleoplasm diffusion and spatially inhomogeneous interconversion of acetylated and methylated histones via epigenetic regulation (Fig 1C). The coupled energetic and kinetic contributions lead to a far-from-equilibrium segregation of chromatin into compacted heterochromatin domains in the interior as

well as at the periphery of the nucleus. We integrate our theoretical predictions with the experimentally observed chromatin organization quantified via super-resolution STORM imaging to extract the genome-wide distributions of key biophysical parameters, including the strength of chromatin-lamina interactions and methylation rates.

- There are still misleading or unclear statements:

Comment 5: “in the interior of the nucleus shows a characteristic mean (FigS11), in excellent agreement...” the first part of the sentence refers to interior HC while Fig.S11 deals with Tlad

Our response: We thank the reviewer for catching this inconsistency. The figure reference was indeed incorrect. We have corrected the reference to Figure S13, which presents the relevant data on interior heterochromatin domain sizes.

Change in manuscript: Our model predicts that at steady-state, the size distribution of the heterochromatin domains in the interior of the nucleus shows a characteristic mean (Fig S13), in excellent agreement with the size-distribution of condensed chromatin domains observed via STORM imaging (Fig 1B). Importantly, the model also predicts that near the nuclear periphery, LADs form either discrete domains or continuous layers (Fig 2B) and show a size distribution that peaks at a characteristic length-scale, as shown in (Fig S13).

Comment 6: “... is the far-field heterochromatin concentration...” : please define far-field

Our response: We thank the reviewer for pointing out the need to clarify the term “*far-field heterochromatin concentration*.” In the revised manuscript, we have now explicitly defined *far-field* at its first occurrence in the main text. In our theoretical framework, this refers to the steady-state heterochromatin concentration in the euchromatin region far (i.e. at distances larger than the heterochromatin domain size scale) from the heterochromatin domain. This is schematically shown in Fig S2 and discussed in detail in SI Section S4 , and we have now explicitly referenced this section in the main text.

Change in manuscript: $\bar{\phi}_h^*$ is the far-field heterochromatin concentration- that is, the steady-state heterochromatin concentration in the euchromatin regions distant from heterochromatin domain. A more detailed discussion of $\bar{\phi}_h^*$ including how it is affected by peripheral sequestration, is provided in Section S4 of the SI (see also Fig. S2).

Comment 7: “...using the theoretical framework are accurate despite ...”: what accurate means here?

Our response: We thank the reviewer for this important point. In this context, the term “*accurate*” refers to the predictive consistency between the theoretical framework and experimental observations. Specifically, the mean values of the extracted biophysical parameters (Γ_{me} and V_{LAD}), when used in numerical simulations of chromatin organization, result in predicted changes in LAD and interior domain size-scales that closely align with those quantified from STORM imaging data (see Table S4). This comparison validates the parameter extraction process despite the presence of simplifying assumptions, the extracted parameters yield a consistent result with numerical simulations which are independent of such assumptions. We have revised the manuscript text to clarify this and avoid any ambiguity regarding the use of the term.

Change in manuscript: Further, to assess the reliability of the extracted distributions of Γ_{me} and V_{LAD} , we compare simulation-based predictions of chromatin organization with experimental observations, despite the simplifying assumptions involved in the theoretical derivation of Eqs. 1–2 and S36 (numerically depicted in Fig. 2E). Specifically, the mean values of the extracted parameters Γ_{me} and V_{LAD} are used as inputs to simulate chromatin organization in the nucleus via numerical solutions of the phase-field equations (Eqs. 3–5) using COMSOL Multiphysics. The pharmacological or biophysical perturbation driven changes in the distributions of biophysical parameters, extracted from the theoretical framework, are incorporated into the simulation by a proportionate change in the mean of the parameters Γ_{me} and V_{LAD} . We then quantify the simulation-predicted chromatin reorganization and assess the relative changes in LAD and interior domain sizes, validating these against STORM-imaged data (Fig. 3D, 4C, 5B, 6D, 7C; Table S4). To incorporate spatial heterogeneity, local values of Γ_{me} and V_{LAD} are sampled from a normal distribution, whose mean matches the values obtained from STORM observations. This approach allows for biologically realistic variability in the simulated biophysical landscape while preserving consistency with the experimentally inferred averages. Heterochromatin domains in the simulation emerge naturally from the interplay of spatially varying parameters and reaction-diffusion dynamics. These domains can also interact freely with one another, capturing the intrinsically emergent behavior characteristic of chromatin organization.

Comment 8: "...spatially sampled from a normal distribution...": it is unclear if authors use the average parameter values (inferred from the STORM data) or draw them from a normal distribution.

Our response: We appreciate the reviewer's request for clarification. In our numerical simulations, we incorporate spatial heterogeneity by sampling local values of the methylation rate (Γ_{me}) and the chromatin-lamina interaction strength (V_{LAD}) from a normal distribution. The mean of this distribution is set to the corresponding value extracted from experimental STORM data using our framework. This allows the spatial variation in parameter values to reflect biologically realistic heterogeneity, while anchoring the simulations in experimentally grounded mean estimates. We have now revised the manuscript to explicitly describe this sampling approach.

Change in manuscript: Further, to assess the reliability of the extracted distributions of Γ_{me} and V_{LAD} , we compare simulation-based predictions of chromatin organization with experimental observations, despite the simplifying assumptions involved in the theoretical derivation of Eqs. 1–2 and S36 (numerically depicted in Fig. 2E). Specifically, the mean values of the extracted parameters Γ_{me} and V_{LAD} are used as inputs to simulate chromatin organization in the nucleus via numerical solutions of the phase-field equations (Eqs. 3–5) using COMSOL Multiphysics. The pharmacological or biophysical perturbation driven changes in the distributions of biophysical parameters, extracted from the theoretical framework, are incorporated into the simulation by a proportionate change in the mean of the parameters Γ_{me} and V_{LAD} . We then quantify the simulation-predicted chromatin reorganization and assess the relative changes in LAD and interior domain sizes, validating these against STORM-imaged data (Fig. 3D, 4C, 5B, 6D, 7C; Table S4). To incorporate spatial heterogeneity, local values of Γ_{me} and V_{LAD} are sampled from a normal distribution, whose mean matches the values obtained from STORM observations. This approach allows for biologically realistic variability in the simulated biophysical landscape while preserving consistency with the experimentally inferred averages. Heterochromatin domains in the simulation emerge naturally from the interplay of spatially varying parameters and reaction-diffusion dynamics. These domains can also interact freely with one another, capturing the intrinsically emergent behavior characteristic of chromatin organization.

Comment 9: “The predicted spectrum of....aligns qualitatively....”: experimentally, chromatin-LaminB1 interaction strengths are not observed using ChIP. Here the authors make the assumption that strong predicted interactions correspond to Type 1 LAD and intermediate predictions to Type 2 LAD, this is an assumption (more or less justified) but it is not a demonstration or a qualitative alignment.

Our response: We thank the reviewer for this important clarification. We agree that Lamin B1 ChIP-seq does not directly quantify physical interaction strength. Given this, our interpretation is informed by the classification reported in Shah et al. (Genome Biology, 2023), where LAD subtypes—Type 1 LADs (T1-LADs), Type 2 LADs (T2-LADs), and non-LADs—are defined in descending order of Lamin B1 occupancy using a three-state hidden Markov model applied to LB1 ChIP-seq data across twelve human cell types. Based on this, we interpret regions of high predicted chromatin-lamina affinity as corresponding to T1-LADs and intermediate affinity regions to T2-LADs. Nonetheless, we have revised the manuscript to explicitly acknowledge that this comparison is an inferred correspondence grounded in qualitative similarity, rather than a direct experimental validation of interaction strength.

Change in manuscript: We emphasize that this comparison is an inferred mapping based on qualitative similarity in the predicted spatial distribution of chromatin-lamina affinities and the genomic distribution of Lamin B1-associated domains, rather than a direct measure of interaction strength.

Comment 10: A recent Hi-C work on TSA was released (Paldi et al., Cavalli, bioRxiv 2024). Could be interesting to discuss it.

Our response: We thank the reviewer for highlighting this insightful work. While Paldi et al. explore the effects of HDAC inhibition across timescales, their study primarily focuses on changes in Hi-C contact maps and H3K27ac modifications, without addressing LADs. Although we find their findings to be a valuable perspective on TSA-induced chromatin reorganization, the overlap with our study is limited. Nevertheless, we now cite this work in Section 2.5 to acknowledge its relevance and complementarity.

Change in manuscript: Complementing our LAD-focused imaging and modeling approach, a recent Hi-C study [47] reported widespread TSA-induced changes in chromatin contact patterns and histone acetylation, providing a genome-wide perspective on the multiscale effects of HDAC inhibition

Comment 11: “our numerical simulations can also capture the time-dependent dynamics...”: not dynamical results are shown in the paper, so unclear if this statement is justified.

Our response: We appreciate the reviewer’s careful attention to this point. While our modeling framework is capable of simulating time-dependent chromatin dynamics (as indicated by the equation 4-5), we do not present dynamical results in the current manuscript. Instead, we focus on steady-state solutions of the dynamic equations, which are most relevant to the experiments. To clarify this, we have moved the corresponding statement to the newly added “Limitations and Future Directions” paragraph, where it is now clearly framed as a potential application of the framework in future studies.

Change in manuscript: our current analysis focuses on steady-state chromatin architecture, the underlying reaction-diffusion model can be extended to simulate time-dependent dynamics. This

opens avenues to study transient chromatin remodeling events in response to temporally varying mechanical or biochemical cues, particularly when paired with dynamic imaging experiments.

Reviewer #2

The authors have made a very significant effort to address my comments by providing detailed and documented explanations and by revising the manuscript where relevant. I have no further comments.

General response: We sincerely thank the reviewer for their positive assessment and for their constructive feedback throughout the review process. We are glad that the revised manuscript has addressed all concerns, and we appreciate their acknowledgment of the improvements made.

1. Alisafaei, F., et al., *Regulation of nuclear architecture, mechanics, and nucleocytoplasmic shuttling of epigenetic factors by cell geometric constraints*. Proceedings of the National Academy of Sciences, 2019. **116**(27): p. 13200-13209.
2. Jain, N., et al., *Cell geometric constraints induce modular gene-expression patterns via redistribution of HDAC3 regulated by actomyosin contractility*. Proc Natl Acad Sci U S A, 2013. **110**(28): p. 11349-54.
3. Somech, R., et al., *The nuclear-envelope protein and transcriptional repressor LAP2beta interacts with HDAC3 at the nuclear periphery, and induces histone H4 deacetylation*. J Cell Sci, 2005. **118**(Pt 17): p. 4017-25.

We again sincerely thank the referees for their careful review of the manuscript, and their constructive feedback on it. We have now addressed the remaining concerns raised by the referees below. (The reviewer comments are in blue, our response in plain text, and the edits made to the manuscript are highlighted in yellow)

Reviewer #1

The authors have addressed my concerns. Revisions lead to a very nice paper.

General response: We sincerely thank the reviewer for their positive assessment and for their constructive feedback throughout the review process. We are glad that the revised manuscript has addressed all concerns, and we appreciate their acknowledgment of the improvements made.